# Human ribonuclease 1 serves as a secretory ligand of ephrin A4 receptor and induces breast tumor initiation

Heng-Huan Lee [1,16], Ying-Nai Wang [1,16], Wen-Hao Yang [1,2,16], Weiya Xia[1], Yongkun Wei[1], Li-Chuan Chan[1], Yu-Han Wang[1,2], Zhou Jiang [1], Shouping Xu[3], Jun Yao[1], Yufan Qiu[1,4], Yi-Hsin Hsu[1], Wei-Lun Hwang[5,6], Meisi Yan[1,7], Jong-Ho Cha [1,8], Jennifer L. Hsu[1], Jia Shen[1], Yuanqing Ye[9], Xifeng Wu[9], Ming-Feng Hou[10,11], Lin-Ming Tseng[12,13], Shao-Chun Wang [2], Mei-Ren Pan[11], Chin-Hua Yang[12], Yuan-Liang Wang[2], Hirohito Yamaguchi[1,2], Da Pang[3], Gabriel N. Hortobagyi [14], Dihua Yu[1] & Mien-Chie Hung [2,1,15 ✉]

Human ribonuclease 1 (hRNase 1) is critical to extracellular RNA clearance and innate immunity to achieve homeostasis and host defense; however, whether it plays a role in cancer remains elusive. Here, we demonstrate that hRNase 1, independently of its ribonucleolytic activity, enriches the stem-like cell population and enhances the tumor-initiating ability of breast cancer cells. Specifically, secretory hRNase 1 binds to and activates the tyrosine kinase receptor ephrin A4 (EphA4) signaling to promote breast tumor initiation in an autocrine/paracrine manner, which is distinct from the classical EphA4-ephrin juxtacrine signaling through contact-dependent cell-cell communication. In addition, analysis of human breast tumor tissue microarrays reveals a positive correlation between hRNase 1, EphA4 activation, and stem cell marker CD133. Notably, high hRNase 1 level in plasma samples is positively associated with EphA4 activation in tumor tissues from breast cancer patients, highlighting the pathological relevance of the hRNase 1-EphA4 axis in breast cancer. The discovery of hRNase 1 as a secretory ligand of EphA4 that enhances breast cancer stemness suggests a potential treatment strategy by inactivating the hRNase 1-EphA4 axis.

[1] Department of Molecular and Cellular Oncology, The University of Texas MD Anderson Cancer Center, Houston, TX, USA. [2] Graduate Institute of Biomedical Sciences, Research Center for Cancer Biology, and Center for Molecular Medicine, China Medical University, Taichung, Taiwan. [3] Department of Breast Surgery, Harbin Medical University Cancer Hospital, Harbin, China. [4] Deptartment III of Breast Surgery, Tianjin Medical University Cancer Institute and Hospital, Tianjin, China. [5] Department of Biotechnology and Laboratory Science in Medicine, National Yang Ming Chiao Tung University, Hsinchu, Taiwan. [6] Department of Biotechnology and Laboratory Science in Medicine, National Yang-Ming University, Taipei, Taiwan. [7] Department of Pathology, Harbin Medical University, Harbin, China. [8] Department of Biomedical Sciences, College of Medicine, Inha University, Incheon, Korea. [9] Department of Epidemiology, The University of Texas MD Anderson Cancer Center, Houston, TX, USA. [10] Division of Breast Surgery, Department of Surgery, Kaohsiung Medical University Hospital, Kaohsiung, Taiwan. [11] Graduate Institute of Clinical Medicine, Kaohsiung Medical University, Kaohsiung, Taiwan. [12] Comprehensive Breast Health Center and Division of General Surgery, Department of Surgery, Taipei Veterans General Hospital, Taipei, Taiwan. [13] Institute of Clinical Medicine and Faculty of Medicine, School of Medicine, National Yang-Ming University, Taipei, Taiwan. [14] Department of Breast Medical Oncology, The University of Texas MD Anderson Cancer Center, Houston, TX, USA. [15] Department of Biotechnology, Asia University, Taichung, Taiwan. [16] These authors contributed equally: Heng-Huan Lee, Ying-Nai Wang, Wen-Hao Yang. ✉email: mhung@cmu.edu.tw

The pancreatic hRNase A superfamily is comprised of eight canonical hRNases, including human ribonuclease 1 (hRNase 1), and five noncanonical hRNases, all of which can be detected in human plasma/serum[1,2]. Secretory hRNase 1 is found in various organs[3,4], and its biochemical properties and post-translational modifications, such as glycosylation, have been extensively investigated[5–7]. Nonetheless, the biological function of hRNase 1 has not yet been completely defined. Studies on bovine homolog RNase A indicated that its human homolog hRNase 1 also plays a role in the digestive system by degrading dietary RNAs[4,8]. Moreover, hRNase 1 regulates hemostasis, inflammation, and innate immunity[2], indicating that hRNase 1 possesses multiple functions in addition to RNA clearance[9–12]. Recent studies indicated that hRNase 5 serves as a ligand for the receptor tyrosine kinase (RTK) epidermal growth factor receptor (EGFR) and plexin-B2 (PLXNB2) receptor[13–15] in solid and hematopoietic cancers. Together, those findings reveal an unconventional ligand–receptor relationship and a role of the hRNase A superfamily in tumor progression[16–18].

The family of erythropoietin-producing hepatocellular carcinoma (Eph) receptors is the largest among the RTKs with important functions in embryonic development and adult tissue homeostasis[19]. Eph receptors are traditionally thought to bind to their membrane-bound ligands, ephrins, on neighboring cells to stimulate juxtacrine signals through contact-dependent cell–cell communication[20]. Interestingly, soluble ephrin-A ligands derived from their membrane-bound forms have been detected in the culture media of cell lines in models of glioblastoma, breast and other cancers, where they activate EphA receptors and downstream signaling, and cellular responses[21–24]. It is noteworthy that metalloprotease-mediated proteolytic shedding of membrane-anchored ephrins has been further demonstrated as one major mechanism to generate soluble ephrins in the extracellular space[25,26]. The potential function of Eph receptors in human cancers, including breast cancer, suggest that they can be targeted for cancer treatment[27–29]. Notably, ephrin A4 (EphA4) is upregulated in breast cancer stem-like cells (CSCs), and EphA4-mediated juxtacrine signaling maintains the stem cell state through their interactions with tumor-associated monocytes/macrophages and CSCs[30]. CSC-like cells have been shown to contribute to tumorigenicity, invasiveness, and immune evasion, and are ideal targets in cancer treatment[31–33]. In addition to the classical juxtacrine signaling, further investigations into the autocrine/paracrine signaling involved in EphA4 activation may provide a more comprehensive picture of Eph receptor regulation and shed light on potential therapeutic strategies against CSC-like cells by blocking EphA4 signaling.

RNases and RTKs, which were considered as two unrelated families, have been recently revealed the significance of a novel ligand–receptor relationship in pancreatic and liver cancers[14,34]. In this report, we uncovered a ligand–receptor pair of hRNase 1–EphA4 critical for breast cancer initiation, namely that hRNase 1 acts as a natively secretory ligand of EphA4 to stimulate EphA4 signaling in an autocrine/paracrine manner, leading to upregulation of stem-like cell properties in breast cancer.

## Results

**Higher hRNase 1 expression predicts poorer clinical outcome in several cancer types.** To systematically identify members of the hRNase A superfamily that contribute to cancer progression in addition to hRNase 5, we first examined the prognostic significance of hRNases in breast cancer using the median values for patient stratification analyzed by the Kaplan-Meier Plotter, a tool for meta-analysis-based biomarker assessment. Unexpectedly, only expression of hRNase 1 (Fig. 1a) exhibited a significant

negative correlation with breast cancer patient survival, but not that of the other seven hRNases, including hRNases 2, 3, 4, 5, 6, 7, and 11 (Supplementary Fig. 1a, b). Of note, the survival analysis was based on the database of Kaplan-Meier Plotter, in which the microarray contains 22,277 genes detected for breast cancer prognosis, but RNases 8, 9, and 10 are not included in the analysis[35]. Moreover, higher hRNase 1 expression significantly correlated with poorer survival in tumors with grade 2 in a near-significant trend (Fig. 1b), and hRNase 1 appeared to be a more significant poor prognostic factor in breast cancer patients with lymph node (LN)-negative non-metastasis, compared with those with LN-positive metastasis (Fig. 1c). Among the four molecular subtypes of breast cancer[36], hRNase 1 correlated negatively with the survival of breast cancer patients with the basal-like, luminal A and luminal B subtypes, in a trend toward statistical significance (Supplementary Fig. 2a). Interestingly, the luminal B subtype in each grade displayed a higher hazard ratio and a smaller $p$ value, suggesting a stronger correlation between high hRNase 1 expression and poor patient survival in the luminal B subtype, although there is a lack of statistical significance (Supplementary Fig. 2b). In addition, hRNase 1 expression predicted poorer outcome in LN-positive patients with the luminal A subtype and LN-negative patients with the luminal B subtype (Supplementary Fig. 2c). The notion that higher hRNase 1 expression associated with poorer patient survival was also demonstrated in two other independent databases of patients with breast cancer[37,38] (Supplementary Fig. 3a, b).

To further investigate the role of hRNase 1 in breast cancer, we examined tumor histology using a breast tumor tissue microarray (TMA) and found that the average expression level of hRNase 1 in breast tumors (T) was significantly higher than that in normal (N) breast tissues (Fig. 1d). Of note, the levels of tissue hRNase 1 had no statistically significant difference among the four subtypes of breast cancer (Fig. 1e). Consistently, serum hRNase 1 level was significantly higher in breast cancer patient group (T) compared with that in normal group (N) (Fig. 1f), which was also observed in three other independent cohorts (Supplementary Fig. 4a–c), and shown no significant difference among breast cancer subtypes (Fig. 1g and Supplementary Fig. 4d, e). Likewise, the average expression level of the conditioned medium (CM) hRNase 1 in eight breast cancer cell lines was significantly higher than that in two normal breast cells (Fig. 1h). Further analysis indicated that these eight breast cancer cell lines were determined at a concentration of CM hRNase 1 lower than 15 ng/ml, except KPL4 cells carrying CM hRNase 1 up to 100 ng/ml (Supplementary Fig. 4f). We further performed western blotting (WB) to screen a panel of 11 breast cancer cell lines, including basal-like (BT-549, MDA-MB-231, MDA-MB-468, and BT-20), luminal A (MCF7, T-47D, and ZR-75-1), luminal B (BT-474), and HER2 + (KPL4, SKBR3, and MDA-MB-453) to measure the levels of CM hRNase 1 (Fig. 1i). The results showed that five cell lines across subtypes (MDA-MB-231, BT-20, ZR-75-1, BT-474, and KPL4) exhibited apparent expression of hRNase 1, similar to the results from the tissue and serum hRNase 1 protein expression with no significant restriction on specific molecular subtype(s) of breast cancer (Fig. 1e, g). All these results indicated that the level of hRNase 1 is elevated in tissue and serum samples from breast cancer patients than those from normal individuals.

**The ribonucleolytic activity-independent function of hRNase 1 enriches the population of breast CSC-like cells and enhances the tumor-initiating capability.** Next, we asked whether the elevated hRNase 1 is involved in breast cancer tumorigenesis and whether this requires its ribonucleolytic activity. To this end, we first expressed hRNase 1 in MCF7 luminal A breast cancer cells

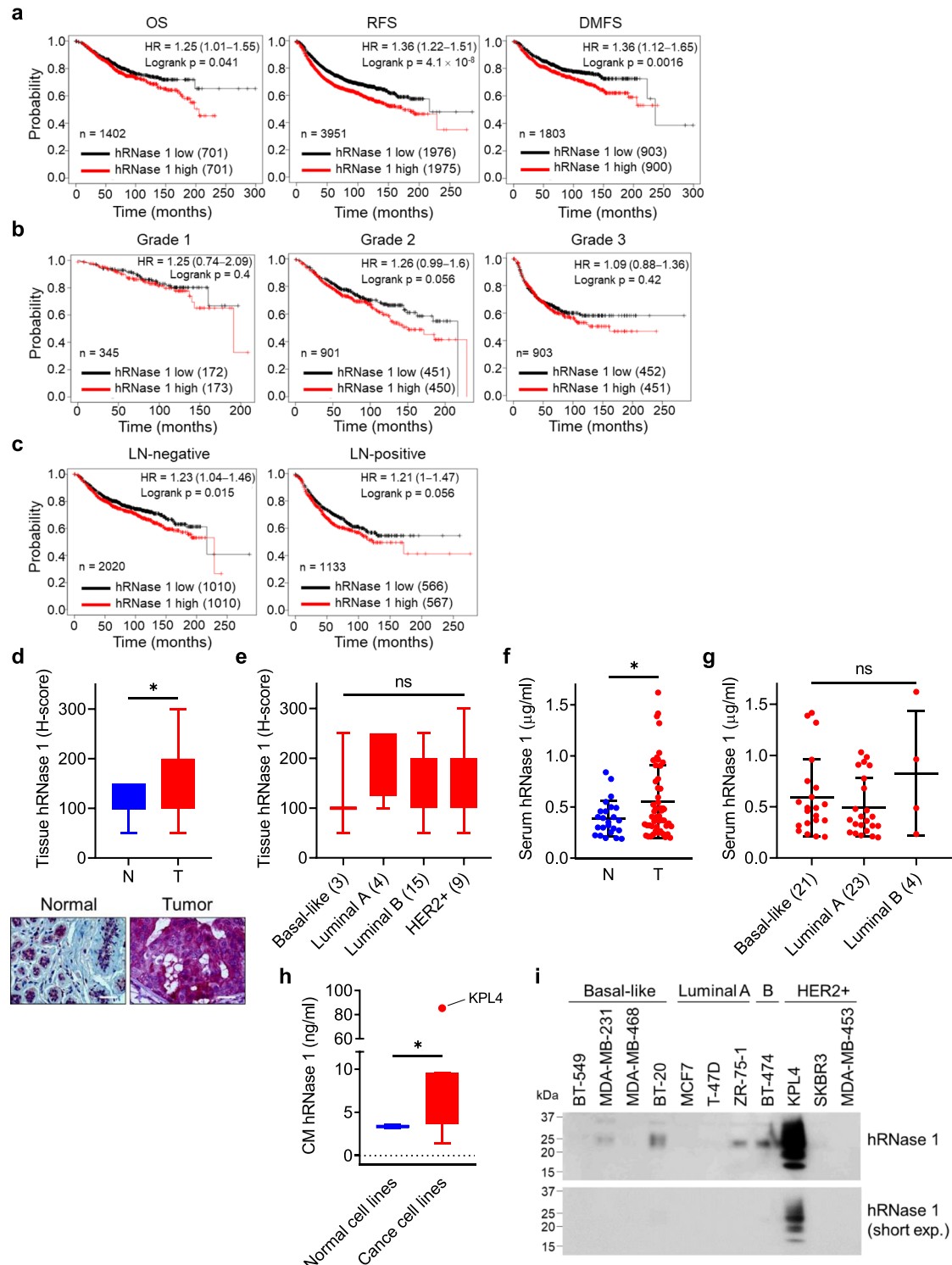

(Supplementary Fig. 5a) and found that hRNase 1 significantly increased the number of sphere formation in primary, secondary, and tertiary passages (Fig. 2a, b). Moreover, hRNase 1 enriched the population of CD24−/CD44+ cells, a molecular signature of breast CSC-like cells[39], in MCF7 parental cells and formed spheres (Fig. 2c, d). These results demonstrated that the elevated hRNase 1 promotes stem-like properties in breast cancer. Considering histidine 12 (H12) is critical for ribonucleolytic activity to cleave RNA by hRNase 1[40], we generated a catalytically inactive H12A variant of hRNase 1 with the amino-acid histidine to alanine substitution (R1-H12A; Supplementary Fig. 5b), which

does not disturb the overall three-dimensional structure of hRNase 1 protein[41]. Indeed, the H12 variant exhibited virtually no enzymatic activity compared with its wild-type (WT) counterpart (R1; Supplementary Fig. 5c), and both were expressed in BT-549 basal-like breast cancer cells (Supplementary Fig. 5d). It is worthwhile to mention that the CM containing abundant hRNase 1 expression, either WT or H12A, supposedly contributes to the major population for the detection of ribonucleolytic activity, although there would also exist a certain level of other endogenous RNase activities in the CM from HEK cells. In addition, hRNase 1 has three N-linked glycosylation sites reported at Asn-

**Fig. 1 High hRNase 1 expression predicts poor clinical outcome in breast cancer patients. a**, Prognostic correlation of survival analyses of breast cancer patients with high and low hRNase 1 levels. OS, overall survival; RFS, relapse-free survival; DMFS, distant-metastasis-free survival. HR, hazard ratio. **b, c** Prognostic correlation of the RFS with different grades (**b**) and lymph node statuses (**c**) of patients with breast cancer in the indicated groups with different hRNase 1 levels. LN, lymph node. **d** Top, plots of IHC scores of hRNase 1 level in breast tumor tissues (T; n = 31) vs normal tissues (N; n = 12) (Pantomics Inc., #BRC961). Bottom, representative images of IHC staining of hRNase 1 expression in normal and breast tumor tissues. Bar, 25 μm. *p = 0.0188. **e** Box plots of IHC scores of hRNase 1 level in breast tumor tissues subdivided into four molecular subtypes. Case numbers are shown in the parentheses. **f** ELISA of hRNase 1 levels in serum samples of breast tumor patients (T; n = 50) compared with noncancerous individuals (N; n = 24). *p = 0.0382. **g**, ELISA of hRNase 1 levels in breast tumor serum samples subdivided into three molecular subtypes. Case numbers are shown in the parentheses. **h** Box plots of ELISA of hRNase 1 expression in the conditioned medium (CM) of eight breast cancer cells and two normal cells. KPL4 cell line as an outlier. **i** Western blotting (WB) of hRNase 1 expression level secreted in breast cancer cells; the experiment was repeated a second time with similar results. *p = 0.0415. Survival data were analyzed using Kaplan-Meier Plotter (Probe: 201785_at) and p values were calculated by Log-rank test (**a–c**). Error bars represent mean ± SD (**f, g**). *p < 0.05, ns, not significant, two-tailed unpaired t test (**d, f, h**), ANOVA analysis (**e, g**). Box plots indicate minima (lower end of whisker), maxima (upper end of whisker), median (center), 25th percentile (bottom of box), and 75th percentile (top of box) (**d, e, h**) as well as outlier (single point) (**h**). Source data are provided as a Source Data file.

34, Asn-76, and Asn-88, and the heterogeneous pattern of hRNase 1 on gel electrophoresis reflected different levels of glycosylation depending on cell and tissue types by a range of bands from 15 to 36 kDa, in which the relatively lowest molecular weight band at ~15–20 kDa corresponds to a non-glycosylated hRNase 1[6,7]. Indeed, treatment with N-glycosidase (peptide-N-glycosidase F; PNGase F) in Flag-tagged BT-549-R1 cells resulted in a homogenous pattern of hRNase 1 at the relatively lowest molecular weight band below 20 kDa corresponding to a non-glycosylated hRNase 1 (Supplementary Fig. 5e). We observed that overexpression of either R1 or R1-H12A in BT-549 cells increased the colony-forming capacity in soft agar (Supplementary Fig. 5f), suggesting a potential role of hRNase 1 that promotes in vitro oncogenic transformation independently of its ribonucleolytic activity. Moreover, R1- or R1-H12A-expressing BT-549 cells demonstrated increased stem-like properties, including the sphere-forming ability (Fig. 2e) and the population of CD24−/CD44+ cells (Fig. 2f). Notably, results from an in vivo limiting dilution assay showed that either R1- or R1-H12A-expressing BT-549 cells increased tumor-initiating cell (TIC) frequency, calculated using ELDA web-tool[42] through inoculation of three diluted dosages of BT-549 cells (Fig. 2g and Supplementary Fig. 5g). The p values of pairwise comparisons for differences in TIC frequencies between BT-549-Ctrl versus BT-549-R1 and BT-549-Ctrl versus BT-549-R1-H12A were significantly displayed as 0.031 and 0.008, respectively, whereas there was no significant difference in TIC frequencies between BT-549-R1 and BT-549-R1-H12A (p = 0.641), suggesting that hRNase 1 overexpression significantly increased TIC frequency independently of its ribonucleolytic activity. We further established stable clones expressing R1 and R1-H12A in T-47D luminal A breast cancer cells (Supplementary Fig. 5h–j) and similar results were observed (Supplementary Fig. 5k–m). Together, these results indicated that the stemness-promoting feature of hRNase 1 occurs in both basal-like and luminal subtypes of breast cancer cells and does not require its ribonucleolytic activity.

**Silencing hRNase 1 reduces the population of breast CSC-like cells and decreases the tumor-initiating capability.** In contrast, knocking down hRNase 1 by small hairpin RNA (shRNA; sh-R1#1 and sh-R1#2) in KPL4 HER2+ breast cancer cells which express high levels of hRNase 1 (Supplementary Fig. 6a, b) significantly reduced the sphere-forming ability, the population of CD24−/CD44+ cells, and the TIC frequency (Fig. 2h–j and Supplementary Fig. 6c). The results were further validated by the reconstitution of hRNase 1 in KPL4 stable clones knocking down hRNase 1 (KPL4-sh-R1#2), in which hRNase 1 was successfully reconstituted into KPL4-sh-R1#2 cells (+ R1 vs + vector;

Supplementary Fig. 6d), showing restored sphere-forming ability (KPL4-sh-R1#2 + R1 vs KPL4-sh-R1#2 + vector; Fig. 2k). We further studied the effects of hRNase 1 on the induction of cancer stemness by examining the activity of aldehyde dehydrogenase 1 (ALDH1), a putative marker for breast CSC-like cells[43]. Similar to the above results, hRNase 1 knockout KPL4 (KO-R1) cells generated by CRISPR-Cas9 exhibited reduced ALDH1 activity (Fig. 2l, m), further supporting a positive role of hRNase 1 in the stem-like properties and tumor initiation in breast cancer.

**hRNase 1 activates EphA4 signaling and associates with EphA4.** Considering the above results that an abundance of hRNase 1 was detected in the serum of breast cancer patients and that hRNase 1 played a positive role in breast tumor initiation, we explored the mechanistic aspects of secretory hRNase 1 and its biological impacts on tumor cells. To this end, we performed an unbiased antibody array of human phospho-RTKs in HeLa epithelial cancer cells as a general model for pilot experiments before pursuing this topic in breast cancer research (Fig. 3a) and BT-549 breast cancer cells (Supplementary Fig. 7a). The results showed that treatment with recombinant hRNase 1 (purified from HEK293 cells) increased tyrosine phosphorylation of EphA4, hepatocyte growth factor receptor (HGFR), and TEK receptor tyrosine kinase (Tie-2) in HeLa cells, and that of EphA4, EphA10, and RTK like orphan receptor 2 (ROR2) in BT-549 cells. Among them, EphA4 was the only receptor whose phosphorylation increased in both cell lines and previously reported to maintain breast CSC-like cells[30]. Hence, we focused on whether hRNase 1 affects the EphA4 pathway. First, we demonstrated that hRNase 1 increased EphA4 phosphorylation in BT-549 cells in a time- (Fig. 3b) and dose-dependent manner with a half-maximal effective concentration (EC$_{50}$) of 9.85 nM (Fig. 3c), which was within the nanomolar range comparable to that of ephrin-A1/EphA2[44]. hRNase 1 induced phosphorylation of EphA4, but not of EphA3 or EphA5 (Supplementary Fig. 7b), and activation of phospholipase C-γ (PLCγ), a downstream signaling molecule of EphA4[30], in BT-549 basal-like breast cancer cells (Supplementary Fig. 7c). Similar results were observed in luminal A breast cancer cells including MCF7 and T-47D (Fig. 3d and Supplementary Fig. 7d), suggesting the hRNase 1–EphA4 axis might be a general ligand–receptor relationship across breast cancer subtypes. We also observed similar results in HeLa cells (Supplementary Fig. 7e, f). To further address this notion, we demonstrated the association between hRNase 1 and EphA4 in vivo and in vitro by co-immunoprecipitation (co-IP) assay (endogenous pull-down; Fig. 3e), Duolink in situ proximity ligation assay (PLA) combined with phase contrast imaging (Fig. 3f and Supplementary Fig. 7g–i), GST pull-down (GST-tagged hRNase 1 and Myc-tagged EphA4; Fig. 3g), and pull-down of recombinant hRNase 1

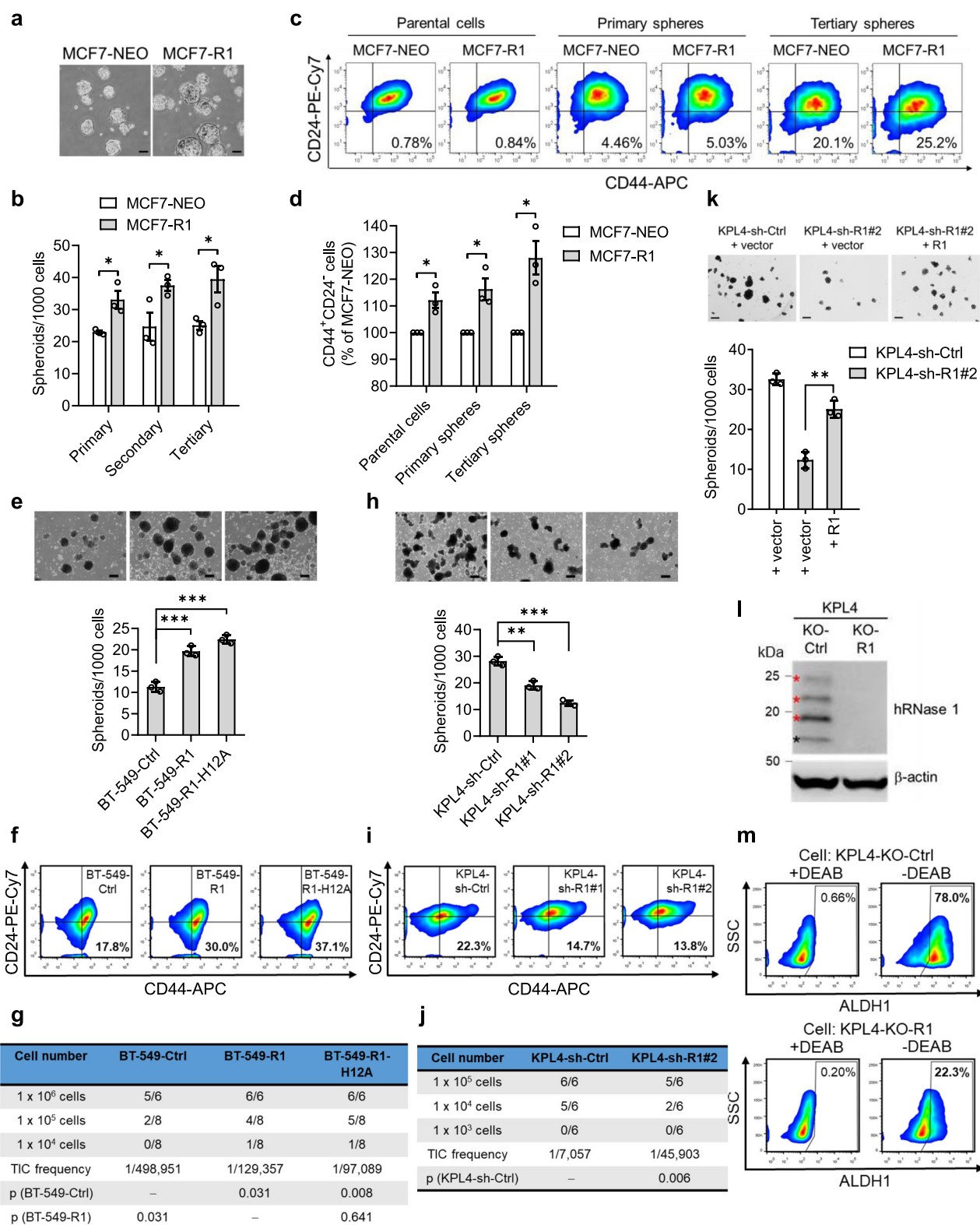

protein purified from *Escherichia coli* (Fig. 3h) and HEK293 cells (Fig. 3i, j) by purified N-terminal EphA4 chimera (N-EphA4-Fc). Notably, numerous PLA signals were solely detected by anti-EphA4 and anti-Flag antibodies in the recipient BT-549 and KPL4 cells treated with the CM containing Flag-tagged hRNase 1 (293T-pCDH-R1), but not the CM containing Flag-tagged hRNase 5 (293T-pCDH-hRNase 5) (Fig. 3f). These results suggested that EphA4 is in close proximity to hRNase 1 but not

hRNase 5 as a comparison, in line with the previous results from phospho-RTK antibody array indicating that hRNase 5 as an EGFR ligand induces phosphorylation of EGFR but not other RTKs[14]. Considering hRNase 1 as a *N*-linked glycosylated protein[5–7], we performed in vitro binding assay by incubating N-EphA4-Fc with recombinant hRNase 1 protein produced from HEK293 cells, which we pretreated with glycosidase PNGase F (Fig. 3i), and found that non-glycosylated hRNase 1 still harbored

**Fig. 2 hRNase 1 increases the tumor-initiating ability of breast cancer cells independently of its ribonucleolytic activity. a** Representative images of primary spheres in MCF7 control cells (MCF7-NEO) and hRNase 1 expressing MCF7 cells (MCF7-R1). Bar, 100 μm. **b** Quantification of spheroid-formation assay of primary, secondary, and tertiary spheres derived from the indicated MCF7 stable clones. $n = 3$ independent experiments. $^*p = 0.0218$ for primary spheres, $^*p = 0.0489$ for secondary spheres, $^*p = 0.0289$ for tertiary spheres. **c** Flow cytometric analysis of membrane CD44 and CD24 expression in the indicated MCF7 stable clones (parental cells) and primary and tertiary spheres derived from the indicated MCF7 stable clones. **d** Quantification of flow cytometric analysis from (**c**). $n = 3$ independent experiments. $^*p = 0.0148$ for parental cells, $^*p = 0.0164$ for primary spheres, $^*p = 0.0111$ for tertiary spheres. **e** Spheroid-formation assay of the indicated BT-549 stable clones. Bar, 200 μm. Ctrl vs R1, $^{***}p = 0.00099$, Ctrl vs R1-H12A, $^{***}p = 0.0002$. **f** Flow cytometric analysis of membrane CD44 and CD24 expression in the indicated BT-549 stable clones. **g** Limiting dilution assay of the stable transfectants derived from BT-549 as indicated. **h** Spheroid-formation assay of the indicated KPL4 stable clones. Bar, 200 μm. sh-Ctrl vs sh-R1#1, $^{**}p = 0.0023$, sh-Ctrl vs sh-R1#2, $^{***}p = 0.0002$. **i** Flow cytometric analysis of membrane CD44 and CD24 expression in the indicated KPL4 stable clones. **j** Limiting dilution assay of the stable transfectants derived from KPL4 as indicated. **k** Representative images (top) and quantification (bottom) of spheroid-formation assay of the indicated stable clones in KPL4. Bar, 200 μm. $n = 3$ independent experiments. $^{**}p = 0.0020$. **l** WB of KPL4 stable transfectants knocking out hRNase 1 and empty control with hRNase 1 (Sigma-Aldrich, #HPA001140) and β-actin (Sigma-Aldrich, #A2228) antibodies. Red asterisk, glycosylated hRNase 1; black asterisk, non-glycosylated hRNase 1. The experiment was repeated a second time with similar results. **m** Flow cytometric analysis of ALDH1 activity by ALDEFLUOR assay in cells from (**l**). The flow cytometry plots showing side scatter (SSC) vs fluorescence intensity. The percentages of ALDEFLUOR-positive cells are shown in the right upper quadrant of each panel. Cells treated with ALDH1 inhibitor diethylaminobenzaldehyde (DEAB) were used as negative control for this assay. In (**e, h**) representative images (top) and quantification (bottom) of spheroid-formation assay of the indicated stable clones. Data represent three independent experiments in three technical replicates. In (**c, f, i**) the percentages of CD44$^+$CD24$^-$ cells are shown in the right lower quadrant of each panel. In (**g, j**) the number of tumor-forming mice within each group is shown in the panel. The $p$ values of pairwise comparisons for differences in TIC frequencies using the ELDA web-tool[42], Chi-squared test. Error bars represent mean ± SEM (**b, d**) and mean ± SD (**e, h, k**). $^*p < 0.05$, $^{**}p < 0.01$, $^{***}p < 0.001$, two-tailed unpaired $t$ test (**b, d, e, h, k**). Source data are provided as a Source Data file.

the binding ability to N-EphA4-Fc (Fig. 3k). These results suggested that the $N$-linked glycosylation is not required for binding to N-EphA4-Fc; however, we do not rule out a possibility that heterogeneous glycosylation of hRNase 1 may contribute to its binding to EphA4 at different levels under various cellular, physiological, or pathological conditions. Collectively, hRNase 1 binds directly to EphA4 and functions as a ligand to induce downstream signaling of EphA4.

**Binding of hRNase 1 to EphA4 ligand-binding domain requires its C-terminus.** Next, we asked which region of EphA4 interacts with hRNase 1. We generated a GST-hRNase 1 and a series of constructs of Myc-tagged EphA4 (Fig. 4a). We found that hRNase 1 interacted with the extracellular domain (ECD) but not intracellular domain (ICD) of EphA4 (Fig. 4b), and deletion of the EphA4 ligand-binding domain (LBD) within its ECD abrogated its association with hRNase 1 (Supplementary Fig. 8). Moreover, hRNase 1-stimulated EphA4 phosphorylation in cells expressing EphA4-WT but not EphA4-ΔLBD (Fig. 4c) or the kinase-dead mutant (EphA4-K653A) (Fig. 4d), suggesting that hRNase 1 binds to EphA4 ECD through the LBD and stimulates phosphorylation of EphA4. To map the region of hRNase 1 required for EphA4 binding, we generated GST-hRNase 1 deletion mutants of N-terminus (ΔN; amino acids 1–21 deletion) and C-terminus (ΔC; amino acids 113–128 deletion), which lost the domains containing the catalytically active residues, H12 and H119, respectively (Fig. 4e, f). The results from Duolink in situ PLA demonstrated that the hRNase 1–EphA4 interaction significantly decreased in the absence of hRNase 1 C-terminus (ΔC) but not WT or the N-terminal deletion mutant (ΔN) as indicated by the reduced PLA signals, suggesting that the C-terminal domain of hRNase 1 is required for its binding to EphA4 (Fig. 4g). These results suggested that hRNase 1 binds to EphA4 via its C-terminus and that their interaction requires the extracellular LBD of EphA4.

**hRNase 1 induces EphA4 dimerization/oligomerization and its binding to EphA4 partially overlaps with that of ephrin-A5.** The above results prompted us to further characterize the ligand–receptor relationship between hRNase 1 and EphA4 and compare that with classical membrane-associated ephrin ligands. First, we demonstrated that hRNase 1 induced EphA4

dimerization/oligomerization (Fig. 5a), similar to the classical ephrin ligand-dependent Eph receptor activation, followed by the higher-order clustering for Eph's and ephrin's interactions assembled into dimerization, tetramerization, and oligomerization[45–47]. It is worthwhile to mention that the recruitment of A- and B-type Eph receptors into signaling clusters can occur independent of ephrin contacts via direct Eph–Eph receptor interactions[48,49]. In addition, the estimated dissociation constant (Kd) of hRNase 1 binding to EphA4 (92.4 nM equivalent to 1.7 μg/ml) was within the nanomolar range for the cognate ephrin and Eph ligand–receptor complex formation[19] (Fig. 5b). The Kd of ephrin-A5 for EphA4 (5.6 nM equivalent to 0.15 μg/ml) was similar to a previously reported Kd value of less than 10 nM[50] (Fig. 5c). Results from a competitive binding assay revealed a partial concentration-dependent inhibition of hRNase 1–EphA4 binding with increasing amounts of ephrin-A5 (Fig. 5d). Those results were further validated by treating cells with the KYL peptide acting as an EphA4 kinase inhibitor, which is a linear 12 amino-acid peptide (KYLPYWPVLSSL) that selectively binds to EphA4 and inhibits EphA4–ephrin-A5 interaction[51,52]. The interaction between hRNase 1 and EphA4 was significantly but not completely reduced in the presence of KYL compared with the KYL-P7A mutant control, known to completely lose its EphA4 binding activity[51] (Fig. 5e). These results indicated a partial overlap between the hRNase 1 and ephrin-A5 binding sites on the EphA4 receptor and supported the proposed role of hRNase 1 as an EphA4 ligand. We further asked whether hRNase 1-stimulated EphA4 activation might not be directly mediated by hRNase 1, but an indirect result of elevated levels of classical ephrin ligands. To this end, we screened the entire ephrin ligand family, including ephrin A1–A5 and B1–B3. Among them, six ephrin ligands, ephrin-A1, -A2, -A4, -A5, -B1, and -B2, did not increase significantly in BT-549 clones overexpressing hRNase 1 compared with vector control cells (Fig. 5f, g), and the levels of ephrin-A3 and -B3 were undetectable in BT-549 cells (Fig. 5h). These results suggested that hRNase 1-mediated EphA4 activation is specific and direct, instead of coming from an indirect effect by augmenting expression levels of classical ephrin ligands.

**hRNase 1 induces spheroid formation via IKK/NF-κB and MEK/Erk activating pathways.** Studies have shown that EphA4

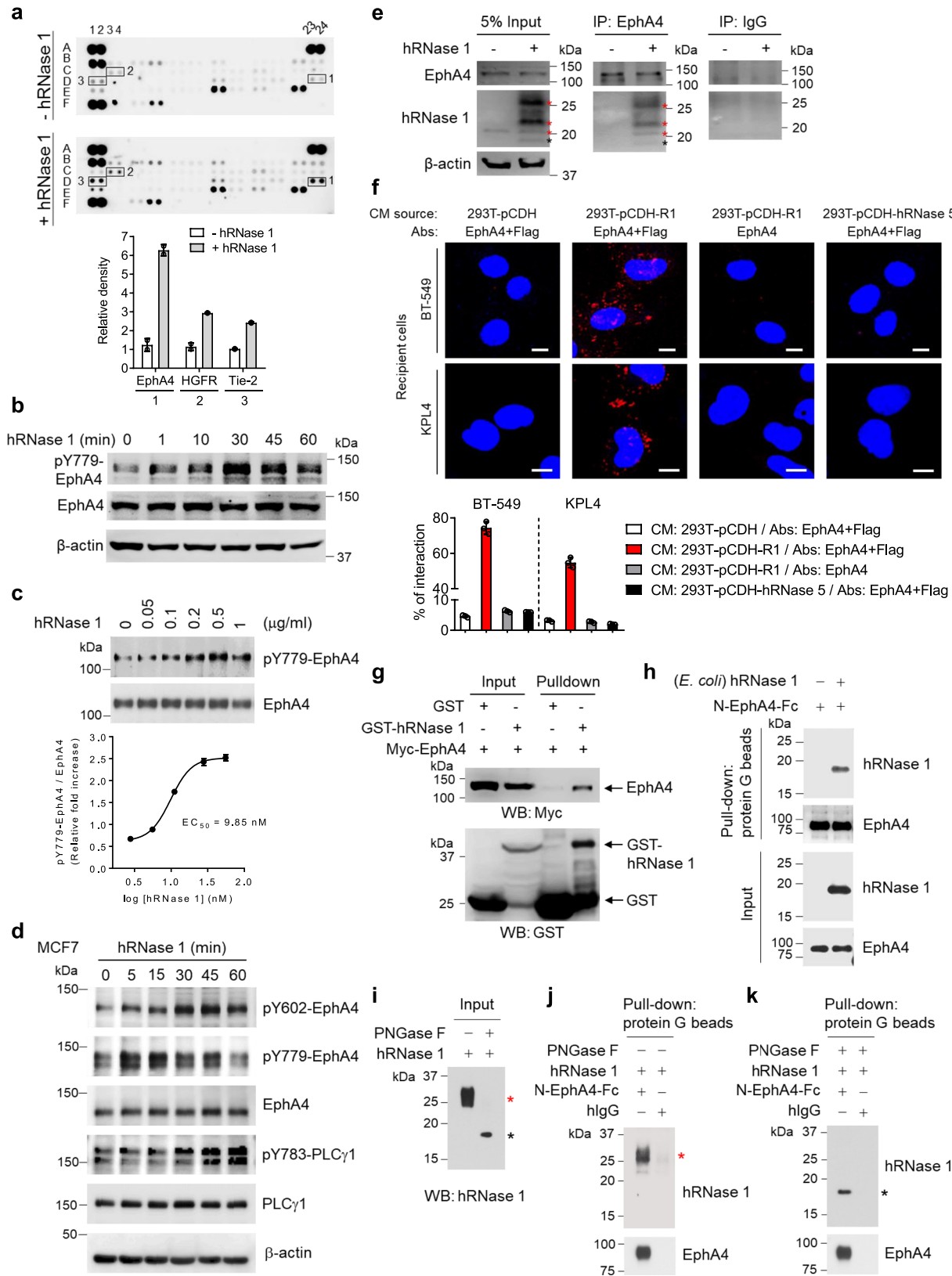

downstream signaling molecules, such as NF-κB, Erk, Src, and Akt, are activated in different cell types in response to the classical ephrin–EphA juxtacrine signaling[28,30]. In line with those observations, we found that NF-κB activation, as indicated by the nuclear accumulation of the NF-κB subunits (p50 and p65), was much higher in BT-549 cells expressing hRNase 1 compared with

control cells (Supplementary Fig. 9a). Only phosphorylation of Erk, but not Src or Akt, was detected in BT-549 cells expressing hRNase 1 (Supplementary Fig. 9b) or treated with hRNase 1 (Supplementary Fig. 9c). In addition to BT-549 cells, we also observed nuclear accumulation of p65 NF-κB and phosphorylation of Erk, but not Src or Akt, time-dependently in response to

**Fig. 3 hRNase 1 activates EphA4 signaling and associates with EphA4. a** Top, human phospho-RTK antibody array analysis of HeLa cells treated with or without recombinant hRNase 1 protein purified from HEK293 cells (1 μg/ml) for 5 min after serum starvation for 3 h. Three pairs of positive signals in duplicate coordinates (minus hRNase 1 comparing to plus hRNase 1) are shown in EphA4 (D23/D24), HGFR (C3/C4), and Tie-2 (D1/D2). Bottom, quantification of detected signals by ImageJ. **b** WB of BT-549 cells treated with recombinant hRNase 1 protein purified from HEK293 cells (1 μg/ml) at various time points, blotted with phospho-EphA4-Y779 (pY779-EphA4), EphA4, and β-actin antibodies. **c** Top, WB of BT-549 cells treated with hRNase 1 at various concentrations for 30 min with pY779-EphA4 and EphA4 antibodies. Bottom, quantification expressed as fold increases of hRNase 1 induction in EphA4 phosphorylation by normalizing against the EphA4 total protein expression, as compared with untreated control. **d** WB of MCF7 cells treated with or without hRNase 1 at various time points with the indicated antibodies. **e** Immunoprecipitation (IP) of MCF7 cells treated with or without hRNase 1 for 30 min with antibodies targeting EphA4 or normal IgG followed by WB with the indicated antibodies. Left, input lysates. **f** Duolink in situ PLA of BT-549 and KPL4 cells, treated with CM collected from the indicated 293T stable transfectants for 30 min. Bar, 10 μm. Bar diagram, the percentage of cells showing positive interaction calculated from three independent fields of each pool. Negative control, EphA4 antibody only or CM source as 293T-pCDH-hRNase 5. **g** In vitro GST pull-down assay of GST-tagged hRNase 1/GST-binding magnetic beads incubated with lysate from 293T transfected with Myc-tagged EphA4. Left, input lysates. The 26 kDa GST band is a product of GST-hRNase 1 degradation. **h**, Top, in vitro binding assay of N-EphA4-Fc incubated with or without hRNase 1 protein purified from *E. coli*. Protein G beads were used for pull-down. Bottom, input lysates. **i–k**, Left, input lysates of recombinant hRNase 1 without or with PNGase F treatment (**i**). In vitro binding assay of recombinant hRNase 1 treated without (**j**) or with (**k**) PNGase F and incubated with N-EphA4-Fc or human IgG. In (**e, i–k**) red asterisk, glycosylated hRNase 1; black asterisk, non-glycosylated hRNase 1. Data are presented as mean ± SD of two (**a, c**) or three (**f**) independent experiments. Each experiment was repeated an additional time with similar results (**b, d, e, g–k**). Source data are provided as a Source Data file.

hRNase 1 treatment in MCF7 cells (Supplementary Fig. 9d, e). We then performed spheroid-formation assay in BT-549 stable clones treated with various inhibitors, including those against IKK/NF-κB (QNZ) and MEK/Erk (GSK1120212 and PD-0325901), at effective but relatively low concentrations (low nM to low μM) which did not significantly damage the sphere-forming ability of control clones (BT-549-Ctrl), in comparison to the results of hRNase 1-expressing stable cells (BT-549-R1). We found that hRNase 1-expressing BT-549 stable cells treated with those inhibitors exhibited attenuated sphere-forming ability compared with vector control (Supplementary Fig. 10a). Of note, although the sphere-forming ability in BT-549-Ctrl cells was not significantly attenuated by those inhibitors, the inhibitory effect was much more profound in BT-549-R1 stable clones, suggesting that BT-549-R1 cells may be more addicted to IKK/NF-κB and MEK/Erk activating pathways, leading to a higher sensitivity to those inhibitors. We further validated that phosphorylation of p65 NF-κB and Erk were indeed declined under inhibitor treatment in BT-549-Ctrl cells, supporting the effectiveness of inhibitors (Supplementary Fig. 10b). Similar inhibitory effects were also observed in hRNase 1-expressing MCH7 stable cells treated with inhibitors against IKK/NF-κB (Bay 11-7821) and MEK/Erk (PD-0325901), but not those against Akt (MK-2206) and Src (Dasatinib) (Supplementary Fig. 10c, d), consistent with the results showing that hRNase 1 activated NF-κB and Erk, but not Akt and Src, in MCF7 cells (Supplementary Fig. 9d, e). The effectiveness of those four inhibitors were validated in MCF7 control cells (Supplementary Fig. 10e–g). The above results demonstrated that hRNase 1 activates some well-recognized EphA4 downstream pathways, e.g., IKK/NF-κB and MEK/Erk, which positively regulate stem-like cell properties, but not others, e.g., Src and Akt pathways.

**EphA4 positively regulates hRNase 1-mediated breast tumor initiation and tumorigenesis.** To further gain insights into the mechanism underlying hRNase 1–EphA4 induced stem-like cell properties, we first ectopically expressed EphA4 in ZR-75-1 luminal A breast cancer cells (Supplementary Fig. 11a) and observed the upregulated sphere formation (Fig. 6a and Supplementary Fig. 11b) and the enriched population of CD24−/CD44+ stem-like cells (Fig. 6b, c), whereas shRNA knockdown of hRNase 1 abrogated those effects. Similar results were also demonstrated in BT-474 luminal B breast cancer cells (Supplementary Fig. 11c) from sphere-formation assay (Supplementary Fig. 11d, e) and flow cytometry analysis of CD24−/CD44+ expression

(Supplementary Fig. 11f, g). Furthermore, we silenced hRNase 1 by CRISPR-Cas9 knockout in EphA4-expressing KPL4 HER2+ cells (Supplementary Fig. 11h), and found that expression of EphA4 upregulated ALDH1 activity and enriched the population of stem-like KPL4 cells, whereas knockout of hRNase 1 abrogated those effects (Fig. 6d and Supplementary Fig. 11i, j). Knocking out hRNase 1 also neutralized EphA4-mediated sphere-forming frequency (Fig. 6e). In addition, we examined the role of hRNase 1–EphA4 axis on TIC frequency by an in vivo limiting dilution assay in a subcutaneous mouse model. The estimated TIC frequency of EphA4-expressing KPL4 was 3-fold higher than control cells (1/5,082 vs 1/16,026; KPL4-A4 vs KPL4-NEO; Fig. 6f), but the increase was attenuated when we knocked out hRNase 1 (1/12,472 vs 1/4,878; KPL4-A4-KO-R1 vs KPL4-A4-KO-Ctrl; Fig. 6f).

In contrast, silencing EphA4 by shRNA knockdown in hRNase 1-expressing MCF7 cells (Supplementary Fig. 12a) attenuated the hRNase 1-induced sphere-forming frequency (Fig. 6g and Supplementary Fig. 12b) and stem-like cell population (Fig. 6h, i). Furthermore, we asked whether EphA4 positively regulates hRNase 1-mediated in vitro oncogenic transformation potential as well as the stem-like properties. Indeed, silencing EphA4 by shRNA knockdown or CRISPR-Cas9 knockout (Supplementary Fig. 12c, d) in hRNase 1-expressing BT-549 cells attenuated colony-forming capacity in soft agar (shRNA only; Supplementary Fig. 12e) and sphere-forming frequency (Fig. 6j and Supplementary Fig. 12f). hRNase 1-expressing BT-549 cells initiated tumors more frequently by 6.5-fold compared with control counterparts (1/52,780 vs 1/351,989; BT-549-R1 vs BT-549-NEO; Fig. 6k), whereas knocking out EphA4 in hRNase 1-expressing BT-549 cells decreased hRNase 1-mediated TIC frequency (1/412,616 vs 1/69,591; BT-549-R1-KO-A4 vs BT-549-R1-KO-Ctrl; Fig. 6k). Together, these results highlighted the biological significance of hRNase 1–EphA4 axis in promoting in vitro oncogenic transformation and breast tumor initiation.

To further validate the contribution of the RNase 1–EphA4 axis to breast tumorigenicity, we established mouse 4T1 mammary tumor cells, which harbored low endogenous mouse RNase 1 (mRNase 1), stably expressing mRNase 1 and vector control (Supplementary Fig. 13a, b). Nude mice orthotopically injected with 4T1-mRNase 1 developed tumors that weighed more than those from mice injected with vector control cells ($p = 0.009$; Supplementary Fig. 13c). The mRNase 1-mediated tumorigenesis was suppressed when mice were treated with compound 1 (cpd1), a small molecule that binds to the EphA4-

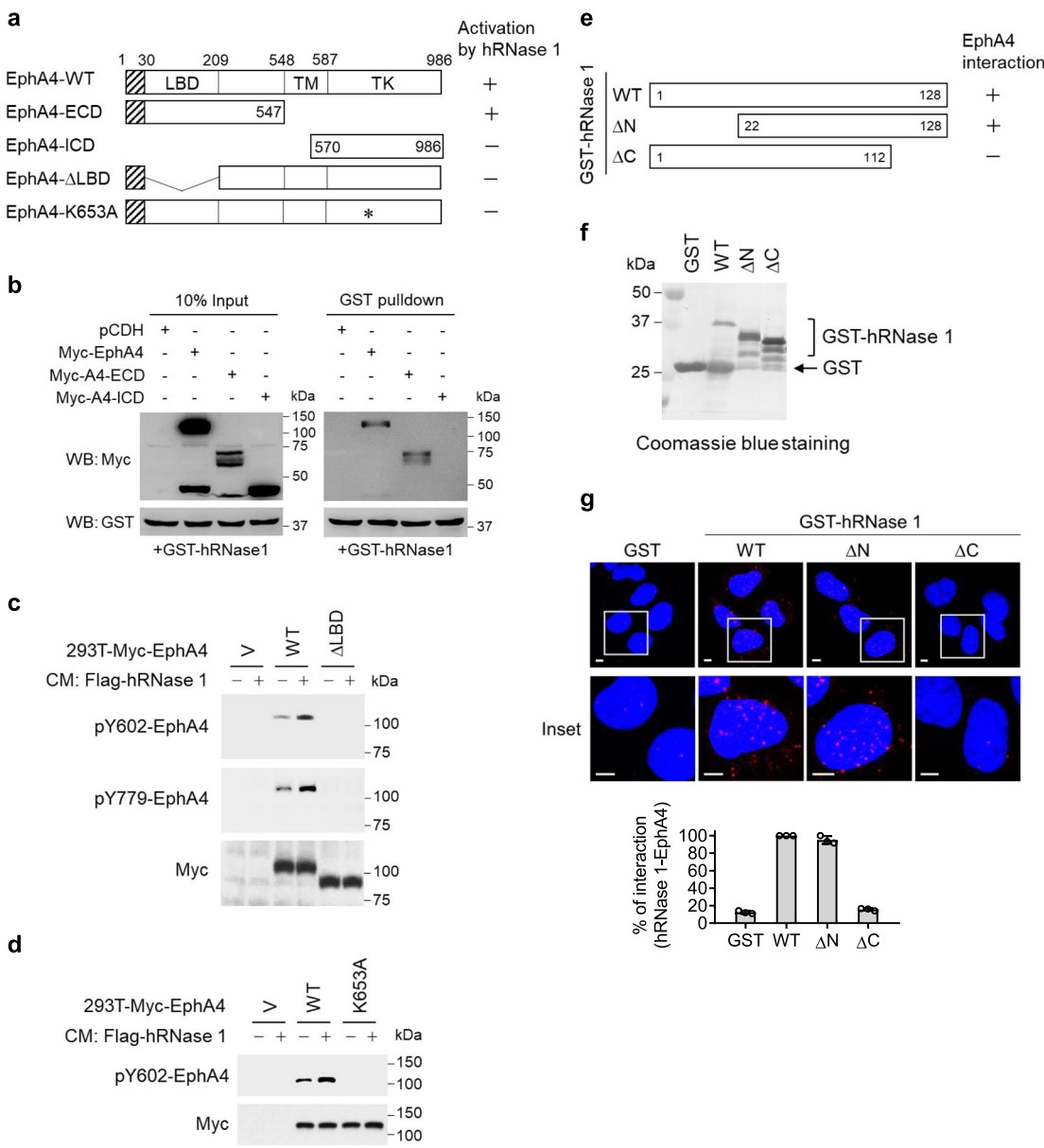

**Fig. 4 hRNase 1 binds to EphA4 ligand-binding domain and binding requires its C-terminus. a** Schematic diagram of various constructs of Myc-tagged EphA4. The numbers represent amino-acid residues. EphA4-ECD, amino acids 1–547 of EphA4; EphA4-ICD, amino acids 570–986 of EphA4; *, EphA4-K653A, mutation of EphA4 phosphorylation site. **b** In vitro GST pull-down assay of GST-tagged hRNase 1/glutathione magnetic beads incubated with lysate from 293T transfected with the indicated expression plasmids, including WT, ECD and ICD of EphA4, and pCDH empty vector, followed by three times of PBS washing. Left, input lysates. **c**, **d** WB of 293T stable transfectants generated by constructs of (**a**) or empty vector (V) treated with or without CM collected from 293T stable transfectant containing hRNase 1 for 30 min. **e** Schematic diagram of various constructs of GST-tagged hRNase 1. **f** Coomassie blue staining of GST-hRNase 1 plasmids as indicated. The 26 kDa GST band is a product of GST-hRNase 1 degradation. **g** Duolink in situ PLA of KPL4 cells expressing EphA4 and hRNase 1 knockout (KPL4-A4-KO-R1) treated with GST vector or GST-hRNase 1 proteins from (**f**) for 30 min. Insets, 6.25× magnification. Bar, 5 µm. Bar diagram indicates the percentage of cells showing positive hRNase 1 and EphA4 interaction calculated from a pool of 50 cells; error bars represent mean ± SD, n = 3 independent experiments. Data in (**b**–**d**, **f**) are representative of two independent experiments with similar results. Source data are provided as a Source Data file.

LBD[53] ($p = 0.026$; Supplementary Fig. 13c). Collectively, these findings suggested that the RNase 1–EphA4 axis contributes to breast tumor progression and stemness by positively regulating tumor-initiating ability and enriching the CSC-like cell population.

**Pathological relevance among hRNase 1 expression, EphA4 activation, and CD133 in breast cancer.** To determine the pathological relevance of secretory hRNase 1 and EphA4 activation, we evaluated the expression levels of secretory hRNase 1 in

plasma samples and phospho-EphA4 levels in matched tumor tissues from patients with breast cancer; notably, a significantly positive correlation was observed ($p = 0.042$; Fig. 7a). Moreover, TMA analysis of two independent human breast cancer cohorts also revealed a positive association between hRNase 1 and phospho-EphA4 expression (Fig. 7b and Supplementary Fig. 14a, b). The stem cell marker CD133 was positively associated with hRNase 1 expression (Fig. 7c) and EphA4 activation (Fig. 7d), supporting the pathological features of the hRNase 1–EphA4 axis in breast tumor initiation (representative cases are shown in

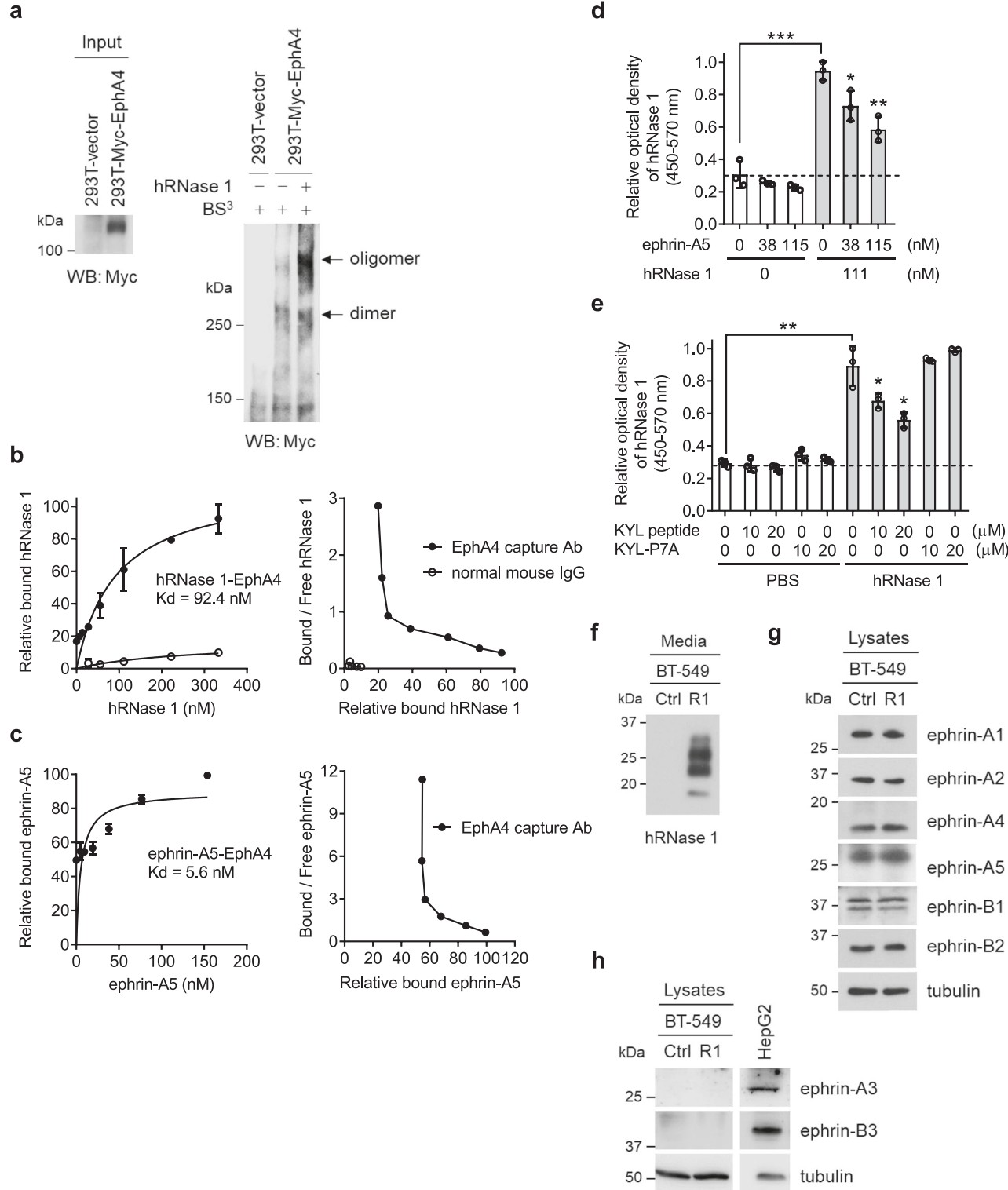

**Fig. 5 hRNase 1 induces EphA4 dimerization/oligomerization and its binding to EphA4 partially overlaps with that of ephrin A5. a** Dimerization assay of 293T cells ectopically expressing Myc-tagged EphA4 or vector control, cross-linked with bis(sulfosuccinimidyl)suberate ($BS^3$) followed by WB under a non-reducing and non-denaturing condition. Left, input lysates. **b, c** Left, saturation binding assay of the Kd values determination for hRNase 1 (**b**) or ephrin-A5 (**c**) binding toward EphA4 in BT-549 cell lysates. Right, scatchard plot. Negative control, normal mouse IgG. Each experiment was performed twice in triplicate. **d, e** Binding assay measuring hRNase 1 binding affinity toward EphA4 in BT-549 cells with increasing concentrations of ephrin-A5 (**d**) and KYL or KYL-P7A peptide (**e**) as indicated. The optical density was determined at 450 nm, corrected by subtraction of reading at 570 nm. *$p < 0.05$, **$p < 0.01$, ***$p < 0.001$, two-tailed unpaired $t$ test. **f–h** WB of secreted proteins from CM (**f**) and lysates (**g, h**) in BT-549-Ctrl and BT-549-R1 stable clones. Positive control, HepG2 cell lysates. Data are representative of two (**a**) or three (**f–h**) independent experiments with similar results. Error bars represent mean ± SD, $n = 3$ independent experiments (**b–e**). Source data are provided as a Source Data file.

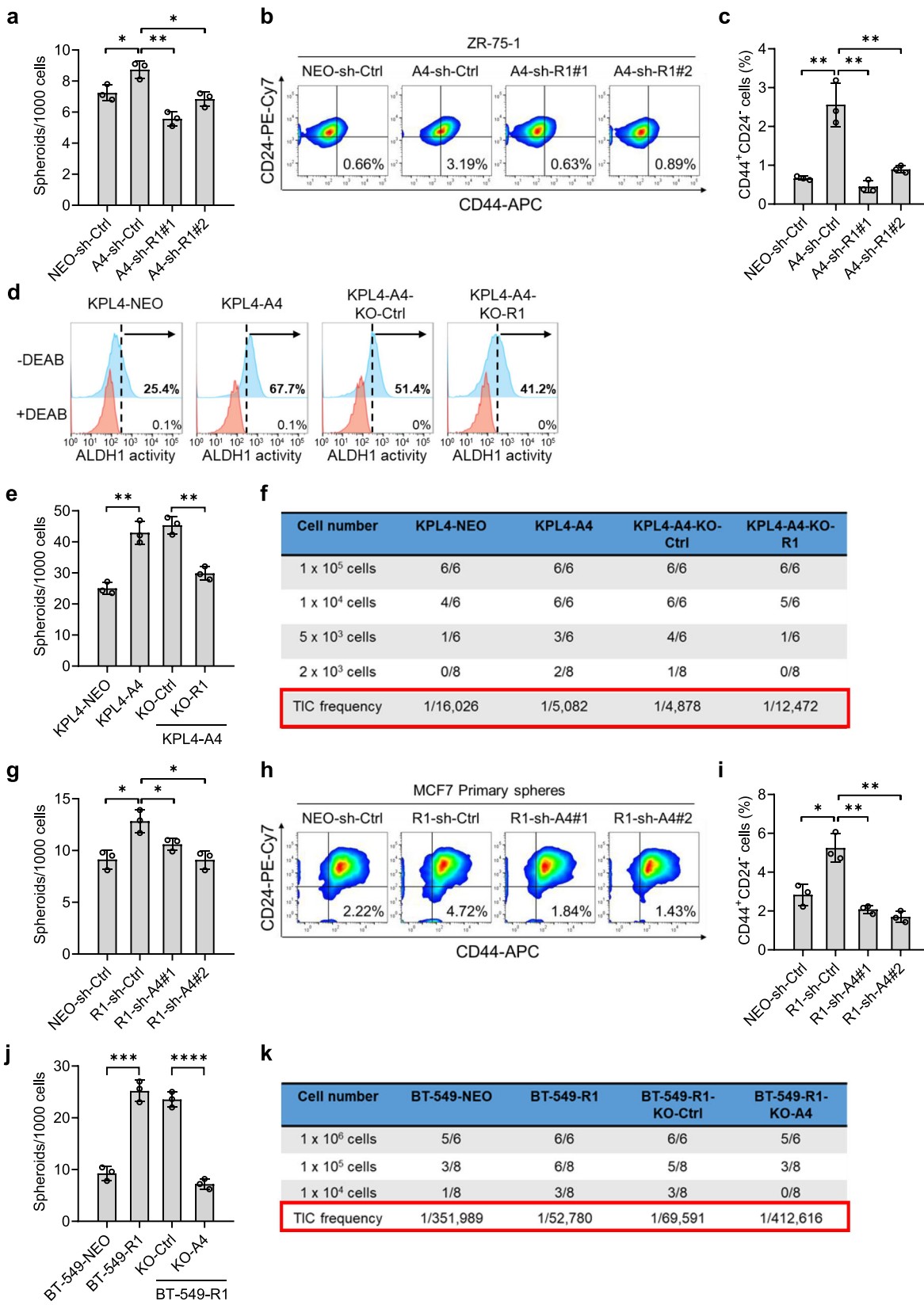

Fig. 7e). Together, these positive correlations between hRNase 1 level, EphA4 activation, and stem-like cell population supported the pathological relevance of the hRNase 1–EphA4 axis in tumor initiation with potential clinical implications for breast cancer treatment.

## Discussion

On the basis of our findings, we present a model (Fig. 7f) showing an unconventional role of hRNase 1 as a secretory ligand of EphA4 that induces breast tumor initiation. Elevated serum hRNase 1 binds to EphA4 and triggers EphA4 signaling in an

**Fig. 6 EphA4 positively regulates hRNase 1-mediated breast tumor initiation. a** Quantification of spheroid-formation assay of the indicated ZR-75-1 stable clones. NEO-sh-Ctrl vs A4-sh-Ctrl, $^*p = 0.0260$, A4-sh-Ctrl vs A4-sh-R1#1, $^{**}p = 0.0015$, A4-sh-Ctrl vs A4-sh-R1#2, $^*p = 0.0105$. **b** Flow cytometric analysis of membrane CD44 and CD24 expression in the indicated ZR-75-1 stable clones. **c** Quantification of flow cytometric analysis from (**b**). NEO-sh-Ctrl vs A4-sh-Ctrl, $^{**}p = 0.0044$, A4-sh-Ctrl vs A4-sh-R1#1, $^{**}p = 0.0033$, A4-sh-Ctrl vs A4-sh-R1#2, $^{**}p = 0.0072$. **d** Flow cytometric analysis of ALDH1 activity by ALDEFLUOR assay of the indicated KPL4 stable clones. Right of each panel, the percentage of ALDEFLUOR-positive cells. **e** Quantification of spheroid-formation assay of the indicated KPL4 stable clones. NEO vs A4, $^{**}p = 0.0016$, A4-KO-Ctrl vs A4-KO-R1, $^{**}p = 0.0016$. **f** Limiting dilution assay of the indicated KPL4 stable clones. **g** Quantification of spheroid-formation assay of the indicated MCF7 stable clones. NEO-sh-Ctrl vs R1-sh-Ctrl, $^*p = 0.0109$, R1-sh-Ctrl vs R1-sh-A4#1, $^*p = 0.0371$, R1-sh-Ctrl vs R1-sh-A4#2, $^*p = 0.01001$. **h** Flow cytometric analysis of membrane CD44 and CD24 expression in the indicated MCF7 stable clones. **i** Quantification of flow cytometric analysis from (**h**). NEO-sh-Ctrl vs R1-sh-Ctrl, $^*p = 0.0103$, R1-sh-Ctrl vs R1-sh-A4#1, $^{**}p = 0.0019$, R1-sh-Ctrl vs R1-sh-A4#2, $^{**}p = 0.0014$. **j** Quantification of spheroid-formation assay of the indicated BT-549 stable clones. NEO vs R1, $^{***}p = 0.0003$, R1-KO-Ctrl vs R1-KO-A4, $^{****}p < 0.0001$. **k** Limiting dilution assay of the indicated BT-549 stable clones. In (**b**, **h**) the percentages of CD44+CD24− cells are shown in the right lower quadrant of each panel. In (**f**, **k**) the number of tumor-forming mice within each group is shown in the panel. All error bars represent mean ± SD. Data are representative of three independent experiments; $^*p < 0.05$, $^{**}p < 0.01$, $^{***}p < 0.001$, $^{****}p < 0.0001$, two-tailed unpaired t test (**a**, **c**, **e**, **g**, **i**, **j**). Source data are provided as a Source Data file.

autocrine/paracrine manner, which in turn promotes breast cancer initiation via the IKK/NF-κB and MEK/Erk activating pathways. In addition to the physiological roles of hRNase 1 involved in hemostasis, inflammation, and innate immunity[2], the overabundance of hRNase 1 can contribute to cancer progression as a dysregulation disease process. Notably, hRNase 1 differs from classical ephrin ligands in that hRNase 1 is a native protein secreted in serum and plasma, thus it activates EphA4 in an autocrine/paracrine manner, whereas classical ephrin ligands are membrane-bound proteins to interact with EphA4 in a cell–cell contact and juxtacrine manner. To the best of our knowledge, it has been well documented that a soluble ligand, such as EGF and FGF, can activate its cognate receptors on the cells of its origin (autocrine) or in nearby cells (paracrine)[54,55], whereas a ligand that remains membrane-bound, such as ephrins, activates receptors mainly through a juxtacrine manner[20]. In our studies, hRNase 1 as a natively secretory protein freely circulating in various body fluids, e.g., serum and plasma, is demonstrated as a ligand of EphA4 receptor that enhances breast cancer stem-like properties, namely that hRNase 1-mediated EphA4 activation is mainly through an autocrine/paracrine mechanism. However, we do not exclude a possibility that any other unknown pathways are involved as well.

Notably, the levels of serum/plasma hRNase 1 were determined at a concentration ranging from 0.1 to 1.5 μg/ml in patients with different cancer types[14,56], which belongs to a group of plasma proteins with medium-to-high abundance (medium abundance: 0.1–1 μg/ml; high abundance: >1 μg/ml)[57–59]. The relatively high concentration of serum/plasma hRNase 1 in the tumor microenvironment may predominantly contribute to EphA4 activation via an autocrine or paracrine pathway under low cell density in early stage of tumorigenesis. Moreover, this hRNase 1-initiated EphA4 activation and breast cancer stemness may be sustained in conjunction with contact-dependent juxtacrine signaling by ephrins from different cells such as tumor-associated monocytes and macrophages[30]. Thus, targeting the hRNase 1–EphA4 axis may be a promising therapeutic strategy against breast cancer by decreasing tumor-initiating capability and may open a new direction that has been overlooked in our understanding of receptor biology in cancer progression.

The binding affinity of hRNase 1 to EphA4 (Kd = 92.4 nM) was about 16.5-fold less than that of ephrin-A5 to EphA4 (Kd = 5.6 nM) (Fig. 5b, c), but accumulating evidence indicates that ligand–receptor interaction with low binding affinity can still contribute to significant functionality. For example, ephrin-B2 ligand plays a crucial role in neural development through EphA4 as an axonal guidance receptor[60,61], although in vitro binding affinity of ephrin-B2 toward EphA4 is 5–300 times weaker than class A ephrins interacting with EphA4[62]. Moreover, two low-

affinity EGFR ligands, epiregulin and epigen, induce weaker receptor dimerization than EGF (a high-affinity EGFR ligand) and result in sustained EGFR activating signaling, leading to cell differentiation rather than EGF-mediated proliferation[63]. Another human RNase called hRNase 5 acts as an EGFR ligand and a serum biomarker to predict EGFR inhibitor response in pancreatic cancer, but EGF fails to have such predictive function although the binding affinity of hRNase 5 to EGFR is about 23-fold less than EGF for EGFR binding[14].

Of note, although EphA4 and EphA5 share high sequence homology and structural similarity, their LBDs exhibit distinct ligand-binding specificities through conformational changes[64]. In addition, crystal structural analyses show that EphA4 has a high degree of conformational plasticity in its LBD, able to facilitate the ephrin binding and signaling cross-class A and B[62]. This significant conformational plasticity of EphA4 as a structural chameleon may provide another molecular basis for hRNase 1 reactivity. Further investigations and a co-crystal structure analysis would be required to reveal more detailed mechanistic insights toward molecular interactions between EphA4 and hRNase 1 in the future. Previously, Tsuda and colleagues reported that a protein called amyotrophic lateral sclerosis 8 (ALS8) known to be associated with the pathogenesis of ALS is cleaved and then secreted to bind directly to the EphA4 ectodomain;[65] however, it is not yet clear whether ALS8 activates downstream signaling of EphA4. In addition to binding to EphA4, our study indicated that hRNase 1 as a native secretory protein also activates EphA4 signaling. It will be worthwhile to determine whether ALS8 also recognizes downstream substrates of EphA4 signaling like hRNase 1.

It is possible that hRNase 1 could also interact with and activate other Eph receptors or RTKs through hRNase 1-mediated heterodimerization/oligomerization with EphA4 similar to EGFR's interaction with HER2, HER3, IGF-1R, and cMET[66–68]. Secondary signaling events could also indirectly induce the interaction between hRNase 1 and other receptors. These possibilities remain to be further investigated. A recent study indicated that hRNase 5 preserves the stemness of hematopoietic stem/progenitor cells and attenuates radiation-induced bone marrow failure[69]. Whether the stemness-promoting role of hRNase 1 applies to stem cell development remains unexplored. Although the biological functions of hRNase 1 are not completely defined, its biochemical properties and post-translational modifications, such as glycosylation, have been extensively investigated[7,70]. There are three N-linked glycosylation sites of hRNase 1 reported (Asn-34, Asn-76, and Asn-88), and the pattern of hRNase 1 on gel electrophoresis is usually heterogeneous due to heavy glycosylation as illustrated by a range of migrated bands (~15–36 kDa)[6,7]. In addition, hRNase 1 exhibits differential levels of

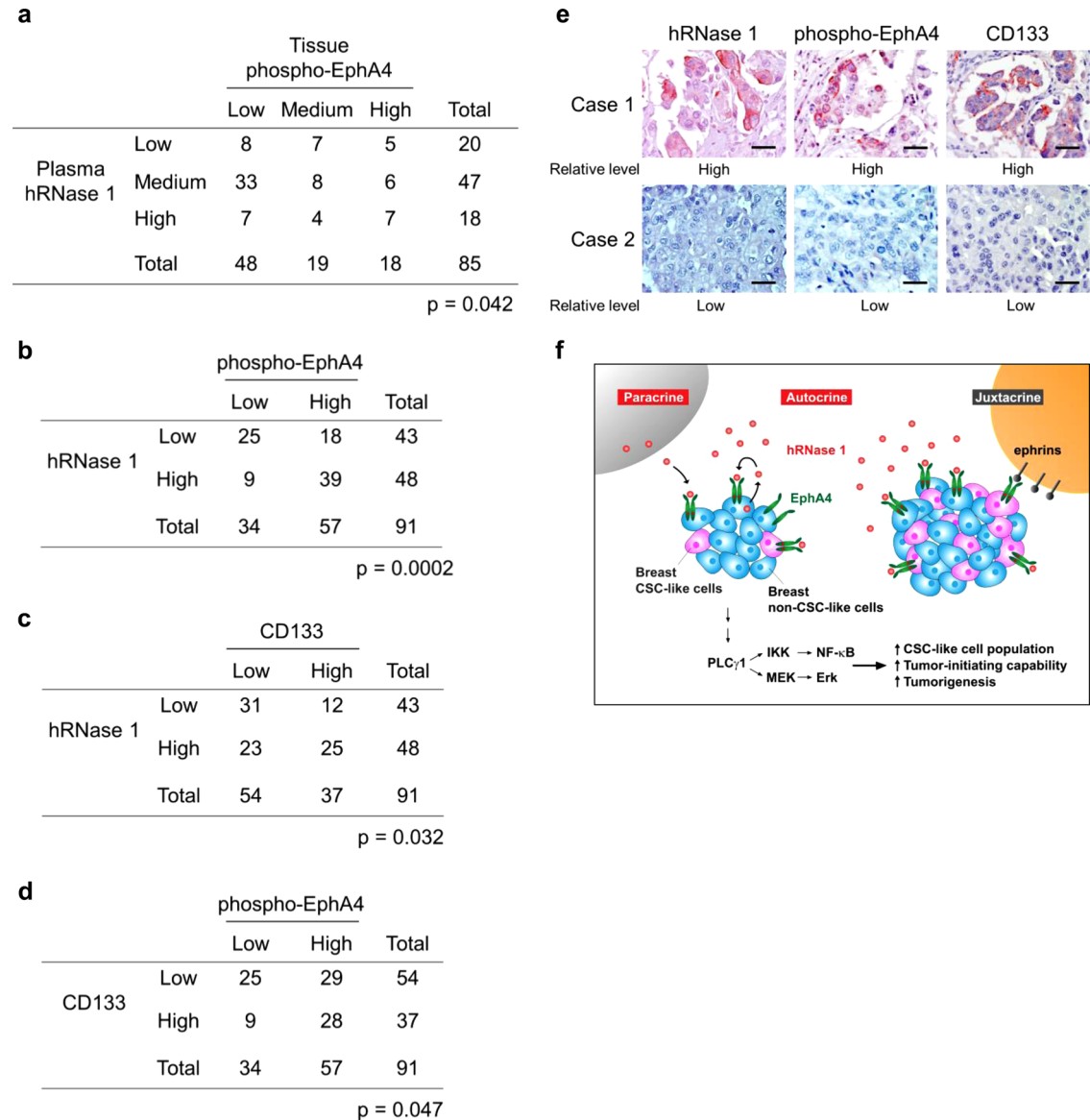

**Fig. 7 Pathological relevance of hRNase 1 expression, EphA4 activation, and CD133 in breast cancer. a** Quantification for the correlation between tissue phospho-EphA4-Y779 and plasma hRNase 1 in the paired breast cancer patients. Chi-squared test. **b–d** Quantification of IHC staining for the correlation between hRNase 1 and phospho-EphA4-Y779 (**b**), hRNase 1 and CD133 (**c**), and CD133 and phospho-EphA4-Y779 (**d**) by human breast TMA analysis (Pantomics Inc., #BRC1021). Fisher's exact test. **e** Two representative cases of IHC staining for (**b–d**). The experiment was performed an additional time with similar results. Bar, 50 μm. **f** A proposed model of hRNase 1 as a secretory ligand of EphA4 to positively regulate breast tumor initiation. In brief, elevated levels of serum hRNase 1 induces its binding to EphA4 and triggers EphA4 signaling (red stars) in breast tumor cells in an autocrine/paracrine manner, which in turn promotes breast tumor initiation via the IKK/NF-κB and MEK/Erk activating pathways. Such EphA4 activation and CSC-like state triggered by hRNase 1 may be sustained in conjunction with juxtacrine signaling via the classical membrane-bound EphA4 ligands, ephrins. Artwork by H-H.L., Y-N.W., and M-C.H.

glycosylation in the urine, seminal plasma, kidney, and brain[5,71], raising a possibility that glycosylation may affect certain tissue-specific roles of hRNase 1. Moreover, the N-linked glycan on Asn 88 of serum hRNase 1 acts as a diagnostic marker for pancreatic cancer[40]. Thus, it would be of interest to further explore whether different glycosylated forms of hRNase 1 associate with any specific roles during cancer progression. It is worthwhile to note that, in addition to hRNase 1, hyperglycosylation of RNase proteins in cancer cell lines was also observed in human RNase 2 in promyelocytic leukemia cells; treatment of PNGase F glycosidase resulted in a change of RNase 2 from a heterogeneous to a homogenous pattern, in terms of molecular weight from ~22–45 kDa reduced to ~15 kDa, respectively[72]. Together, these studies

have suggested that RNase protein hyperglycosylation would be a common feature with heterogeneity shared in cancer cells.

It is noteworthy that another RNase protein, called hRNase 5, has been known as a ligand of EGFR RTK in pancreatic cancer as evidenced by the results of a phospho-RTK antibody array[14], which is a similar approach used to identify the hRNase 1–EphA4 pair. Analysis through primary sequence alignment between hRNase 5 and EGF, an EGFR classical ligand, shows that a conserved residue on glutamine 93 (Q93) of the C-terminus of hRNase 5 is critical for EGFR binding, and EGFR-ECD is required for binding to hRNase 5[14]. In line with these results, we demonstrated that hRNase 1 binds to EphA4 via its C-terminus and the interaction requires EphA4-ECD, suggesting an

intriguing mechanism that the C-terminal region of hRNases may generally participate in the essential binding of their cognate RTKs-ECD. In addition, the ribonucleolytic activity of hRNase 5 is not required for EGFR ligand-like function[14]; likewise, we found that hRNase 1 increased breast cancer stem-like properties independently of its ribonucleolytic activity. Recently, human RNase 7 has been found to serve as a high-affinity ligand for ROS1 RTK in hepatocellular carcinoma[34]. Together, our findings with another two studies including hRNase 5–EGFR and hRNase 7–ROS1 pairs uncover important roles of secretory RNases in human malignancies, which may help to create a paradigm shift in the understanding of the ligand–receptor relationship between secretory RNase and cell membrane RTK families. Together, our findings reveal an interesting concept that the cell-contact-dependent signaling of the Eph family via secretory ligands may exist to regulate the receptors' activities. Given that the RNase–RTK ligand–receptor axes contribute to cancer development, furthering our understanding of the unconventional RNase–RTK pairs may lead to more non-invasive serum biomarkers for precision medicine and new therapeutic targets for cancer treatment as RNases are enriched in serum.

## Methods

**Cell lines and treatment**. All cell lines used in this study were obtained from American Type Culture Collection (ATCC). Cell lines have been validated by STR DNA fingerprinting at MD Anderson Cancer Center and routinely tested for mycoplasma contamination. All cell lines were cultured in the medium recommended by ATCC. Treatment with recombinant hRNase 1 protein purified from HEK293 cells (Sino Biological Inc. #13468-H08H-100) was carried out at a concentration of 1 μg/ml for 30 min or the indicated time after serum-free starvation for 3 h. The concentrations of inhibitors used were QNZ (30 nM, Selleck Chemical, #S4902), Bay 11-7821 (0.2 μM, Tocris Bioscience, #1744/10), GSK1120212 (1 nM, Apexbio Technology, #A301850), PD-0325901 (0.2 or 2 μM, Tocris Bioscience, #4192/10), MK-2206 (20 nM, Apexbio Technology, #A301010), and Dasatinib (2 nM, LC Laboratories, #D-3307).

**Plasmids, shRNA clones, and knocking out constructs**. The pCDH-R1 (Flag-tagged hRNase 1), pCDH-hRNase 5 (Flag-tagged hRNase 5), and pCDH-A4 (Myc-tagged EphA4) expression vectors were generated by inserting the full-length cDNA (hRNase 1: NM_002933.4; hRNase 5: NM_001145; EphA4: BC105002) into the lentiviral vector pCDH-CMV-MCS-EF1-puro/NEO. pCDH-R1-H12A (Flag-tagged hRNase 1 with H12A mutation) was generated and derived from pCDH-R1 plasmid by site-directed mutagenesis (Supplementary Table 1). The inserts of pCDH-A4-ECD and pCDH-A4-ICD (ECD: 1-547 amino acids of EphA4; ICD: 570-986 amino acids of EphA4) were obtained by PCR amplification from pCDH-A4 and inserted to pCDH-CMV-MCS-EF1-NEO. Full-length hRNase 1 was cloned into GST expression vector pGEX6P1 to generate pGEX-R1 (GST-tagged hRNase 1). The pCDH-CMV-MCS-EF1-puro/NEO expression vectors (CD510B-1 and CD514B-1) were purchased from System Biosciences. The pGEX6P1 (#28-9546-48) was purchased from GE Healthcare Bio-Sciences. The shRNA vectors pGIPZ-sh-Ctrl (Non-silencing shRNA, RHS4346), pGIPZ-sh-R1#1 against hRNase 1 (Dharmacon, V3LHS_313141), and pGIPZ-sh-R1#2 against hRNase 1 (Dharmacon, V2LHS_32407) were obtained from the shRNA/ORF Core Facility at the University of Texas MD Anderson Cancer Center. The shRNA vectors pLKO.1-sh-Ctrl (non-target shRNA control, SHC016), pLKO.1-sh-A4#1 against EphA4 (TRCN0000344511), and pLKO.1-sh-A4#2 against EphA4 (TRCN0000196950) were purchased from Sigma-Aldrich. For knockout experiments, we inserted a non-targeting control gRNA sequence (TAAACAAAAAGGAAATAGTT) from the GeCKOv2 libraries, which does not target any human genes based on prediction[73], into pLentiCRISPRv2 vector as a control (KO-Ctrl). To knockout hRNase 1 (KO-R1) and EphA4 (KO-A4), three different regions of hRNase 1 (NM_002933.4) and EphA4 (NM_004438.4) were targeted using pLentiCRISPRv2 vectors (Addgene #98290). The targeting sequences (5′ to 3′) are listed in the Supplementary Table 1. All plasmids were verified by DNA sequencing.

**Generation of hRNase 1 knockdown and reconstitution stable cells**. To establish stable cell lines with hRNase 1 knockdown or with the control counterpart, we conducted lentiviral packaging via transient transfection of 1 μg pGIPZ-sh-R1#1, pGIPZ-sh-R1#2 or pGIPZ-sh-Ctrl together with 1 μg pCMV-VSVG and 0.5 μg pCMV-dR8.91 expression plasmids in $5 \times 10^5$ 293T cells. After 72 h, 3 ml of conditioned medium from the transfectants were collected, centrifuged at 6000g for 15 min, and flew through 0.45 μm filter, followed by incubating with targeted cells at 5 μg/ml polybrene (EMD Millipore, #TR-1003-G) for lentiviral transduction.

After transduction for 16 h, cells were replenished with 3 ml of complete medium for 1 day, and subjected to puromycin selection at 1 μg/ml for another 3 days to establish stable cells. For the reconstitution of hRNase 1 resistant to sh-R1#2-mediated knockdown in the stable cells, we first generated a modified hRNase 1 construct by introducing silent mutations of hRNase 1 (from TCCACCTACTG-TAACCAA to TCAACATATTGCAATCAA corresponding to the peptide sequence STYCN at amino acids 23–27) on the pCDH-R1 plasmid through site-directed mutagenesis. Then the modified pCDH-R1 with silent mutations and vector control plasmid were utilized in pGIPZ-sh-R1#2-mediated knockdown stable cells through lentiviral transduction as mentioned above, followed by G418 antibiotic selection at 750 μg/ml to generate stable cells for hRNase 1 reconstitution or the control counterpart, respectively.

**Generation of stable cell lines**. Stable cells were selected with antibiotics according to the type of vector used. G418 (400 μg/ml) or puromycin (0.5 μg/ml) was used to select for BT-549 stable cells; G418 (800 μg/ml) or puromycin (1 μg/ml) for KPL4 stable cells; and puromycin (0.3 μg/ml) for 293T stable cells. All stable transfectants were selected from a pool of clones. The duration of stable cell selection was at least 4 weeks. G418 (#11811031) was purchased from Thermo Fisher Scientific. Puromycin (ant-pr-1) was obtained from InvivoGen Corporation.

**Generation of recombinant hRNase 1 proteins**. We produced recombinant hRNase 1 protein from *E. coli* by using a similar protein expression protocol referred to published studies[14,34]. For producing purified GST-hRNase 1 recombinant protein, hRNase 1 cDNA without signal peptide sequence was inserted into pGEX6P1 to express GST-hRNase 1 protein in BL21(DE3)-competent *E. coli*, followed by GST-tagged protein purification assay. For generating purified hRNase 1 protein with a Myc and a 6XHis tag at the C-terminus, hRNase 1-Myc-His fusion cDNA fragment was amplified from the hRNase 1 expression plasmid previously established in pcDNA6/Myc-His A vector, and inserted into pSJ3 vector to express hRNase 1 protein in BL21(DE3)-competent *E. coli*, followed by His-tagged protein purification assay.

**Glycosidase pretreatment of recombinant hRNase 1 protein**. Following the manufacturer's instruction of glycoprotein treatment with PNGase F glycosidase (NEB Inc., #P0704), 3 μg of recombinant hRNase 1 protein (Sino Biological Inc., #13468-H08H) was combined with 1 μl of 10× Glycoprotein Denaturing Buffer and water to make up a 10 μl total reaction volume. The mixture was denatured by heating at 100 °C for 10 min and chilled on ice for 2 min, and 2 μl of 10× Glyco-Buffer 2, 2 μl of 10% Nonidet P-40, and 6 μl of water were then added to make up a 20 μl total reaction volume. The mixture was then incubated at 37 °C overnight with or without 1 μl of PNGase F to keep the final glycerol concentration equal to 5% and subjected to in vitro binding assay.

**Prognostic analysis of cancer patients from databases**. The Kaplan-Meier Plotter database (http://kmplot.com/analysis/)[35] was used to analyze the correlation between the expression levels of hRNase family and survival of cancer patients. A Kaplan-Meier overall survival analysis of breast cancer patients divided by the median expression level of hRNase 1 was performed in a platform for exploring Gene Expression patterns across Normal and Tumor tissues named GENT2 (http://gent2.appex.kr/gent2/)[37]. The UCSC Cancer Genome Browser (http://xena.ucsc.edu/welcome-to-ucsc-xena/)[38] was utilized to validate prognostic effect of hRNase 1 with Kaplan-Meier survival analysis. In brief, survival analysis was performed using the interpreted expression profile of TCGA breast invasive carcinoma by RNA sequencing (dataset ID: TCGA_BRCA_exp_HiSeqV2) downloaded from the UCSC Cancer Genome Browser. The median expression of hRNase 1 was used for patient stratification. A corrected *p* value < 0.05 was considered as significant.

**Immunohistochemical staining and scoring of human breast tumor tissues**. Human breast tumor tissue array was obtained from Pantomics (#BRC961 or #BRC1021). Immunohistochemical staining[74] was performed using hRNase 1 antibody (1:100; Sigma-Aldrich, #HPA001140), phospho-EphA4 antibody (1:150; LifeSpan BioSciences, #LS-C381624), or CD133 antibody (1:200; Abcam, #AB216323) (Supplementary Table 2). Tissue specimens were incubated with primary antibody and biotin-conjugated secondary antibody, and then mixed with an avidin–biotin–peroxidase complex. Amino-ethylcarbazole chromogen was used for visualization. Protein expression was ranked according to Histoscore (H-score) method. H-score was evaluated by a semi-quantitative assessment of both the intensity of staining and the percentage of positive cells. The range of scores was from 0 to 300. Cases with H-score higher than average were considered as high expression and those with H-score equal to or less than average as low expression.

**Human serum and tissue samples**. Human serum, plasma, and tissue samples used were collected following the guidelines approved by the Institutional Review Board at The University of Texas MD Anderson Cancer Center (LAB05-0127 and LAB05-0131), Harbin Medical University Cancer Hospital (KY2016-

34), China Medical University Hospital (CMUH105-REC1-064), Taipei Veterans General Hospital (2018-09-001AC), and Kaohsiung Medical University Hospital [KMUHIRB-G(II)-20170030 and KMUHIRB-F(I)-20160006]. All patients are females with breast cancer (Fig. 1f, mean ± SD age, 52.2 ± 9.8 years; median age, 52.5 years; Supplementary Fig. 4a, mean ± SD age, 56.1 ± 9.5 years; median age, 56.5 years; Supplementary Fig. 4b, mean ± SD age, 52.1 ± 12.1 years; median age, 53.5 years; Supplementary Fig. 4c, mean ± SD age, 53.9 ± 10.2 years; median age, 56.0 years). Written informed consent was obtained from patients in all cases at the time of sample collection. All clinical information validated our results without selection bias.

**Scoring of the paired human plasma and tissue samples**. The paired human plasma and tissue samples were obtained from 85 patients with breast cancer. The range of plasma hRNase 1 concentration was from 67 to 801 ng/ml. For patient stratification in this experiment, samples of plasma hRNase 1 with the concentration value below 25% percentile (<85 ng/ml) were regarded as low expression, above 75% percentile (≥223 ng/ml) as high expression, and 25–75% percentile as medium expression. Tissue samples with the staining intensity scored as 0 and 1 were regarded as low expression, 2 as medium expression, and 3 as high expression.

**Animal studies**. All animal experiments were performed according to animal welfare guidelines approved by MD Anderson's Institutional Animal Care and Use Committee. Mice were maintained at an ambient temperature of 70 ± 2 °F and relative humidity of 30–70% under a 12-h light/12-h dark cycle. The experiment has no statistical method used to estimate sample size. For the in vivo limiting dilution assay, 6-week-old female BALB/c nude mice were purchased from Jackson Laboratories (Bar Harbor, ME, USA). The total number of mice for each experiment is indicated in the figure or table. The indicated number of cells in suspension in 50 μl of DMEM/F12 (Corning, #10-090-CV) was mixed with 50 μl of the Matrigel (Thermo Fisher Scientific, #CB40230C). The cell mixtures were subcutaneously injected into the flanks of mice. Tumor incidence was monitored 6 weeks after inoculation of tumor cells. TIC frequencies of each experiment were estimated using the ELDA web-tool[42]. For the in vivo tumorigenesis assay, 6-week-old female nude mice were orthotopically injected with $5 \times 10^4$ cells of mouse 4T1 stable clones as indicated. Compound 1 (cpd1; 50 mg/kg solved in 5% DMSO with 10% Tween 80 in PBS; Santa Cruz Biotechnology, #sc-314230) or a matched vehicle was administrated on day 10 after cell injection by intraperitoneal injection once per 2 days, continuing for 2 weeks. Tumor weight was measured at the endpoint.

**Preparation of conditioned medium (CM) from cell culture**. Cells were cultured in 8 ml serum-free medium. After 24 h, the medium containing secreted proteins was collected and filtered by 0.45 μm filters to remove cell debris. Amicon Ultra-15 Centrifugal Filter Units (EMD Millipore, #UFC900324) was used to concentrate the CM at 5000g to 400 μl.

**Detection of hRNase 1 in human serum and CM by ELISA**. Serum collection from breast cancer patients and healthy individuals was approved by Institutional Review Board of MD Anderson Cancer Center and informed consent was obtained from all subjects. The concentration of hRNase 1 in serum or CM was determined by ELISA Kit for hRNase 1 (Cloud-Clone Corp., #SEA297Hu) according to the manufacturer's instructions. Briefly, standards or samples were added to the appropriate wells of a pre-coated and ready-to-use 96-well plate for 1 h at 37 °C, followed by incubating with a biotin-conjugated antibody against hRNase 1 (Detection Reagent A) for 1 h at 37 °C. After washing, horseradish peroxidase (HRP)-conjugated avidin (Detection Reagent B) was added to each well, and the mixture was incubated for 1 h at 37 °C. After washing, 90 μl of TMB substrate solution was added to each well for 10–20 min at 37 °C. Finally, the enzyme–substrate reaction was stopped by 50 μl of sulfuric acid solution. Only the well added with hRNase 1, biotin-conjugated antibody, and HRP-conjugated avidin displayed a change of color, which was subsequently measured at a wavelength of 450 nm by a BioTek Synergy™ Neo multi-mode reader (BioTek Instruments). The concentration of hRNase 1 in the samples was calculated by comparing the optical density value of the samples to the standard curve.

**Soft agar colony-formation assay and sphere-forming assay**. For the soft agar colony-formation assay, cells ($5 \times 10^3$) from each clone were mixed with 1 ml of medium with 0.3% agar (Sigma-Aldrich, #A5421). The agar–medium mixture containing cells was placed on top of a bottom layer (1 ml of medium with 0.6% agar) in each well of a 6-well tissue culture plate (FALCON, #353046). After incubation for 2 weeks, the colonies were stained with crystal violet. The viable colonies were counted. For the sphere-forming assay, cells ($5 \times 10^3$ for BT-549 cells; $1 \times 10^4$ for KPL4, MCF7, ZR-75-1, and BT-474 cells) were suspended in 2 ml of complete MammoCult™ Human Medium (STEMCELL Inc., #05620) and then added into each well of a 6-well ultra-low attachment plate (Corning, #3471). After 14-day incubation, the number of spheroids larger than 50 μm was counted.

**Flow cytometric analysis of CD24/CD44 expression and ALDEFLUOR assay**. For the analysis of CD24 and CD44 expression on cell membrane, $5 \times 10^5$ of cells were collected in Cell Staining Buffer (BioLegend, #420201) and stained with PE-Cy™7-conjugated anti-CD24 (1:100 for 20 min; BD Biosciences, #561646) and APC-conjugated anti-CD44 antibody (1:60 for 20 min; BD Biosciences, #559942) by using PE-Cy™7 Mouse IgG2a (1:100 for 20 min; BD Biosciences, #552868) and APC Mouse IgG2b (1:60 for 20 min; BD Biosciences, #555745) as control staining. Stained cells were analyzed by BD FACSCanto II cytometer and data were acquired by BD FACSDiva v8.0.2 software and processed by FlowJo v10.7.1 software (BD Biosciences). ALDEFLUOR assay was carried out using the ALDEFLUOR assay kit (STEMCELL Inc., #101700) according to the manufacturer's instructions. The ALDH1 inhibitor, diethylaminobenzaldehyde (DEAB), was used as a negative control. The processed cells were evaluated by BD FACSCanto II cytometer and data were acquired by BD FACSDiva v8.0.2 software and analyzed by FlowJo v10.7.1 software (BD Biosciences).

**RNase enzymatic activity assay**. Ambion RNaseAlert Lab Test kit (Thermo Fisher Scientific, #AM1964) was used to detect RNase enzymatic activity according to the manufacturer's instruction. Briefly, 5 μl of 10-fold RNaseAlert buffer was added to a tube containing the fluorescent substrate, and then mixed with a total of 45 μl of the tested CM with 1/1000 dilution by RNase-free water to reduce background RNase activity. The mixture was sequentially placed on a well of a 96-well plate. The real-time fluorescence data were collected at 1 min intervals for 21 min using a BioTek Synergy™ Neo multi-mode reader (BioTek Instruments).

**Human phospho-RTK antibody array**. Proteome Profiler Human Phospho-RTK Array Kit (R&D Systems, #ARY001B) was used to detect the potential activation of RTK signals by hRNase 1 treatment. All procedures were performed according to the manufacturer's instruction. Briefly, capture antibodies for specific RTKs were spotted in duplicate onto nitrocellulose membranes, prepared with the kit. Cell lysates (600 μg) were incubated with the array membrane at 4 °C overnight. After washing away unbound material, proteins in cell lysates containing phosphorylated tyrosine residues bound to the capture antibodies of RTKs were detected by a pan anti-phospho-tyrosine antibody conjugated to HRP. Last, the binding signal was measured by using chemiluminescent detection reagents and ImageQuant LAS 4010 (GE Healthcare).

**Western blotting**. Cells were harvested and lysed in the lysis buffer (1.25 M urea and 2.5% SDS) after washing with phosphate-buffered saline (PBS). The viscosity of the lysate was removed by sonication, and protein concentration measured by Thermo Scientific Pierce BCA Protein Assay (Thermo Fisher Scientific, #PI-23227). Primary antibodies performed are listed in the Supplementary Table 2. WB detection was performed using chemiluminescent detection reagents (Bio-Rad #170-5061 or Thermo Scientific #34075) and ImageQuant LAS 4010 (GE Healthcare).

**Co-immunoprecipitation and GST pull-down assays**. For co-IP assay, before scraping adherent cells from the dish, hRNase 1-treated cells were fixed with 1 mM dithiobis(succinimidylpropionate) (Thermo Scientific, #3407522585) for 30 min at room temperature. Cells were then collected and lysed by Pierce IP Lysis Buffer (Thermo Fisher Scientific, #87787) including 1-fold protease inhibitor cocktail (Roche # 4693116001), and phosphatase inhibitor cocktail 2 and 3 (Sigma-Aldrich, #P5726 and P0044). Cell lysates (1 mg) were mixed with anti-EphA4 (4 μg; Santa Cruz Biotechnology, #sc-921), anti-EphA5 (6 μg; Santa Cruz Biotechnology, #sc-1014), or IgG control antibody (6 μg; Santa Cruz Biotechnology, #sc-2027) overnight at 4 °C, and then pulled-down with protein A/G magnetic beads (1:30; Cell Signaling Technology, #8740 and 8687) at 4 °C for 2 h. For GST pull-down assay, after purification of GST fusion protein using glutathione magnetic beads (Thermo Fisher Scientific, #88821), cell lysates (1 mg) from 293T were incubated with GST fusion protein-binding magnetic beads at 4 °C for 6 h. The magnetic beads bound with target proteins were washed with the same lysis buffer and eluted with Blue Loading Buffer (Cell Signaling Technology, # 7722) at 95 °C for 7 min before WB.

**Duolink in situ proximity ligation assay**. PLA was carried out to investigate the proximity of epitopes recognized by the anti-Flag and anti-EphA4 antibodies that represent the association of hRNase 1 with EphA4 in cancer cells using the Duolink® In Situ Red Starter kit (Sigma-Aldrich Corporation, #DUO92101) according to the manufacturer's instruction. Briefly, cells were fixed on the slide with 4% paraformaldehyde for 15 min and permeabilized with 1% Triton X-100 for 15 min. After blocking, anti-EphA4 (1:50; Santa Cruz Biotechnology, #sc-921) and anti-Flag (1:100; Sigma-Aldrich, #F3165) antibodies were incubated with cells overnight at 4 °C. Subsequent ligations and detections were carried out in accordance with the manufacturer's recommendations. Cells treated with CM from 293T-pCDH-hRNase 5 were shown as a negative result compared to the positive result for experiments. Staining with anti-EphA4 antibody alone was performed as a negative control of the experimental procedure.

**Dimerization/oligomerization assay**. 293T cells ectopically expressing Myc-tagged EphA4 or vector control were starved in serum-free medium for 24 h. After starvation, cold PBS with or without 2 μg/ml hRNase 1 was added onto plates for 2 h at 4 °C. Then, cells were washed with cold PBS three times and incubated for 3 h at 4 °C with 1 mM cross-linker bis(sulfosuccinimidyl)suberate (BS$^3$; Thermo Scientific) in PBS. After washing three times with cold PBS, cross-linking reactions were stopped by incubating cells in 20 mM Tris buffer (pH 7.5) for 15 min at room temperature. Cells were subsequently lysed and cell lysates were analyzed by WB in a non-reducing and non-denaturing condition.

**Detection of ligands-EphA4 binding affinity by ELISA**. ELISA 96-well plates were captured with 3 μg/ml EphA4 antibody (EMD Millipore, #AP1173) or normal mouse IgG (Santa Cruz) as a negative control in 0.2 M sodium phosphate buffer (pH 6.5) at 100 μl/well overnight at room temperature. The plates were then rinsed three times with PBS with 0.05% Tween-20 (PBST) and blocked with 200 μl/well of 1% BSA solution containing 0.05% Tween-20 at 37 °C for 3 h. After rinsing three times with PBST, 100 μl/well of BT-549-RIPA lysates were added and incubated overnight at 4 °C. The plates were then washed three times with PBST, followed by the addition of recombinant hRNase 1 or ephrin-A5 at a series of diluted concentrations in RIPA buffer. After incubation overnight at 4 °C, wells were washed with 400 μl/well of PBST three times with shaking for 1 min, and 100 μl/well of biotin-conjugated detection hRNase 1 or ephrin-A5 antibodies in blocking buffer was added for incubation at room temperature for 2 h. The plates were washed with PBST three more times with shaking, and 100 μl/well of streptavidin-conjugated HRP (1:2,000 in blocking buffer) was added and incubated at room temperature for 1 h. The wells were washed again with PBST three times with shaking, and 100 μl/well of TMB as a peroxidase substrate was added and incubated for 30 min at room temperature. The reaction was terminated by the addition of 50 μl/well of stop solution. The optical density was determined at 450 nm, corrected by subtraction of readings at 570 nm, with use of a BioTek Synergy™ Neo multi-mode reader. The dissociation constant (Kd) was estimated by the above binding data and then transformed to create a Scatchard plot with the GraphPad Prism program (version 8; Prism Software Inc., San Diego, USA).

**Statistical analysis**. Data are shown as mean ± standard deviation (SD) or standard error of mean (SEM) as stated. A two-tailed unpaired $t$ test was used to compare the continuous variables between the two groups unless otherwise noted. ANOVA analysis was used if there were more than two data groups to compare. Chi-squared test was used to compare dichotomous variables. Kaplan-Meier estimation and log-rank test were used to compare the differences in overall survival period between patient groups. The control groups for all the statistical analyses were usually the first groups in the panels, unless specified otherwise in the figure legends. All statistical data of biological function assays were collected from at least two independent experiments and contained at least three technical replicates. The level of statistical significance was set at 0.05 for all tests. Statistical analyses were performed using GraphPad Prism software (version 8.0.0). Flow cytometry data were analyzed using FlowJo software (version 10.7.1). Data were quantified using ImageJ software program (version 1.52a; National Institutes of Health).

**Reporting summary**. Further information on research design is available in the Nature Research Reporting Summary linked to this article.

## Data availability

The publicly available databases used in this study are available in the Kaplan-Meier Plotter database (http://kmplot.com/analysis/), the Gene Expression patterns across Normal and Tumor tissues named GENT2 (http://gent2.appex.kr/gent2/), and the UCSC Cancer Genome Browser (http://xena.ucsc.edu/welcome-to-ucsc-xena/). The remaining data are available within the Article, Supplementary information or available from the authors upon request. Source data are provided with this paper.

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

## Acknowledgements

This study was supported in part by the following: an MDA Startup Fund; The University of Texas MD Anderson-China Medical University and Hospital Sister Institution Fund (to M-C.H.); Breast Cancer Research Foundation (Grant No. BCRF-20-070 to G.N. H., Y-N.W., M-C.H.); Health and welfare surcharge of tobacco products, China Medical University Hospital Cancer Research Center of Excellence in Taiwan (MOHW108-TDU-B-212-122015 and MOHW108-TDU-B-212-124024; to S-C.W.); Drug Development Center, China Medical University from The Featured Areas Research Center Program within the framework of the Higher Education Sprout Project by the Ministry of Education (MOE) in Taiwan (to S-C.W., M-C.H.); National Breast Cancer Foundation, Inc.; T32 Training Grant in Cancer Biology (5T32CA186892 to H-H.L. and L-C.C.); National Institutes of Health (CCSG CA016672); YingTsai Young Scholar Award (CMU108-YTY-04 to W-H.Y.); Ministry of Science and Technology Oversees Project for Post Graduate Research (MOST; 104-2917-I-564-003; to W-H.Y.); Ministry of Science and Technology (MOST; 109-2314-B-039-054; to W-H.Y.); Ministry of Education (Taiwan) Joint of International Talent Training Program (1040082029B to Y-H.W.); The 2019 AACR-Pfizer Immuno-oncology Research Fellowship (Grant No. 19-40-49 to Z.J.); Project Nn10 of Harbin Medical University Cancer Hospital (Nn102017-02 to D.P.); National Natural Science Foundation of China (81602323 and 81872149 to S.X.); Outstanding Youth Project of Heilongjiang Provincial Natural Science Foundation (YQ2019H027 to S.X.); and the INHA UNIVERSITY Research Grant (to J-H.C.).

## Author contributions

H-H.L., Y-N.W., W-H.Y., and M-C.H. designed and conceived the study; H-H.L., Y-N.W., W-H.Y., J.L.H., and M-C.H. wrote the manuscript; H-H.L., Y-N.W., W-H.Y., W.X., Y.W., L-C.C., Y-H.W., Z.J., Y.Q., Y-H.H., W-L.H., M.Y., J-H.C., and J.S. performed the experiments and analyzed the data; J.Y. assisted in the bioinformatics analysis; S.X. and D. P. provided breast cancer patient tissue and plasma samples and performed experiments; Y.Y., X.W., M-F.H., L-M.T., S-C.W., M-R.P., C-H.Y., and Y-L.W. provided the serum samples from healthy individuals and patients with breast tumor, and analyzed data; H.Y., D.Y., and G.N.H. provided scientific input; M-C.H. supervised the entire project and funding acquisition;

## Competing interests

The authors declare no competing interests.
