## [Peer Review File. · Nature Communications]

REVIEWER COMMENTS

Reviewer #1 (Remarks to the Author):

The authors showed that hRNase I binds to the extracellular domain of EphA4 and activates EphA4, leading to maintenance of breast cancer stem-like cells. It is interesting that hRNase I has unexpected roles for cancer stem-like cells. However, it is still unclear the physiological and pathological relevance of hRNase I for EphA4-signaling, since its native ligand ephrin is able to stimulate EphA4 in the tumor microenvironment and the affinity of EphA4 for ephrin seems to be much higher than for hRNase I (Fig. 5b, c). It is also very important to demonstrate the findings by using patient-derived cancer cells.

Specific points:

Fig. 1

It is important to show the Kaplan-Meier survival curves by using several cohorts and to use the median values for patient stratification. It is not appropriate to use Prognoscan for the analysis, since it does not use the median values for patient stratification. It is also helpful to show number of patients for each population belonging to hRNase I high and low.

Fig. 2b, Extended Fig. 8e

They performed soft agar colony assays. Though it is not very clear, they probably added serum into the media. This assay is useful for evaluation of anchorage-independent cell growth but not of stem-like properties.

Fig. 2j,k

It is important to show the results by using ELDA.

Fig. 5e

What are KYL peptides?

Fig. 7a

They should describe the levels of hRNase I in plasma in each population.

Reviewer #2 (Remarks to the Author):

This manuscript by Lee et al. describes a novel interaction between human ribonuclease 1 (hRNase 1) and EphA4 receptor tyrosine kinase. Serving as a novel ligand for EphA4, hRNase 1 activation of EphA4 promote breast tumor initiation in a paracrine manner that is distinct from canonical signaling through activation by membrane-bound ephrin-A ligands. The authors also report correlations between high hRNase 1 plasma levels and EphA4 activation in tumor tissue for breast cancer patients. The work is novel and of broad interest to scientists and clinicians in cancer research. However, there were several concerns. One major concern was about the choice of cell line models for experimental analyses that do not reflect breast cancer subtypes in which correlations between hRNase 1 and outcomes were found (i.e. use of TNBC models when correlations were found for patients with Luminal disease). Another major concern was the apparent lack of molecular subtyping for patients in which hRNase 1 levels in sera were analyzed. The use of ectopic expression in cell lines that already express high levels of hRNase 1 was also a concern for signaling studies, as was the apparent lack of BT-549 control cell responsiveness to pharmacologic inhibitors of NF- κ B and Mek. Minor issues included difficulty evaluating PLA data from small panels. It is recommended that the concerns outlined below be addressed prior to publication, which should significantly strengthen the story and impact of the paper:

1. Introduction: "Currently, there are no known cognate secretory ligands for the Eph receptors." This is not quite correct. Soluble ephrin-A ligands have been detected in the culture media of cell lines in models of breast and other cancers, where they activate EphA receptors and downstream signaling and cellular responses [Alford et al. 2007 *Exp Cell Res* 313: 4170-4179; Wykosky et al. 2008 *Oncogene* 27: 7260-73; Alford et al. 2010 *Cancer Cell Int.* 10: 41; Song et al. 2013 *PLoS One* 8: e74464]. Ephrin-A1 can be cleaved from the cell surface by metalloproteases [Ieguchi et al. 2013 *Biochem Biophys Res Commun* 440: 623-9; Ieguchi et al. 2014 *Oncogene* 33: 2179-90].

2. A transition paragraph between the end of the Introduction and the beginning of the Results sections would be helpful, with a few sentences that tie hRNase 1 to EphA4.

3. Figure 1 – Correlations between survival and hRNase 1 expression in breast cancers across subtypes. While the data are stratified by subtype in Extended Data, correlations by grade and LN status are not stratified by subtype. Are these correlations with grade and LN status stronger/more significant with the Luminal subtype, as data in Extended Data Figure 5 might suggest? It is also unclear which subtypes are represented in serum hRNase 1 data in E and in TMA data in D. Trends in cell lines screened in F don't appear to track with clinical data in Extended Figure 5, with relatively high levels of hRNase 1 expression in HER2+ lines (BT-474 – luminal but also HER2+, KPL4) and TNBC lines (BT-549, BT20, MDA-MB-468) that are on par or higher than levels in Luminal line T47D. Do other Luminal lines (MCF7, ZR751) express high/higher levels of hRNase 1 as the patient data from Extended Figure 5 would predict?
4. Figure 2 - BT-549 and KPL4 are TNBC lines. As hRNase 1 expression only significantly correlates with poor outcomes in human Luminal breast cancers and not TNBC, it is unclear why this cell line was chosen and how physiologically relevant the data are. Why not use Luminal line T47D, which is more relevant and expresses higher levels of hRNase 1 than BT-549 (Figure 1F)?
5. Figure 3 – F PLA data difficult to interpret with small panels. Were PLA data quantified?
6. Figure 4 – G PLA data very difficult to interpret in these small panels.
7. Figure 6 – see comments above: Why use KPL4 and BT-549 rather than luminal lines? Does EphA4 have the same impact on hRNase 1-mediated tumorigenesis in luminal lines? What are endogenous levels of mRNase 1 in 4T1 cells (also model of TNBC rather than Luminal breast cancer)?
8. Extended Data Figure 3 – What BC subtypes are represented in patient sera samples? Luminal A/B?
9. Extended Data Figure 5 – BT-549 was derived from TNBC. As hRNase 1 expression only significantly correlates with poor outcomes in human Luminal breast cancers and not TNBC, it is unclear why this cell line was chosen and how physiologically relevant the data are.
10. Extended Data Figure 5F – PLA signal weak and unconvincing in HeLa model
11. Extended Data Figure 6 – What does the partial band in the deltaLBD lane represent?
12. Extended Data Figure 7 – See comment on Extended Data Figure 5 – why use BT-549 TNBC cell line instead of a luminal breast cancer cell line for analysis? The signaling data are relatively weak, particularly in the model of ectopic hRNase 1 expression in BT-549 cells, which appear to express abundant endogenous protein. Why not stimulate cells with hRNase 1 and look at NF-kB nuclear localization and promoter binding, Erk, Src, and Akt phosphorylation? For D, it seems very odd that pharmacologic inhibitors have no apparent effect on spheroid formation for control BT-549, which should be dependent upon these driver signaling pathways for growth.
13. Extended Data Figure 8 – Same concern with use of TNBC line BT549. KPL-4 is a HER2+ cell line. As hRNase 1 expression only significantly correlates with poor outcomes in human Luminal breast cancers and not HER2+ breast cancers, it is unclear why this cell line was chosen and how physiologically relevant the data are.
14. Extended Data Figure 9 – It would help to show representative images from TMAs to assess staining specificity and localization of EphA4 and hRNase 1.

Reviewer #3 (Remarks to the Author):

The manuscript describes for the first time the interaction of human RNase1 with tyrosine kinase receptor ephrin A4 (EphA4). These are very novel and interesting results with potential applicability to cancer therapy. The paper includes both basic science and translational work to explore the potential of human RNase1 as a prognosis marker and target for breast cancer chemotherapy. Both approaches are very novel and results are of particular interest for the applied medicine field. The authors have proven the hRNase1-receptor interaction and activation by many complementary techniques, including overexpression, silencing and knockout cell assays, together with mutagenesis experiments. Conclusions are well supported by the experimental data. The manuscript is of general interest to Nature Communications readers. However, before publication the presentation of the data needs a bit more polishing. Mainly, due to the high amount of data and great variety of methodologies applied, figure legends must be as self-explanatory as possible. Besides, discussion is a bit poor and some of the presented results are not discussed and contrasted respect to previous available information.

General comments:

- 1- Due to the high amount of data and distinct methodologies applied, it is necessary to assist the reader and provide essential information in figure legends. Specify for each panel the information on timing, protein concentration, cell lines and antibodies to facilitate the results interpretation and evaluate its significance.
- 2- Terminology should be very carefully chosen. It is not correct to say that RNase1 function is to activate tumorigenesis. It could be hypothesized that the protein is involved in tissue regeneration, remodeling, or embryonic development and its overabundance can promote cancer, as a dysregulation disease process. It is important to differentiate the protein physiological putative role in health and disease states.
- 3- Recent work on RNase1 should be cited in the introduction and /or discussion.
- 4- Selection of cell lines and methodological approaches used is not always justified in the text. Likewise, the criteria for the selection of protein and receptor mutants must be indicated.
- 5- Information on the RNase1 recombinant proteins used should be detailed. Glycosylated forms should be described before showing any related results (MW, etc..). Were expressed proteins from HEK cells quantified? When lysates from HEK cells are used, it should be necessary to provide an estimation of the expressed level of proteins. In the methodology section it is important to detail the protocol of characterization of glycosylated and non-glycosylated samples from HEK cell lysate.
- 6- Which is the protocol used in the case of prokaryote expression? Is the purity checked? Is the used recombinant protein properly processed, without its signal peptide? Check for H40A mutant nomenclature (H12A?). Reference to H40 is not correct. A detailed methodology here is crucial as it can greatly influence the results.
- 7- Methodology section should try to indicate always clearly all volumes and concentrations used. Also, try to avoid telegraphic figure legends. Sometimes the information required to reproduce the experiments is missing.
- 8- Choice of cell lines should be justified for each experiment. Also, analysis of data should take care whether the selected cell line or methodology cannot bias the results interpretations. Why so many data are done with transfected cell lines and not with the edited cell lines?

Specific comments:

TITLE: it might be better to substitute "Eph receptor A4" with "ephrin A4 receptor"

ABSTRACT:

revise sentence "Here, we demonstrate that hRNase 1, independently OF its ribonucleolytic activity..."

"The discovery of hRNase 1 as a secretory ligand of EphA4 to enhance breast cancer stemness". Substitute TO, with "that enhances" or another phrasal expression. The physiological role of rnase1 should not be to promote cancer cells.

INTRODUCTION

Page 2, line 3 from bottom: Apart from ref 1, add recent reference on RNase presence in biological fluids (PMID: 29867984)

Page 2, last line. Add together with refs 4 and 5: PMID: 32544330

Page 3, line 4: add ref PMID: 29867984 after "innate immunity"

Page 3, line 5: together with refs 7 and 8: add references PMID: 24201302 and PMID: 25354936
RESULTS

Page 5, line 4. Subtitle and section should be reorganized as not all the data reported here is related to the contribution of the catalytic activity. Maybe the section can be subdivided in two parts with their respective subtitles and specific conclusions.

Page 5: it is important to indicate first the rationale for the choice of the mutation to remove the protein catalytic activity.

Be careful. Which is residue H40? Are the authors counting the signal peptide? Does it refer to His12? It does not make sense to number the residues keeping the immature protein. It should be confirmed that the expression protocol to obtain recombinant protein achieves a proper signal peptide cleavage?

Also indicate that mutation is not altering the protein 3D structure, by including either the own data (for example a circular dichroism) or reference to previous work on H12A.

Page 5, line 10: before commenting on results of glycosylated samples, they should be introduced (number of glycosylated forms, MW, etc...), together with reference to previous available information. The vector used here for expression of wt and mutant should be indicated and justified.

Page 6, line 1. Indicate in text and figure legends the antibodies naming, referred to residue of tyrosine phosphorylation.

Page 6, line 7: indicate protocols and references for expression of recombinant proteins.

Include more details here and in methodology about glycosidase treatment and quantities of recombinant protein added.

Justify selection of cell lines for each assay, every time a new cell type is used.

Page 7, line 1: DuoLink results (Figure 3f) should be explained a bit more. As RNase5 control is included, text must also include the specific drawn conclusions for both proteins.

Page 7, 3 lines from bottom: justify residues selection for mutant construct.

Page 8, line 7: provide more information, maybe a reference specifically talking on receptor formation of clusters.

Page 8: the term ectopic expression might not be the most accurate to make reference of the presence of secretory protein in the media.

As RNase1 is a soluble protein not anchored to the membrane, to what extent does it make sense to evaluate the potential juxtacrine signaling mechanism? Is the term only referencing to an action mediated by ephrins, which are working by a juxtacrine mechanism? This section might need some more justification.

Page 9, line 7: why most of the assays were performed with recombinant proteins or clones overexpressing RNase1 instead of analyzing directly the KO cell line?

Page 10, line 2. Explain first why the soft agar methodology is selected.

Page 10, line 14. Figures 6 j,k Significance is not very good. It would have been important to have more data and more conclusive results.

FIGURES:

Before figure 2a it is necessary to include information on glycosylated forms, as shown in figure 3, with MW, and digestion analysis to identify glycosylated and non-glycosylated samples, etc...

Figure 2. it is necessary to include the quantities of protein added, specify better the antibodies used. Stars for glycosylated and non-glycosylated are only indicated later in figure 3.

Figure 3a: screening of phosphorylated kinases seem to reveal some other positive hits apart from ephrin receptor, egfr and Tie-2. Detail information for ABCDEF in figure legend. Likewise, explain better in figure legend and text results the screening shown in extended figure 5a.

Figure 3 b panel, indicate RNase1 protein concentration added and from where it comes from (HEK; E. coli...).

Figure 3, panels I,j,k should be shown before figure 2.

Figure 2k, ref. 27. Better add the methodology in the present manuscript instead of merely referencing to ref.27. Also, the reference here is misleading, as it is unclear whether it includes related results or only the methodology description.

Figure 4: better insert panel b before panel a, to describe the chimera used. Specify the two domains of the receptor and indicate that K653A refers to the receptor phosphorylation site.

Figure 5 d,e. Indicate what is exactly measured by optical density.

Figure 6: panel b: nomenclature R1-2 can be misleading and is not explained elsewhere. Panels i,j,k. Better order them by alphabetical order within the figure. Figure 6k is incredibly too small.

Also, indicate more clearly data from knock down cell line, silencing protocol,.. In panel 6j, indicate

each pair analyzed for significance. Why a significance cut-off of $p \leq 0.05$ was not applied? Panels g and h: explain more clearly significance test, mark significant data. Why so many decimals are included?

Extended figure 1: it could be worth to include in the same panel the results for hRNase1, to facilitate a side-by-side direct comparison. What happened with RNases 8, 9 and 10? Indicate in the text why information is not provided for these RNases.

Extended figure 2. Some results seem also significant but the authors are not commenting them. Why differences observed in panel 2d are not mentioned. Is breast cancer the single significant case? How far selection is justified.

Extended figure 4 b. Indicate the number of assays for kinetic measurements. Volumes used for the assay. Graphic values are the average of how many replicates? It is also surprising that no activity is detected in the cell media from HEK cells, as other RNases, such as RNase2, would certainly be expressed in the kidney cell line.

Extended figure 7. Indicate significance in panel 7d.

Extended figure 8 panel a: what is KO control exactly?

DISCUSSION,

first sentence: "showing an unconventional function of hRNase 1 as a secretory ligand of EphA4 to promote breast cancer progression". It is very important to rephrase this sentence. The word "function" here is not correct.

Discussion, line 4: Care must be taken when using terms autocrine/paracrine. Here it might be better not to specify as the only thing we know for certain is that action is not juxtacrine.

Page 12, first line: even if we do not have a cocrystal, we can infer some conclusions from an overall structural comparative analysis. It would certainly be interesting to know if there is any structural similarity between RNases and Ephrins.

Page 12, line 18: Discussion should include reference to results from glycosylated form and recent work in the field (PMID: 30633504; PMID: 32544330).

Page 12, line 4 from bottom: it could be worth to comment on the protein hyperglycosylation observed in some cancer cell lines, such as reported for RNase2 (PMID: 7616105).

Page 12, line 2 from bottom: change "function" with "role".

The authors conclude that glycosylations are not altering the protein interaction with the receptor. They discard an enhancement of activity, but the possibility should be also considered that the glycosylation might cover the receptor binding site and reduce the interaction.

Discussion is a bit poor and too short. It could be worth to comment recent data on RNase 5 and egfr receptor, discuss the results from the screening using the phosphoreceptors kit, the regions identified to be involved in interaction by site directed mutagenesis of protein and receptor, etc.

METHODS

Page 35, elisa. "briefly the plates were precoated with Elisa plates are prepared manually or they are provided with the kit? Specify otherwise the antibodies used.

Page 38: indicate the antibodies against phosphorylated receptors that nomenclature is referred to tyr residue,...

Point-by-Point Response to the Reviewers' Comments
NCOMMS-20-19685-T

Reviewer #1 (Remarks to the Author):

The authors showed that hRNase I binds to the extracellular domain of EphA4 and activates EphA4, leading to maintenance of breast cancer stem-like cells. It is interesting that hRNase I has unexpected roles for cancer stem-like cells. However, it is still unclear the physiological and pathological relevance of hRNase I for EphA4-signaling, since its native ligand ephrin is able to stimulate EphA4 in the tumor microenvironment and the affinity of EphA4 for ephrin seems to be much higher than for hRNase I (Fig. 5b, c). It is also very important to demonstrate the findings by using patient-derived cancer cells.

Specific points:

Fig. 1

It is important to show the Kaplan-Meier survival curves by using several cohorts and to use the median values for patient stratification. It is not appropriate to use Prognoscan for the analysis, since it does not use the median values for patient stratification. It is also helpful to show number of patients for each population belonging to hRNase I high and low.

Fig. 2b, Extended Fig. 8e

They performed soft agar colony assays. Though it is not very clear, they probably added serum into the media. This assay is useful for evaluation of anchorage-independent cell growth but not of stem-like properties.

Fig. 2j,k

It is important to show the results by using ELDA.

Fig. 5e

What are KYL peptides?

Fig. 7a

They should describe the levels of hRNase I in plasma in each population.

Authors' Response: We appreciate the reviewer's thoughtful assessment and constructive criticisms to further strengthen our findings and improve the scientific merits of this manuscript. We have carefully addressed each of the concerns according to the reviewer's suggestions. Before proceeding with the specific points in a point-by-point manner, we would like to respond to the general comments mentioned above as follows:

First, we agree with the reviewer that *"its native ligand ephrin is able to stimulate EphA4 in the tumor microenvironment and the affinity of EphA4 for ephrin seems to be much higher than for hRNase 1 (Fig. 5b, c)."* However, compared with classical ephrin ligands, hRNase 1 harbors unique characteristics serving as a ligand of EphA4, which we apologize that we did not make it clear enough in the original manuscript. hRNase 1 is a natively secretory protein freely circulating in various body fluids, e.g., serum and plasma, thus it activates EphA4 in an autocrine/paracrine manner, whereas classical ephrin ligands are membrane-bound proteins to interact with EphA4 in a cell-cell contact and juxtacrine manner. Notably, the levels of serum/plasma hRNase 1 were determined at a concentration ranging from 0.1 to 1.5 µg/ml in patients with different cancer types (*Biochem J*, 474, 2219-2233, 2017; *Cancer Cell*, 33, 752-769, 2018; current study), which

belongs to a group of plasma proteins with medium-to-high abundance (medium abundance: 0.1–1 µg/ml; high abundance: >1 µg/ml) (*Proteomics*, 5, 3226-3245, 2005; *J Proteome Res*, 9, 4982-4991, 2010; *J Proteome Res*, 18, 4085-4097, 2019). The above features of hRNase 1 promoted us to speculate that the relatively high concentration of serum/plasma hRNase 1 in the tumor microenvironment may predominantly contribute to EphA4 activation via an autocrine or paracrine pathway under low cell density in early stage of tumorigenesis. This hypothesis regarding a role of hRNase 1 in the tumor microenvironment warrants further investigation in the future.

Second, to support the notion that low binding affinity of ligand-receptor pairs, such as hRNase 1-EphA4, can lead to significant physiological and pathological functionality in cancers, we provided several examples in different ligand-receptor studies: **(1)** ephrin-B2 ligand plays a crucial role in neural development through EphA4 as an axonal guidance receptor (*J Comp Neurol*, 462, 90-100, 2003; *Sci Signal*, 1, re2, 2008), although *in vitro* binding affinity of ephrin-B2 toward EphA4 is 5–300 times weaker than class A ephrins interacting with EphA4 (*Structure*, 17, 1386-1397, 2009); **(2)** two low-affinity EGFR ligands, epiregulin and epigen, induce weaker receptor dimerization than EGF (a high-affinity EGFR ligand) and result in sustained EGFR activating signaling, leading to cell differentiation rather than EGF-mediated proliferation (*Cell* 171, 683-695, 2017); **(3)** another human RNase called hRNase 5 (angiogenin/ANG) acts as an EGFR ligand and a serum biomarker to predict EGFR inhibitor response in pancreatic cancer, but EGF fails to have such predictive function although the binding affinity of hRNase 5 to EGFR is about 23-fold less than EGF for EGFR binding (*Cancer Cell*, 33, 752-769, 2018). These accumulating examples indicate that ligand-receptor interaction with low binding affinity can still contribute to significant functionality.

Taken together with our findings, secretory hRNase 1 could play a prominent role in the blood circulatory system to activate EphA4 signaling, which is distinct from classical ephrin ligands dependent on cell-cell contact. We have added more description in the Discussion section of the revised manuscript as shown below.

“Notably, hRNase 1 differs from classical ephrin ligands in that hRNase 1 is a natively secretory protein freely circulating in various body fluids, e.g., serum and plasma, thus it activates EphA4 in an autocrine/paracrine manner, whereas classical ephrin ligands are membrane-bound proteins to interact with EphA4 in a cell-cell contact and juxtacrine manner. The levels of serum/plasma hRNase 1 were determined at a concentration ranging from 0.1 to 1.5 µg/ml in patients with different cancer types (*Biochem J*, 474, 2219-2233, 2017; *Cancer Cell*, 33, 752-769, 2018; current study), which belongs to a group of plasma proteins with medium-to-high abundance (medium abundance: 0.1–1 µg/ml; high abundance: >1 µg/ml) (*Proteomics*, 5, 3226-3245, 2005; *J Proteome Res*, 9, 4982-4991, 2010; *J Proteome Res*, 18, 4085-4097, 2019). The relatively high concentration of serum/plasma hRNase 1 in the tumor microenvironment may predominantly contribute to EphA4 activation via an autocrine or paracrine pathway under low cell density in early stage of tumorigenesis.”

“The binding affinity of hRNase 1 to EphA4 ($K_d = 92.4$ nM) was about 16.5-fold less than that of ephrin-A5 to EphA4 ($K_d = 5.6$ nM) (Fig. 5b, c), but accumulating evidence indicates that ligand-receptor interaction with low binding affinity can still contribute to significant functionality. For example, ephrin-B2 ligand plays a crucial role in neural development through EphA4 as an axonal guidance receptor (*J Comp Neurol*, 462, 90-100, 2003; *Sci Signal*, 1, re2, 2008), although *in vitro* binding affinity of ephrin-B2 toward EphA4 is 5–300 times weaker than class A ephrins interacting with EphA4 (*Structure*, 17, 1386-1397, 2009). Moreover, two low-affinity EGFR ligands, epiregulin and epigen, induce weaker receptor dimerization than EGF (a high-affinity EGFR ligand) and result in sustained EGFR activating signaling, leading to cell differentiation rather than EGF-mediated proliferation (*Cell* 171, 683-695, 2017). Another human RNase called hRNase 5 acts as an EGFR ligand and a serum biomarker to predict EGFR inhibitor response in pancreatic cancer, but EGF fails to have such predictive function although the binding affinity of hRNase 5 to EGFR is about 23-fold less than EGF for EGFR binding (*Cancer Cell*, 33, 752-769, 2018).”

And **third**, we thank the reviewer for the comment that “*It is also very important to demonstrate the findings by using patient-derived cancer cells.*”, and we would be willing to do so. However, the processing time for paperwork to obtain those cells would exceed the time frame expected to submit our revised manuscript. To respond to this comment, we alternatively provided several sets of analysis using breast cancer patient samples to demonstrate the pathological relevance of hRNase 1 EphA4 axis (Figs. 7a–7e of the original and revised versions; Fig. S9 of the original manuscript; Fig. S13a and **New Fig. S13b** of the revised manuscript). In brief, high hRNase 1 levels in plasma samples were positively associated with EphA4 activation in the paired tumor tissues from breast cancer patients (Original Fig. 7a). In addition, analysis of human breast tumor tissue microarray (TMA) revealed a positive correlation between hRNase 1, EphA4 activation, and a stem cell marker CD133 (Figs. 7b–7e of the original and revised versions). Another independent TMA analysis of human breast cancer cohort further supported the positive association between hRNase 1 and phospho-EphA4 expression (Fig. S9 of the original manuscript; Fig. S13a and **New Fig. S13b** of the revised manuscript). Together, these results strongly highlight the pathological relevance of hRNase 1 for stimulating EphA4 signaling in breast cancer.

Original Fig. 7a

		Tissue phospho-EphA4			Total
		Low	Medium	High	
Plasma hRNase 1	Low	8	7	5	20
	Medium	33	8	6	47
	High	7	4	7	18
	Total	48	19	18	85

p = 0.042

Original Fig. 7e

Original Fig. 7b

		phospho-EphA4		
		Low	High	Total
hRNase 1	Low	25	18	43
	High	9	39	48
	Total	34	57	91

p = 0.0002

Original Fig. S9 (Now Fig. S13a)

		phospho-EphA4		
		Low	High	Total
hRNase 1	Low	16	8	24
	High	4	20	24
	Total	20	28	48

p = 0.0001

Original Fig. 7c

		CD133		
		Low	High	Total
hRNase 1	Low	31	12	43
	High	23	25	48
	Total	54	37	91

p = 0.032

New Fig. S13b

Original Fig. 7d

		phospho-EphA4		
		Low	High	Total
CD133	Low	25	29	54
	High	9	28	37
	Total	34	57	91

p = 0.047

Point #1: *Fig. 1. It is important to show the Kaplan-Meier survival curves by using several cohorts and to use the median values for patient stratification. It is not appropriate to use Prognoscan for the analysis, since it does not use the median values for patient stratification. It is also helpful to show number of patients for each population belonging to hRNase I high and low.*

Response to Point #1: We thank the reviewer for the comments. In the original manuscript, we analyzed the data of Kaplan-Meier survival curves in the Figs. 1a–1c, S1a, S1b, S2a, and S2e from the Kaplan-Meier plotter database (<http://kmplot.com/analysis/>). In addition, the data of Original Fig, S2c was also analyzed the Kaplan-Meier survival curve from the UCSC Cancer Genome Browser (<http://xena.ucsc.edu/welcome-to-ucsc-xena/>) using the interpreted expression profile of TCGA breast invasive carcinoma by RNA sequencing (dataset ID: TCGA_BRCA_exp_HiSeqV2). For the original data from the Prognoscan database (Original Figs. S2b and S2d), we have replaced them with **New Figs. S3a and S3c**, respectively, assessed by Kaplan-Meier plotter analysis. In **New Fig. S3a**, we analyzed breast cancer survival curves with hRNase 1 expression by pan-cancer module of Kaplan-Meier plotter in which the gene expression was detected by RNA sequencing. This new data shown high hRNase 1 expression associated with poor overall survival of breast cancer patients was in line with the original data analyzed by Prognoscan (Original Fig. S2b). We further analyzed the prognostic correlation of patient survival with hRNase 1 expression in different cancer types by pan-cancer module of Kaplan-Meier plotter, and similar results were observed in patients with liver hepatocellular carcinoma and esophageal squamous cell carcinoma (**New Fig. S3c**). All of the results in the revised version were analyzed by using the median values for patient stratification. We have deleted the description of Prognoscan analysis and replaced with the modified sentences under “*Prognostic analysis of cancer patients from databases*” in the Methods section of the revised manuscript as shown below. Moreover, we have followed the reviewer’s suggestion to add the number of patients for each population with high or low hRNase 1 levels in the revised manuscript.

“The Kaplan-Meier plotter database (<http://kmplot.com/analysis/>) was used to analyze the correlation between the expression levels of hRNase family and survival of cancer patients. The UCSC Cancer Genome Browser (<http://xena.ucsc.edu/welcome-to-ucsc-xena/>) was utilized to validate prognostic effect of hRNase 1 with Kaplan-Meier survival analysis. In brief, survival analysis was performed using the interpreted expression profile of TCGA breast invasive carcinoma by RNA sequencing (dataset ID: TCGA_BRCA_exp_HiSeqV2) downloaded from the UCSC Cancer Genome Browser. The median expression of hRNase 1 was used for patient stratification. A corrected p-value < 0.05 was considered as significant.”

Original Fig. 1a

Original Fig. 1b

Original Fig. 1c

Original Fig. S1a

Original Fig. S1b

Original Fig. S2a

New Fig. S3a

Original Fig. S2c (Now Fig. S3b)

New Fig. S3c

Original Fig. S2e (Now Fig. S3d)

Point #2: Fig. 2b Extended Fig. 8e. They performed soft agar colony assays. Though it is not very clear, they probably added serum into the media. This assay is useful for evaluation of anchorage-independent cell growth but not of stem-like properties.

Response to Point #2: As the reviewer suspected, we mixed agar solutions with complete media containing 10% FBS. We performed the soft agar colony formation assay in Original Figs. 2b and S8e (Fig. S11e of the revised version) in order to study whether the hRNase 1-EphA4 axis harbored *in vitro* oncogenic transformation potential. We have added the relevant explanation on **pages 7 and 15**. We agree that this assay is useful for evaluation of anchorage-independent cell growth but not of stem-like properties. To avoid confusion, we have moved the Original Fig. 2b to the supplement Fig. S5f in the revised version. In addition, we greatly appreciate the reviewer's comments that led us to conduct several new experiments to examine stem-like properties by using sphere formation assay and flow cytometry analysis of CD24/CD44 expression in multiple luminal type breast cancer cells, including luminal A (MCF7, T-47D, and ZR-75-1) and luminal B (BT-474), to further strengthen our findings of hRNase 1-EphA4 axis in the stem-like cell model of breast cancer. The details are described below.

First, we ectopically expressed hRNase 1 in MCF7 cells (**New Fig. S5a**) and found that hRNase 1 significantly increased the number of sphere formation in primary, secondary, and tertiary passages (**New Figs. 2a and 2b**). Moreover, hRNase 1 enriched the population of CD24⁻/CD44⁺ cells in MCF7 parental cells and formed spheres (**New Figs. 2c and 2d**). These results demonstrated that the elevated hRNase 1 increased the sphere-forming ability and the population of CD24⁻/CD44⁺ cells in MCF7 luminal A breast cancer cells. **Second**, an inactivated form of hRNase 1 (catalytic-deficient R1-H12A mutant) and its wild-type counterpart (R1) were ectopically expressed in T-47D cells (**New Figs. S5g–S5i**). Either R1 or R1-H12A increased the number and size of spheres in T-47D cells (**New Figs. S5j–S5l**), indicating that the ribonucleolytic activity-independent function of hRNase 1 increased both the number and size of spheres in T-47D luminal A breast cancer cells. Together, consistent with the original findings demonstrated in basal-like BT-549 and HER2+ KPL4 breast cancer cells, the stemness-promoting feature of hRNase 1 also occurs in luminal type breast cancer cells independently of its ribonucleolytic activity. **Last**, we found that EphA4 positively regulated hRNase 1-mediated breast cancer stem-like properties in both luminal A and B subtypes of breast cancer cells. In brief, silencing EphA4 by shRNA knockdown in hRNase 1-expressing MCF7 cells (**New Fig. S11a**) attenuated the hRNase 1-induced sphere-forming frequency (**New Figs. 6g and S11b**) and stem-like cell population (**New Figs. 6h and 6i**). Likewise, ectopic expression of EphA4 in ZR-75-1 luminal A breast cancer cells (**New Fig. S10a**) upregulated sphere formation (**New Figs. 6a and S10b**) and enriched the population of stem-like cells (**New Figs. 6b and 6c**), whereas knockdown of hRNase 1 abrogated those effects. We observed similar results in BT-474 luminal B breast cancer cells (**New Fig. S10c**) from sphere formation assay (**New Figs. S10d and S10e**) and flow cytometry analysis of CD24⁻/CD44⁺ expression (**New Figs. S10f and S10g**).

With these new results, we further strengthened our initial observations and provided more insights into the biological significance of hRNase 1-EphA4 axis to provoke the stem-like cell properties among the four major molecular subtypes of breast cancer, including basal-like (BT-549), HER2+ (KPL4), luminal A (MCF7, T-47D, and ZR-75-1), and luminal B (BT-474).

New Fig. S5a

New Fig. 2c

New Fig. 2a

New Fig. 2b

New Fig. 2d

New Fig. S5g

New Fig. S5h

New Fig. S5i

New Fig. S5j

New Fig. S5k

New Fig. S5l

New Fig. S11a

New Fig. 6h

New Fig. S11b

New Fig. 6i

New Fig. 6g

New Fig. S10a

New Fig. S10b

New Fig. 6a

New Fig. 6b

New Fig. 6c

New Fig. S10c

New Fig. S10d

New Fig. S10e

New Fig. S10f

New Fig. S10g

Point #3: Fig. 2j,k. It is important to show the results by using ELDA.

Response to Point #3: We thank the reviewer for the suggestion. We have estimated the tumor-initiating cell (TIC) frequency using ELDA software (<http://bioinf.wehi.edu.au/software/elda/>) in the revised manuscript. In Original Fig. 2j (Fig. 2g of the revised manuscript), the TIC frequencies were 1/347,606 for BT-549-Ctrl, 1/144,270 for BT-549-R1, and 1/101,955 for BT-549-R1-H12A. In Original Fig. 2k (Fig. 2j of the revised version), the TIC frequencies were 1/5,581 for KPL4-sh-Ctrl and 1/24,663 for KPL4-sh-R1#2.

Modified Fig. 2j (Now Fig. 2g)

Modified Fig. 2k (Now Fig. 2j)

Point #4: Fig. 5e. What are KYL peptides?

Response to Point #4: The KYL peptide identified in 2003 by screening a phage display library is a linear 12 amino-acid peptide (KYLPYWPVLSSL) that selectively binds to EphA4 but not other Eph receptors (*Mol Cell Neurosci.* 24, 1000-1011, 2003). Researchers have characterized the KYL peptide as an EphA4 kinase inhibitor that occupies the ephrin-binding pocket of EphA4 ($K_d = 0.8 \pm 0.15 \mu\text{M}$ measured by isothermal titration calorimetry) and inhibits ephrin-A5 interaction with EphA4 ($IC_{50} = 6.34 \mu\text{M}$ in ELISA) (*Mol Cell Neurosci.* 24, 1000-1011, 2003; *Biochem J.* 445, 47-56, 2012). The KYL-P7A peptide is a modified KYL peptide by alanine substitution (Pro7 to Ala) serving as a negative control, which completely loses the binding affinity to EphA4 (*Biochem J.* 445, 47-56, 2012). We have provided a more clear statement to introduce the background of KYL and KYL-P7A peptides in the Results section of the revised manuscript (bold texts shown below).

“Those results were further validated by treating cells with the KYL peptide acting as an EphA4 kinase inhibitor, which is a linear 12 amino-acid peptide (KYLPYWPVLSSL) that selectively binds to EphA4 and inhibits EphA4-ephrin-A5 interaction (*Mol Cell Neurosci.* 24, 1000-1011, 2003; *Biochem J.* 445, 47-56, 2012). The interaction of hRNase 1 and EphA4 was significantly but not completely reduced in the presence of KYL compared with the KYL-P7A mutant control, known to completely lose its EphA4 binding activity (*Biochem J.* 445, 47-56, 2012) (Fig. 5e). These results indicated a partial overlap between the hRNase 1 and ephrin A5 binding sites on the EphA4 receptor and supported the proposed role of hRNase 1 as an EphA4 ligand.”

Original Fig. 5e

Point #5: Fig. 7a. They should describe the levels of hRNase 1 in plasma in each population.

Response to Point #5: We thank the reviewer for the suggestion. We have described the levels of hRNase 1 in plasma in each population under “Scoring of the paired human plasma and tissue samples” in the Methods section of the revised manuscript.

“The paired human plasma and tissue samples were obtained from 85 patients with breast cancer. The range of plasma hRNase 1 concentration was from 67 to 801 ng/ml. For patient stratification in this experiment, samples of plasma hRNase 1 with the concentration value below 25% percentile (< 85 ng/ml) were regarded as low expression, above 75% percentile (≥ 223 ng/ml) as high expression, and 25–75% percentile as medium expression. Tissue samples with the staining intensity scored as 0 and 1 were regarded as low expression, 2 as medium expression, and 3 as high expression.”

Reviewer #2 (Remarks to the Author):

*This manuscript by Lee et al. describes a novel interaction between human ribonuclease 1 (hRNase 1) and EphA4 receptor tyrosine kinase. Serving as a novel ligand for EphA4, hRNase 1 activation of EphA4 promote breast tumor initiation in a paracrine manner that is distinct from canonical signaling through activation by membrane-bound ephrin-A ligands. The authors also report correlations between high hRNase 1 plasma levels and EphA4 activation in tumor tissue for breast cancer patients. The work is novel and of broad interest to scientists and clinicians in cancer research. However, there were several concerns. **First:** One major concern was about the choice of cell line models for experimental analyses that do not reflect breast cancer subtypes in which correlations between hRNase 1 and outcomes were found (i.e. use of TNBC models when correlations were found for patients with Luminal disease). **Second:** Another major concern was the apparent lack of molecular subtyping for patients in which hRNase 1 levels in sera were analyzed. **Third:** The use of ectopic expression in cell lines that already express high levels of hRNase 1 was also a concern for signaling studies, as was the apparent lack of BT-549 control cell*

responsiveness to pharmacologic inhibitors of NF-kB and Mek. **Fourth:** Minor issues included difficulty evaluating PLA data from small panels. It is recommended that the concerns outlined below be addressed prior to publication, which should significantly strengthen the story and impact of the paper:

1. Introduction: “Currently, there are no known cognate secretory ligands for the Eph receptors.” This is not quite correct. Soluble ephrin-A ligands have been detected in the culture media of cell lines in models of breast and other cancers, where they activate EphA receptors and downstream signaling and cellular responses [Alford et al. 2007 *Exp Cell Res* 313: 4170-4179; Wykosky et al. 2008 *Oncogene* 27: 7260-73; Alford et al. 2010 *Cancer Cell Int.* 10: 41; Song et al. 2013 *PLoS One* 8: e74464]. Ephrin-A1 can be cleaved from the cell surface by metalloproteases [Ieguchi et al. 2013 *Biochem Biophys Res Commun* 440: 623-9; Ieguchi et al. 2014 *Oncogene* 33: 2179-90].
2. A transition paragraph between the end of the Introduction and the beginning of the Results sections would be helpful, with a few sentences that tie hRNase 1 to EphA4.
3. Figure 1 – Correlations between survival and hRNase 1 expression in breast cancers across subtypes. While the data are stratified by subtype in Extended Data, correlations by grade and LN status are not stratified by subtype. Are these correlations with grade and LN status stronger/more significant with the Luminal subtype, as data in Extended Data Figure 5 might suggest? It is also unclear which subtypes are represented in serum hRNase 1 data in E and in TMA data in D. Trends in cell lines screened in F don’t appear to track with clinical data in Extended Figure 5, with relatively high levels of hRNase 1 expression in HER2+ lines (BT-474 – luminal but also HER2+, KPL4) and TNBC lines (BT-549, BT20, MDA-MB-468) that are on par or higher than levels in Luminal line T47D. Do other Luminal lines (MCF7, ZR751) express high/higher levels of hRNase 1 as the patient data from Extended Figure 5 would predict?
4. Figure 2 - BT-549 and KPL4 are TNBC lines. As hRNase 1 expression only significantly correlates with poor outcomes in human Luminal breast cancers and not TNBC, it is unclear why this cell line was chosen and how physiologically relevant the data are. Why not use Luminal line T47D, which is more relevant and expresses higher levels of hRNase 1 than BT-549 (Figure 1F)?
5. Figure 3 – F PLA data difficult to interpret with small panels. Were PLA data quantified?
6. Figure 4 – G PLA data very difficult to interpret in these small panels.
7. Figure 6 – see comments above: Why use KPL4 and BT-549 rather than luminal lines? Does EphA4 have the same impact on hRNase 1-mediated tumorigenesis in luminal lines? What are endogenous levels of mRNase 1 in 4T1 cells (also model of TNBC rather than Luminal breast cancer)?
8. Extended Data Figure 3 – What BC subtypes are represented in patient sera samples? Luminal A/B?
9. Extended Data Figure 5 – BT-549 was derived from TNBC. As hRNase 1 expression only significantly correlates with poor outcomes in human Luminal breast cancers and not TNBC, it is unclear why this cell line was chosen and how physiologically relevant the data are.
10. Extended Data Figure 5F – PLA signal weak and unconvincing in HeLa model
11. Extended Data Figure 6 – What does the partial band in the deltaLBD lane represent?
12. Extended Data Figure 7 – See comment on Extended Data Figure 5 – why use BT-549 TNBC cell line instead of a luminal breast cancer cell line for analysis? The signaling data are relatively weak, particularly in the model of ectopic hRNase 1 expression in BT-549 cells, which appear to express abundant endogenous protein. Why not stimulate cells with hRNase 1 and look at NF-kB

nuclear localization and promoter binding, Erk, Src, and Akt phosphorylation? For D, it seems very odd that pharmacologic inhibitors have no apparent effect on spheroid formation for control BT-549, which should be dependent upon these driver signaling pathways for growth.

13. Extended Data Figure 8 – Same concern with use of TNBC line BT549. KPL-4 is a HER2+ cell line. As hRNase 1 expression only significantly correlates with poor outcomes in human Luminal breast cancers and not HER2+ breast cancers, it is unclear why this cell line was chosen and how physiologically relevant the data are.

14. Extended Data Figure 9 – It would help to show representative images from TMAs to assess staining specificity and localization of EphA4 and hRNase 1.

Authors' Response: We appreciate the reviewer's comment on the novelty of our study and that this work is of broad interest to scientists and clinicians in cancer research. The reviewer's thoughtful assessment and constructive criticisms have undoubtedly strengthened our findings and improved the scientific merits of this manuscript. We would like to first respond to the four above-mentioned concerns below and then answer the other comments in a point-by-point manner:

First: *One major concern was about the choice of cell line models for experimental analyses that do not reflect breast cancer subtypes in which correlations between hRNase 1 and outcomes were found (i.e. use of TNBC models when correlations were found for patients with Luminal disease).*

Response: We apologize that we might not clearly describe the correlation data in the original manuscript, which showed that, in addition to the luminal A and B subtypes with the p values being more significant, high hRNase 1 expression also significantly correlated with poor outcomes in patients with basal-like (TNBC) subtype (HR = 1.34; 95% CI, 1 to 1.79; p = 0.048). In the revised version, we have followed the reviewer's suggestions to conduct new experiments to examine the biological significance of hRNase 1-EphA4 axis in multiple luminal type breast cancer cells, including luminal A (MCF7, T-47D, and ZR-75-1) and luminal B (BT-474). The new results together with the original findings demonstrated that the hRNase 1-EphA4 axis provoked the stem-like cell properties among the four major molecular subtypes of breast cancer, including basal-like (BT-549), HER2+ (KPL4), luminal A (MCF7, T-47D, and ZR-75-1), and luminal B (BT-474). Please see the **Authors' Response to Points #4, #7, and #9** for a detailed description.

Second: *Another major concern was the apparent lack of molecular subtyping for patients in which hRNase 1 levels in sera were analyzed.*

Response: We have classified the molecular subtypes for patients in which hRNase 1 levels in sera and tissues were analyzed in the revised manuscript. Please see the **Authors' Response to Points #3 and #8**.

Third: *The use of ectopic expression in cell lines that already express high levels of hRNase 1 was also a concern for signaling studies, as was the apparent lack of BT-549 control cell responsiveness to pharmacologic inhibitors of NF- κ B and Mek.*

Response: There is miscommunication on ectopic expression of hRNase 1 in BT-549 cell line, which was characterized to express low levels of endogenous hRNase 1 in the conditioned medium (CM) by ELISA and Western blotting (WB) in the original manuscript. In brief, the original ELISA results showed that CM hRNase 1 from breast cancer cell lines examined including BT-549 were determined at a concentration lower than 15 ng/ml, except KPL4 cells carrying CM hRNase 1 up to 100 ng/ml. The original WB data also revealed the levels of CM hRNase 1 undetectable in BT-549 control cells expressing empty vector (BT-549-Ctrl). In the revised manuscript, we screened a panel of 11 breast cancer cell lines including BT-549 to measure the levels of CM hRNase 1 using WB, and further clarified that BT-549 cells exhibited relatively low levels of endogenous hRNase 1 among them. The details are described in the **Authors' Response to Points #3 and #12**.

The concern regarding *“the apparent lack of BT-549 control cell responsiveness to pharmacologic inhibitors of NF- κ B and Mek”* could be attributed to the usage of effective but relatively low concentrations (low nM to low μ M) of pharmacologic inhibitors of NF- κ B (QNZ; 30 nM) and Mek (GSK1120212 and PD0325901; 1 nM and 0.2 μ M, respectively), at which the spheroid formation of BT-549 control cells did not change significantly compared with the one in the absence of inhibitors. In the revised manuscript, we further validated the inhibition of NF- κ B and Mek signaling pathways under inhibitor treatment in such concentrations, and observed similar results in MCF7 luminal A breast cancer cells. Please see a detailed description in the **Authors' Response to Point #12**.

Fourth: *Minor issues included difficulty evaluating PLA data from small panels.*

Response: We thank reviewer for pointing this out. We have enlarged the Duolink PLA images and added the quantified results as suggested in the revised manuscript. Please see the **Authors' Response to Points #5, #6, and #10**.

Point #1: *Introduction: “Currently, there are no known cognate secretory ligands for the Eph receptors.” This is not quite correct. Soluble ephrin-A ligands have been detected in the culture media of cell lines in models of breast and other cancers, where they activate EphA receptors and downstream signaling and cellular responses [Alford et al. 2007 Exp Cell Res 313: 4170-4179; Wykosky et al. 2008 Oncogene 27: 7260-73; Alford et al. 2010 Cancer Cell Int. 10: 41; Song et al. 2013 PLoS One 8: e74464]. Ephrin-A1 can be cleaved from the cell surface by metalloproteases [Ieguchi et al. 2013 Biochem Biophys Res Commun 440: 623-9; Ieguchi et al. 2014 Oncogene 33: 2179-90].*

Response to Point #1: We thank the reviewer for the comment. To more accurately introduce the background knowledge regarding ligands of Eph receptors, we have added the information of soluble ephrin-A ligands and replaced with the modified sentences in the Introduction section of the revised manuscript (bold texts shown below).

“Eph receptors are traditionally thought to bind to their membrane-bound ligands, ephrins, on neighboring cells to stimulate juxtacrine signals through contact-dependent cell-cell communication²⁰. **Interestingly, soluble ephrin-A ligands derived from their**

membrane-bound forms have been detected in the culture media of cell lines in models of glioblastoma, breast and other cancers, where they activate EphA receptors and downstream signaling and cellular responses (Alford et al. 2007 *Exp Cell Res* 313: 4170-4179; Wykosky et al. 2008 *Oncogene* 27: 7260-73; Alford et al. 2010 *Cancer Cell Int.* 10: 41; Song et al. 2013 *PLoS One* 8: e74464). It is noteworthy that metalloprotease-mediated proteolytic shedding of membrane-anchored ephrins has been further demonstrated as one major mechanism to generate soluble ephrins in the extracellular space (Ieguchi et al. 2013 *Biochem Biophys Res Commun* 440: 623-9; Ieguchi et al. 2014 *Oncogene* 33: 2179-90).”

Point #2: *A transition paragraph between the end of the Introduction and the beginning of the Results sections would be helpful, with a few sentences that tie hRNase 1 to EphA4.*

Response to Point #2: We thank the reviewer for the comment. We have included a transition paragraph in the end of the Introduction section of the revised version (bold texts shown below).

“...In addition to the classical juxtacrine signaling, further investigations into the autocrine/paracrine signaling involved in EphA4 activation may provide a more comprehensive picture of Eph receptor regulation and shed light on potential therapeutic strategies against CSC-like cells by blocking EphA4 signaling.

RNases and RTKs, which are considered as two unrelated families, have been recently discovered the significance of ligand-receptor relationship in pancreatic and liver cancers (Wang et al. 2018 *Cancer Cell* 33: 752-69; Liu et al. 2020 *J Hepatol* S0168-8278(20)33674-6). In this report, we uncovered a ligand-receptor pair of hRNase 1-EphA4 critical for breast cancer initiation, namely that hRNase 1 acts as a natively secretory ligand of EphA4 to stimulate EphA4 signaling in an autocrine/paracrine manner, leading to upregulation of stem-like cell properties in breast cancer.”

Point #3: *Figure 1 – Correlations between survival and hRNase 1 expression in breast cancers across subtypes. While the data are stratified by subtype in Extended Data, correlations by grade and LN status are not stratified by subtype. Are these correlations with grade and LN status stronger/more significant with the Luminal subtype, as data in Extended Data Figure 2 might suggest? It is also unclear which subtypes are represented in serum hRNase 1 data in E and in TMA data in D. Trends in cell lines screened in F don’t appear to track with clinical data in Extended Figure 2, with relatively high levels of hRNase 1 expression in HER2+ lines (BT-474 – luminal but also HER2+, KPL4) and TNBC lines (BT-549, BT20, MDA-MB-468) that are on par or higher than levels in Luminal line T47D. Do other Luminal lines (MCF7, ZR751) express high/higher levels of hRNase 1 as the patient data from Extended Figure 2 would predict?*

Response to Point #3: We greatly appreciate the reviewer’s comments and suggestions. **First**, we revisited the survival data of tumors with different grades by subtype stratification (**New Fig. S2b**; grade 1 data of HER2+ not provided in the database). Indeed, among the four molecular subtypes, the luminal B subtype in each grade displayed a higher hazard ratio (HR > 1) and a smaller p value, suggesting a stronger/more significant correlation between high hRNase 1

expression and poor patient survival in the luminal B subtype in all grades, although there is a lack of statistical significance ($p = 0.16, 0.19, \text{ and } 0.12$ in grade 1, 2, and 3, respectively). **Second**, we reanalyzed the survival data of patients with LN-positive metastasis and LN-negative non-metastasis by subtype stratification (**New Fig. S2c**). We found that hRNase 1 appeared to be a more significant poor prognostic factor in LN-positive patients with the luminal A subtype (HR = 1.33; $p = 0.073$) and LN-negative patients with the luminal B subtype (HR = 1.3; $p = 0.086$). **Third**, we have classified the molecular subtypes of tissue hRNase 1 in the TMA IHC data (**New Fig. 1e** and Original Fig. 1d) and serum hRNase 1 in the ELISA data (**New Fig. 1g** and Original Fig. 1e; Fig. 1f in the revised version). Interestingly, the levels of either tissue or serum hRNase 1 had no statistically significant difference among the subtypes of breast cancer (ns, not significant, One-way ANOVA test). **Last**, to more clearly elucidate the ELISA data from the cell line conditioned medium (CM), we first reanalyzed it using a box and whisker plot and demonstrated that the average expression level of CM hRNase 1 in eight breast cancer cell lines was significantly higher than that in two normal breast cells (Original Fig. 1f; Fig. 1h in the revised version; KPL4 cell line as an outlier). Further analysis indicated that these eight breast cancer cell lines were determined at a concentration of CM hRNase 1 lower than 15 ng/ml, except KPL4 cells carrying CM hRNase 1 up to 100 ng/ml (Original Fig. 1f; Fig. S4f in the revised version). In the revised manuscript, we then performed Western blotting (WB) to screen a panel of 11 breast cancer cell lines, including basal-like (BT-549, MDA-MB-231, MDA-MB-468, and BT-20), luminal A (MCF7, T-47D, and ZR-75-1), luminal B (BT-474), and HER2+ (KPL4, SKBR3, and MDA-MB-453) to measure the levels of CM hRNase 1 (**New Fig. 1i**). The results showed that five cell lines across subtypes (MDA-MB-231, BT-20, ZR-75-1, BT-474, and KPL4) exhibited apparent expression of hRNase 1, similar to the results from the tissue and serum hRNase 1 protein expression with no significant restriction on specific molecular subtype(s) of breast cancer (**New Figs. 1e and 1g**).

In the Original Fig. S2a, the data of Kaplan-Meier survival curve revealed that hRNase 1 correlated negatively with the survival of breast cancer patients with the subtype of basal-like (HR = 1.34; $p = 0.048$), luminal A (HR = 1.35; $p = 0.00084$), and luminal B (HR = 1.39; $p = 0.0013$), but not HER2+ (HR = 0.85; $p = 0.43$). Of note, our newly-provided pertinent results by IHC, ELISA, and WB demonstrated that the alternation of hRNase 1 protein levels is not restricted to specific molecular subtype(s) of breast cancer. The discrepancy between our results and mRNA analysis from Kaplan-Meier survival curve in the public database may come from the trend of hRNase 1 mRNA expression does not fully correlate with its protein expression profiles among tumor samples. The complication could be caused by post-transcriptional and/or post-translational regulation of hRNase 1 gene in the tumor microenvironment. Thus, to study the significance of the hRNase 1-EphA4 axis more comprehensively in breast cancer, we have followed the reviewer's suggestions to performed new experiments in multiple luminal breast cancer cells line In the revised manuscript. Please also see the **Authors' Response to Points #4, #7, and #9**.

Original Fig. S2a

New Fig. S2b

New Fig. S2c

Original Fig. 1d

Original Fig. 1e (Now Fig. 1f)

New Fig. 1e

New Fig. 1g

Original Fig. 1f (Now Fig. 1h)

Original Fig. 1f (Now Fig. S4f)

New Fig. 1i

Point #4: Figure 2 - BT-549 and KPL4 are TNBC lines. As hRNase 1 expression only significantly correlates with poor outcomes in human Luminal breast cancers and not TNBC, it is unclear why this cell line was chosen and how physiologically relevant the data are. Why not use Luminal line T47D, which is more relevant and expresses higher levels of hRNase 1 than BT-549 (Figure 1F)?

Response to Point #4: We thank the reviewer for the comments and suggestions. As mentioned above, high hRNase 1 expression also significantly correlated with poor outcomes in patients with basal-like (TNBC) subtype (HR = 1.34; 95% CI, 1 to 1.79; p = 0.048), although the p values of luminal A and B subtypes were displayed more significantly. Regarding the choice of basal-like BT-549 and HER2+ KPL4 cell lines, we started with a proof of concept for the role of hRNase 1 in breast cancer, and thus we utilized cell lines which harbored low (BT-549) and high (KPL4) endogenous hRNase 1 protein expression to perform hRNase 1 ectopic expression and

knockdown/knockout experiments, respectively. We agree with the reviewer's comment on the physiological relevance to study luminal subtypes of breast cancer. In the revised manuscript, we have followed the suggestions to validate our initial observations (Original Fig. 2) in MCF7 and T-47D luminal cell lines, which displayed lower levels of endogenous hRNase 1 than the other two luminal cells tested (ZR-75-1 and BT-474; **New Fig. 1i**). The details are described below.

First, we ectopically expressed hRNase 1 in MCF7 cells (**New Fig. S5a**) and found that hRNase 1 significantly increased the number of sphere formation in primary, secondary, and tertiary passages (**New Figs. 2a and 2b**). Moreover, hRNase 1 enriched the population of CD24⁻/CD44⁺ cells in MCF7 parental cells and formed spheres (**New Figs. 2c and 2d**). These results demonstrated that the elevated hRNase 1 increased the sphere-forming ability and the population of CD24⁻/CD44⁺ cells in MCF7 luminal A breast cancer cells. **Second**, an inactivated form of hRNase 1 (catalytic-deficient R1-H12A mutant) and its wild-type counterpart (R1) were ectopically expressed in T-47D cells (**New Figs. S5g–S5i**). Either R1 or R1-H12A increased the number and size of spheres in T-47D cells (**New Figs. S5j–S5l**), indicating that the ribonucleolytic activity-independent function of hRNase 1 increased both the number and size of spheres in T-47D luminal A breast cancer cells. Together, consistent with the original findings demonstrated in basal-like BT-549 and HER2+ KPL4 breast cancer cells, the stemness-promoting feature of hRNase 1 also occurs in luminal type breast cancer cells independently of its ribonucleolytic activity.

Original Fig. S2a

New Fig. 1i

New Fig. S5a

New Fig. 2c

New Fig. 2a

New Fig. 2b

New Fig. 2d

New Fig. S4e

New Fig. S4e

New Fig. S4f

New Fig. S4g

New Fig. S4h

New Fig. S4i

Point #5: Figure 3 – F PLA data difficult to interpret with small panels. Were PLA data quantified?

Response to Point #5: We thank the reviewer for the comments. We have enlarged the Duolink *in situ* PLA images (Top; **Modified Fig. 3f**; BT-549, enlargements of the boxed areas at 9× magnification; KPL4, enlargements of the boxed areas at 12.25× magnification) and added the quantified data counted from three independent fields of each pool (Bottom; **Modified Fig. 3f**). The results showed that the PLA signals representing the interaction between EphA4 and hRNase 1 were solely detected by anti-EphA4 and anti-Flag antibodies in the recipient cells treated with CM containing Flag-tagged hRNase 1 (293T-pCDH-R1). We have moved the original Fig. 3f to the supplement Fig. S6g in the revised version.

Modified Fig. 3f

Original Fig. 3f (Now Fig. S6g)

Point #6: Figure 4 – G PLA data very difficult to interpret in these small panels.

Response to Point #6: We have enlarged the Duolink *in situ* PLA images as shown below (Top; **Modified Fig. 4g**). Insets are enlargements of the boxed areas at 6.25× magnification. The PLA results demonstrated that the hRNase 1-EphA4 interaction significantly decreased in the absence of hRNase 1 C-terminus (Δ C) but not WT or the N-terminal deletion mutant (Δ N) as indicated by

the reduced PLA signals, suggesting that the C-terminal domain of hRNase 1 is required for its binding to EphA4.

Modified Fig. 4g

Point #7: *Figure 6 – see comments above: Why use KPL4 and BT-549 rather than luminal lines? Does EphA4 have the same impact on hRNase 1-mediated tumorigenesis in luminal lines? What are endogenous levels of mRNase 1 in 4T1 cells (also model of TNBC rather than Luminal breast cancer)?*

Response to Point #7: As we mentioned in the **Authors' Response to Point #4**, we agree and thank for the reviewer's comments on the physiological relevance to study luminal subtypes of breast cancer. Following the reviewer's suggestions, we conducted new experiments to examine the biological significance of hRNase 1-EphA4 axis in multiple luminal breast cancer cell lines, including luminal A (MCF7 and ZR-75-1) and luminal B (BT-474). These new results showed that EphA4 positively regulated hRNase 1-mediated breast cancer stem-like properties in both luminal A and B subtypes of breast cancer cells. The details are described as follows. **First**, silencing EphA4 by shRNA knockdown in hRNase 1-expressing MCF7 cells (**New Fig. S11a**) attenuated the hRNase 1-induced sphere-forming frequency (**New Figs. 6g and S11b**) and stem-like cell population (**New Figs. 6h and 6i**). **Second**, ectopic expression of EphA4 in ZR-75-1 cells (**New Fig. S10a**) upregulated sphere formation (**New Figs. 6a and S10b**) and enriched the population of stem-like cells (**New Figs. 6b and 6c**), whereas knockdown of hRNase 1 abrogated those effects. **Third**, we observed similar results in BT-474 cells (**New Fig. S10c**) from sphere formation assay (**New Figs. S10d and S10e**) and flow cytometry analysis of CD24/CD44 expression (**New Figs. S10f and S10g**). Thus, together with the original results, we further strengthened our initial observations and provided more insights into the biological significance of hRNase 1-EphA4 axis to provoke the stem-like

cell properties among the four major molecular subtypes of breast cancer, including basal-like (BT-549), HER2+ (KPL4), luminal A (MCF7, T-47D, and ZR-75-1), and luminal B (BT-474).

Regarding the endogenous levels of mRNase 1 in 4T1 cells, it was undetectable in 4T1 cells expressing empty vector (4T1-vector) using RNase 1 antibody (1:1,000; Santa Cruz Biotechnology, #sc-169198; **New Fig. S12b**), suggesting that 4T1 cells harbor low abundance of endogenous mRNase 1.

New Fig. S11a

New Fig. S11b

New Fig. 6g

New Fig. 6h

New Fig. 6i

New Fig. S10a

New Fig. S10b

New Fig. 6a

New Fig. 6b

New Fig. 6c

New Fig. S10c

New Fig. S10f

New Fig. S10d

New Fig. S10g

New Fig. S10e

New Fig. S12b

Point #8: *Extended Data Figure 3 – What BC subtypes are represented in patient sera samples? Luminal A/B?*

Response to Point #8: We thank the reviewer for the comment. To address the reviewer's question, we have classified the molecular subtypes in sera samples of patient group (T) from three independent cohorts (Original Figs. S3a–S3c; Figs. S4a–S4c of the revised manuscript). In the cohort of the original Fig. S3a (Now Fig. S4a), the levels of serum hRNase 1 had no statistically

significant difference among the four subtypes of breast cancer (One-way ANOVA test; **New Fig. S4d**). In the cohort of the original Fig. S3b (Now Fig. S4b), patients were classified into luminal A and B subtypes with no significant difference of their serum hRNase 1 levels (Unpaired t test; **New Fig. S4e**). All patients in the cohort of the original Fig. S3c (Now Fig. S4c) belong to HER2+ subtype. Please also see the **Authors' Response to Point #3** for relevant description.

Original Fig. S3a
(Now Fig. S4a)

Original Fig. S3b
(Now Fig. S4b)

Original Fig. S3c
(Now Fig. S4c)

New Fig. S4d

New Fig. S4e

Point #9: *Extended Data Figure 5 – BT-549 was derived from TNBC. As hRNase 1 expression only significantly correlates with poor outcomes in human Luminal breast cancers and not TNBC, it is unclear why this cell line was chosen and how physiologically relevant the data are.*

Response to Point #9: As we explained in the **Authors' Response to Point #4**, high hRNase 1 expression also significantly correlated with poor outcomes in patients with basal-like (TNBC) subtype (HR = 1.34; 95% CI, 1 to 1.79; p = 0.048). However, we do agree that studying the hRNase 1-EphA4 axis in luminal breast cancer cells might be more physiologically relevant based on the results of Kaplan-Meier survival curve suggested (Original Fig. S2a). To this end, we performed new experiments relevant to the original Fig. S5 (Fig. S6 of the revised version) to see whether hRNase 1 activates EphA4 signaling in luminal breast cancer cells. Indeed, hRNase 1 also induced

EphA4 phosphorylation on Y602 and Y779 and triggered EphA4 downstream PLC γ 1 signaling in a time-dependent manner in MCF7 (**New Fig. 3d**) and T-47D (**New Fig. S6d**) luminal breast cancer cells. Moreover, we have demonstrated hRNase 1 association with EphA4 by co-immunoprecipitation assay in MCF7 cells in the original manuscript (Original Fig. 3e). Together, these results are consistent with the original findings shown in basal-like BT-549 cells, suggesting the hRNase 1-EphA4 axis might be a general ligand-receptor relationship across breast cancer subtypes.

New Fig. 3d

New Fig. S6d

Original Fig. 3e

Point #10: *Extended Data Figure 5F – PLA signal weak and unconvincing in HeLa model*

Response to Point #10: We thank the reviewer for the comment. We used HeLa cell line as a general model for pilot experiments in order to explore which RTK was activated by hRNase 1 before pursuing this topic in breast cancer research. To make the PLA signal clearer, we enlarged the PLA images as shown below (**Modified Fig. S5f**; Fig. S6i of the revised version). Insets are enlargements of the boxed areas at 9 \times magnification. We also added the quantified results in parentheses showing the percentage of cells with positive PLA signals calculated from a pool of 50 cells.

Modified Fig. S5f (Now Fig. S6i)

Point #11: *Extended Data Figure 6 – What does the partial band in the deltaLBD lane represent?*

Response to Point #11: We thank the reviewer for pointing this out. To improve data quality, we repeated the experiment and added a negative control of CM source, namely the CM from the empty vector-expressing stable transfectant, in comparison to that from hRNase 1-expressing stable transfectant (lanes 1–3, 293T-pCDH vs. lanes 4–6, 293T-pCDH-R1; **New Fig. S7**). The new results showed that hRNase 1 collected from CM in 293T-pCDH-R1 solely interacted with cells expressing EphA4-WT (lane 5), and deletion of the EphA4 ligand-binding domain (Δ LBD) abrogated its association with hRNase 1 (lane 6; improved with no partial band), indicating the LBD of EphA4 is critical for hRNase 1 binding.

New Fig. S7

Point #12: *Extended Data Figure 7 – See comment on Extended Data Figure 5 – why use BT-549 TNBC cell line instead of a luminal breast cancer cell line for analysis? The signaling data are relatively weak, particularly in the model of ectopic hRNase 1 expression in BT-549 cells, which appear to express abundant endogenous protein. Why not stimulate cells with hRNase 1 and look at NF- κ B nuclear localization and promoter binding, Erk, Src, and Akt phosphorylation? For D, it seems very odd that pharmacologic inhibitors have no apparent effect on spheroid formation for control BT-549, which should be dependent upon these driver signaling pathways for growth.*

Response to Point #12: We thank the reviewer for the insightful comments. As we mentioned in the **Authors' Response to Points #4, #7, and #9**, we have followed the reviewer's suggestions to perform new experiments in multiple luminal breast cancer cell lines; high hRNase 1 expression also significantly correlated with poor outcomes in patients with basal-like (TNBC) subtype (HR = 1.34; 95% CI, 1 to 1.79; $p = 0.048$). In addition, we apologize for the confusion about the use of ectopic expression of hRNase 1 in BT-549 cell line, which was characterized to express low levels of endogenous hRNase 1 in the CM by ELISA and WB in the original manuscript. In brief, the original ELISA results showed that CM hRNase 1 from breast cancer cell lines examined including BT-549 were determined at a concentration lower than 15 ng/ml, except KPL4 cells carrying CM hRNase 1 up to 100 ng/ml (Original Fig. 1f; Fig. S4f of the revised version). The original WB data also revealed the levels of CM hRNase 1 undetectable in BT-549 control cells expressing empty vector (BT-549-Ctrl) (Original Fig. 5f). In the revised manuscript, we screened a panel of 11 breast cancer cell lines including BT-549 to measure the levels of CM hRNase 1 using WB, and further clarified that BT-549 cells exhibited relatively low levels of endogenous hRNase 1 among them (**New Fig. 1i**). Please also see the **Authors' Response to Point #3**.

Next, we followed the reviewer's suggestion to stimulate MCF7 luminal breast cancer cells with hRNase 1 to check EphA4 downstream signaling molecules, including Erk, Src, Akt and NF- κ B (**New Figs. S8d and S8e**). We found that nuclear localization of p65 NF- κ B was accumulated time-dependently in response to hRNase 1 treatment in MCF7 cells (**New Fig. S8d**). Moreover, phosphorylation of Erk, but not Src or Akt, was detected in MCF7 treated with hRNase 1 in a time-dependent manner (**New Fig. S8e**). In line with the original observations in basal-like BT-549 cells, hRNase 1 also activated EphA4 downstream signaling pathways including NF- κ B and Erk, but not Akt and Src, in MCF7 luminal cells.

As we mentioned earlier, regarding the concern "*it seems very odd that pharmacologic inhibitors have no apparent effect on spheroid formation for control BT-549*", we selected effective but relatively low concentrations (low nM to low μ M) of pharmacologic inhibitors of IKK/NF- κ B (QNZ; 30 nM) and MEK/Erk (GSK1120212 and PD0325901; 1 nM and 0.2 μ M, respectively) which did not significantly damage the sphere-forming ability of control BT-549 (BT-549-Ctrl), in comparison to the results of hRNase 1-expressing stable cells (BT-549-R1) (Original Fig. S7d; Fig. S9a of the revised version). Of note, although the sphere-forming ability in BT-549-Ctrl cells was not significantly attenuated by IKK/NF- κ B and MEK/Erk inhibitors, the inhibitory effect was much more profound in BT-549-R1 clones, suggesting that BT-549-R1 cells may be more addicted to IKK/NF- κ B and MEK/Erk activating pathways, leading to a higher sensitivity to those inhibitors. In the revised manuscript, we further validated that phosphorylation of p65 NF- κ B and Erk were

indeed declined under inhibitor treatment in such concentrations in BT-549-Ctrl cells, supporting the effectiveness of inhibitors (**New Fig. S9b**).

Furthermore, we performed new sphere-forming experiments in MCF7 luminal cells in response to inhibitor treatment and observed similar results. In brief, hRNase 1-expressing MCH7 stable cells treated with various inhibitors, including those against IKK/NF- κ B (Bay 11-7821; 0.2 μ M) and MEK/Erk (PD-0325901; 2 μ M), exhibited attenuated sphere-forming ability compared with control cells (**New Figs. S9c and S9d**). However, there is no inhibitory effect on sphere-forming ability in cells treated with inhibitors against Akt (MK-2206; 20 nM) and Src (Dasatinib; 2 nM) (**New Figs. S9c and S9d**), consistent with the results from MCF7 cells showing that hRNase 1 activated NF- κ B and Erk, but not Akt and Src (**New Figs. S8d and S8e**). We have validated that phosphorylation of p65 NF- κ B, Erk, Akt, and Src were indeed attenuated after inhibitor treatment in MCF7 control cells (**New Figs. S9e–S9g**). These results together with the original findings in BT-549 cells demonstrated that hRNase 1 activates IKK/NF- κ B and MEK/Erk pathways, which positively regulates stem-like cell properties, but not Src and Akt pathways.

Original Fig. 1f (Now Fig. S4f)

Original Fig. 5f

New Fig. 1i

New Fig. S8d

New Fig. S8e

Original Fig. S7d (Now Fig. S9a)

New Fig. S9b

New Fig. S9c

New Fig. S9d

New Fig. S9e

New Fig. S9f

New Fig. S9g

Point #13: *Extended Data Figure 8 – Same concern with use of TNBC line BT549. KPL-4 is a HER2+ cell line. As hRNase 1 expression only significantly correlates with poor outcomes in human Luminal breast cancers and not HER2+ breast cancers, it is unclear why this cell line was chosen and how physiologically relevant the data are.*

Response to Point #13: We thank the reviewer for the comment. Although mRNA analysis from Kaplan-Meier survival curve showed no significant correlation between hRNase 1 and survival in patients with HER2+ breast cancers (Original Fig. S2a), as we discussed in the **Authors’ Response to Point #3**, our new results by IHC, ELISA, and WB demonstrated that the alternation of hRNase 1 protein levels is not restricted to specific molecular subtype(s) of breast cancer. The discrepancy between our results and mRNA analysis in the public database may come from the trend of hRNase 1 mRNA expression does not fully correlate with its protein expression profiles among tumor samples. The complication could be caused by post-transcriptional and/or post-translational regulation of hRNase 1 gene in the tumor microenvironment. To study the significance of the hRNase 1-EphA4 axis more comprehensively in breast cancer, we followed the reviewer’s suggestions to performed new experiments in multiple luminal breast cancer cells line. The new results together with the original findings demonstrated that the hRNase 1-EphA4 axis provoked the stem-like cell properties among the four major molecular subtypes of breast cancer, including basal-like (BT-549), HER2+ (KPL4), luminal A (MCF7, T-47D, and ZR-75-1), and luminal B (BT-474). Please also see the **Authors’ Response to Points #4, #7, and #9**.

Point #14: *Extended Data Figure 9 – It would help to show representative images from TMAs to assess staining specificity and localization of EphA4 and hRNase 1.*

Response to Point #14: We thank the reviewer for the suggestion. We have included representative images from the TMA data (Original Fig. S9; Fig. S13a in the revised version) as shown below (Case 1 and Case 2; **New Fig. S13b**).

Original Fig. S9 (Now Fig. S13a)

		phospho-EphA4		Total
		Low	High	
hRNase 1	Low	16	8	24
	High	4	20	24
Total		20	28	48

p = 0.0001

New Fig. S13b

Reviewer #3 (Remarks to the Author):

The manuscript describes for the first time the interaction of human RNase1 with tyrosine kinase receptor ephrin A4 (EphA4). These are very novel and interesting results with potential applicability to cancer therapy. The paper includes both basic science and translational work to explore the potential of human RNase1 as a prognosis marker and target for breast cancer chemotherapy. Both approaches are very novel and results are of particular interest for the applied medicine field. The authors have proven the hRNase1-receptor interaction and activation by many complementary techniques, including overexpression, silencing and knockout cell assays, together with mutagenesis experiments. Conclusions are well supported by the experimental data. The manuscript is of general interest to Nature Communications readers. However, before publication the presentation of the data needs a bit more polishing. Mainly, due to the high amount of data and great variety of methodologies applied, figure legends must be as self-explanatory as possible. Besides, discussion is a bit poor and some of the presented results are not discussed and contrasted respect to previous available information.

General comments:

1- *Due to the high amount of data and distinct methodologies applied, it is necessary to assist the reader and provide essential information in figure legends. Specify for each panel the information on timing, protein concentration, cell lines and antibodies to facilitate the results interpretation and evaluate its significance.*

2- *Terminology should be very carefully chosen. It is not correct to say that RNase1 function is to activate tumorigenesis. It could be hypothesized that the protein is involved in tissue regeneration, remodeling, or embryonic development and its overabundance can promote cancer, as a dysregulation disease process. It is important to differentiate the protein physiological putative role in health and disease states.*

3- *Recent work on RNase1 should be cited in the introduction and /or discussion.*

4- *Selection of cell lines and methodological approaches used is not always justified in the text. Likewise, the criteria for the selection of protein and receptor mutants must be indicated.*

5- *Information on the RNase1 recombinant proteins used should be detailed. Glycosylated forms should be described before showing any related results (MW, etc..). Were expressed proteins from HEK cells quantified? When lysates from HEK cells are used, it should be necessary to provide an estimation of the expressed level of proteins. In the methodology section it is important to detail the protocol of characterization of glycosylated and non-glycosylated samples from HEK cell lysate.*

6- *Which is the protocol used in the case of prokaryote expression? Is the purity checked? Is the used recombinant protein properly processed, without its signal peptide? Check for H40A mutant nomenclature (H12A?). Reference to H40 is not correct. A detailed methodology here is crucial as it can greatly influence the results.*

7- *Methodology section should try to indicate always clearly all volumes and concentrations used. Also, try to avoid telegraphic figure legends. Sometimes the information required to reproduce the experiments is missing.*

8- *Choice of cell lines should be justified for each experiment. Also, analysis of data should take care whether the selected cell line or methodology cannot bias the results interpretations. Why so many data are done with transfected cell lines and not with the edited cell lines?*

Specific comments:

TITLE: it might be better to substitute “Eph receptor A4” with “ephrin A4 receptor”

ABSTRACT:

revise sentence “Here, we demonstrate that hRNase 1, independently OF its ribonucleolytic activity...”

“The discovery of hRNase 1 as a secretory ligand of EphA4 to enhance breast cancer stemness”. Substitute TO, with “that enhances” or another phrasal expression. The physiological role of rnase1 should not be to promote cancer cells.

INTRODUCTION

Page 2, line 3 from bottom: Apart from ref 1, add recent reference on RNase presence in biological fluids (PMID: 29867984)

Page 2, last line. Add together with refs 4 and 5: PMID: 32544330

Page 3, line 4: add ref PMID: 29867984 after “innate immunity”

Page 3, line 5: together with refs 7 and 8: add references PMID: 24201302 and PMID: 25354936

RESULTS

Page 5, line 4. Subtitle and section should be reorganized as not all the data reported here is related to the contribution of the catalytic activity. Maybe the section can be subdivided in two parts with their respective subtitles and specific conclusions.

Page 5: it is important to indicate first the rationale for the choice of the mutation to remove the protein catalytic activity.

Be careful. Which is residue H40? Are the authors counting the signal peptide? Does it refer to His12? It does not make sense to number the residues keeping the immature protein. It should be confirmed that the expression protocol to obtain recombinant protein achieves a proper signal peptide cleavage?

Also indicate that mutation is not altering the protein 3D structure, by including either the own data (for example a circular dichroism) or reference to previous work on H12A.

Page 5, line 10: before commenting on results of glycosylated samples, they should be introduced (number of glycosylated forms, MW, etc...), together with reference to previous available information. The vector used here for expression of wt and mutant should be indicated and justified.

Page 6, line 1. Indicate in text and figure legends the antibodies naming, referred to residue of tyrosine phosphorylation.

Page 6, line 7: indicate protocols and references for expression of recombinant proteins.

Include more details here and in methodology about glycosidase treatment and quantities of recombinant protein added.

Justify selection of cell lines for each assay, every time a new cell type is used.

Page 7, line 1: DuoLink results (Figure 3f) should be explained a bit more. As RNase5 control is included, text must also include the specific drawn conclusions for both proteins.

Page 7, 3 lines from bottom: justify residues selection for mutant construct.

Page 8, line 7: provide more information, maybe a reference specifically talking on receptor formation of clusters.

Page 8: the term ectopic expression might not be the most accurate to make reference of the presence of secretory protein in the media.

As RNase1 is a soluble protein not anchored to the membrane, to what extent does it make sense to evaluate the potential juxtacrine signaling mechanism? Is the term only referencing to an

action mediated by ephrins, which are working by a juxtacrine mechanism? This section might need some more justification.

Page 9, line 7: why most of the assays were performed with recombinant proteins or clones overexpressing RNase1 instead of analyzing directly the KO cell line?

Page 10, line 2. Explain first why the soft agar methodology is selected.

Page 10, line 14. Figures 6 j,k Significance is not very good. It would have been important to have more data and more conclusive results.

FIGURES:

Before figure 2a it is necessary to include information on glycosylated forms, as shown in figure 3, with MW, and digestion analysis to identify glycosylated and non-glycosylated samples, etc...

Figure 2. it is necessary to include the quantities of protein added, specify better the antibodies used. Stars for glycosylated and non-glycosylated are only indicated later in figure 3.

Figure 3a: screening of phosphorylated kinases seem to reveal some other positive hits apart from ephrin receptor, egfr and Tie-2. Detail information for ABCDEF in figure legend. Likewise, explain better in figure legend and text results the screening shown in extended figure 5a.

Figure 3b panel, indicate RNase1 protein concentration added and from where it comes from (HEK; E. coli...).

Figure 3, panels l,j,k should be shown before figure 2.

Figure 2k, ref. 27. Better add the methodology in the present manuscript instead of merely referencing to ref.27. Also, the reference here is misleading, as it is unclear whether it includes related results or only the methodology description.

Figure 4: better insert panel b before panel a, to describe the chimera used. Specify the two domains of the receptor and indicate that K653A refers to the receptor phosphorylation site.

Figure 5 d,e. Indicate what is exactly measured by optical density.

Figure 6: panel b: nomenclature R1-2 can be misleading and is not explained elsewhere. Panels i,j,k. Better order them by alphabetical order within the figure. Figure 6k is incredibly too small. Also, indicate more clearly data from knock down cell line, silencing protocol,.. In panel 6j, indicate each pair analyzed for significance. Why a significance cut-off of $p \leq 0.05$ was not applied? Panels g and h: explain more clearly significance test, mark significant data. Why so many decimals are included?

Extended figure 1: it could be worth to include in the same panel the results for hRNase1, to facilitate a side-by-side direct comparison. What happened with RNases 8, 9 and 10? Indicate in the text why information is not provided for these RNases.

Extended figure 2. Some results seem also significant but the authors are not commenting them. Why differences observed in panel 2d are not mentioned. Is breast cancer the single significant case? How far selection is justified.

Extended figure 4b. Indicate the number of assays for kinetic measurements. Volumes used for the assay. Graphic values are the average of how many replicates? It is also surprising that no activity is detected in the cell media from HEK cells, as other RNases, such as RNase2, would certainly be expressed in the kidney cell line.

Extended figure 7. Indicate significance in panel 7d.

Extended figure 8 panel a: what is KO control exactly?

DISCUSSION,

first sentence: “showing an unconventional function of hRNase 1 as a secretory ligand of EphA4 to promote breast cancer progression”. It is very important to rephrase this sentence. The word “function” here is not correct.

Discussion, line 4: Care must be taken when using terms autocrine/paracrine. Here it might be better not to specify as the only thing we know for certain is that action is not juxtacrine.

Page 12, first line: even if we do not have a cocrystal, we can infer some conclusions from an overall structural comparative analysis. It would certainly be interesting to know if there is any structural similarity between RNases and Ephrins.

Page 12, line 18: Discussion should include reference to results from glycosylated form and recent work in the field (PMID: 30633504; PMID: 32544330).

Page 12, line 4 from bottom: it could be worth to comment on the protein hyperglycosylation observed in some cancer cell lines, such as reported for RNase2 (PMID: 7616105).

Page 12, line 2 from bottom: change “function” with “role”.

The authors conclude that glycosylations are not altering the protein interaction with the receptor. They discard an enhancement of activity, but the possibility should be also considered that the glycosylation might cover the receptor binding site and reduce the interaction.

Discussion is a bit poor and too short. It could be worth to comment recent data on RNase 5 and egfr receptor, discuss the results from the screening using the phosphoreceptors kit, the regions identified to be involved in interaction by site directed mutagenesis of protein and receptor, etc.

METHODS

Page 35, elisa. “briefly the plates were precoated with Elisa plates are prepared manually or they are provided with the kit? Specify otherwise the antibodies used.

Page 38: indicate the antibodies against phosphorylated receptors that nomenclature is referred to tyr residue,...

Authors’ Response: We would like to thank the reviewer for recognizing the novelty of our study with potential applicability to cancer therapy and general interest to *Nature Communications* readers. The reviewer’s thoughtful assessment and constructive criticisms have undoubtedly strengthened our findings and improved the scientific merits of this manuscript. We have carefully addressed each of the concerns according to the reviewer’s suggestions as shown below in a point-by-point manner.

General comments:

Point #1: *Due to the high amount of data and distinct methodologies applied, it is necessary to assist the reader and provide essential information in figure legends. Specify for each panel the information on timing, protein concentration, cell lines and antibodies to facilitate the results interpretation and evaluate its significance.*

Response to Point #1: We thank and agree with the reviewer’s comment. We have provided the detailed information as suggested in the Figure Legends section of the revised manuscript. Please also see the **Authors’ Responses to FIGURES Points #1–#14.**

Point #2: *Terminology should be very carefully chosen. It is not correct to say that RNase1 function is to activate tumorigenesis. It could be hypothesized that the protein is involved in tissue*

regeneration, remodeling, or embryonic development and its overabundance can promote cancer, as a dysregulation disease process. It is important to differentiate the protein physiological putative role in health and disease states.

Response to Point #2: We thank the reviewer for the insightful comment. We have rephrased the first few sentences of the Discussion section in the revised manuscript as bold texts shown below. Please also see the **Authors' Responses to ABSTRACT and DISCUSSION Points #1.**

“On the basis of our findings, we present a model (Fig. 7f) showing **an unconventional role of hRNase 1 as a secretory ligand of EphA4 that induces breast tumor initiation.** Elevated serum hRNase 1 binds to EphA4 and triggers EphA4 signaling in an autocrine/paracrine manner, which in turn promotes breast cancer initiation via the IKK/NF- κ B and MEK/Erk activating pathways. **In addition to the physiological roles of hRNase 1 involved in hemostasis, inflammation, and innate immunity², the overabundance of hRNase 1 can contribute to cancer progression as a dysregulation disease process.**”

Point #3: Recent work on RNase1 should be cited in the introduction and /or discussion.

Response to Point #3: We have followed the reviewer's suggestion to cite recent work on hRNase 1 in the revised manuscript. Please see the **Authors' Responses to INTRODUCTION Points #1–#4 and DISCUSSION Point #4.**

Point #4: Selection of cell lines and methodological approaches used is not always justified in the text. Likewise, the criteria for the selection of protein and receptor mutants must be indicated.

Response to Point #4: We thank the reviewer for the comments. We have provided a more clear justification of cell line selection and detailed description of methodology as suggested. Please see the **Authors' Responses to RESULTS Points #3, #4, and #5; FIGURES Points #4, #6, #9, #12, and #14; METHODS Points #1 and #2.** In addition, we have indicated the criteria for the selection of hRNase 1 protein and EphA4 receptor mutants in the revised manuscript. Please see the **Authors' Responses to RESULTS Points #7 and FIGURES Point #7.**

Point #5: Information on the RNase1 recombinant proteins used should be detailed. Glycosylated forms should be described before showing any related results (MW, etc..). Were expressed proteins from HEK cells quantified? When lysates from HEK cells are used, it should be necessary to provide an estimation of the expressed level of proteins. In the methodology section it is important to detail the protocol of characterization of glycosylated and non-glycosylated samples from HEK cell lysate.

Response to Point #5: We thank the reviewer for the comments. Regarding the recombinant hRNase 1 protein purified from HEK293 cells, we purchased it from Sino Biological Inc. (#13468-H08H), which was constructed by expressing a DNA sequence encoding the human RNASE1 (Accession# P07998) (Met1-Thr156) containing a poly-histidine tag at the C-terminus with > 95 %

of purity as determined by SDS-PAGE. We have provided the suggested information and protocol in the revised manuscript. Please also see the **Authors' Responses to RESULTS Points #3 and #5; FIGURES Points #1 and #2.**

***Point #6:** Which is the protocol used in the case of prokaryote expression? Is the purity checked? Is the used recombinant protein properly processed, without its signal peptide? Check for H40A mutant nomenclature (H12A?). Reference to H40 is not correct. A detailed methodology here is crucial as it can greatly influence the results.*

Response to Point #6: We thank the reviewer for pointing this out. Indeed, the residue H40 that we numbered in the original manuscript was based on the immature protein including the signal peptide, and it does refer to His12 of the mature hRNase 1 protein as the reviewer advised. We have corrected the residue number throughout the manuscript and figures accordingly. In addition, we have referred to previous studies (PMID: 25336120; PMID: 29606349) and confirmed that the expression protocol to obtain recombinant protein can achieve a proper signal peptide cleavage. We also provided the detailed protocol for the prokaryote expression of recombinant proteins. Please see the **Authors' Responses to RESULTS Points #2 and #5.**

***Point #7:** Methodology section should try to indicate always clearly all volumes and concentrations used. Also, try to avoid telegraphic figure legends. Sometimes the information required to reproduce the experiments is missing.*

Response to Point #7: We thank the reviewer for the comments. We have provided a more detailed description of methodology as suggested. Please see the **Authors' Responses to RESULTS Points #3, #4, and #5; FIGURES Points #4, #6, #9, #12, and #14; METHODS Points #1 and #2.**

***Point #8:** Choice of cell lines should be justified for each experiment. Also, analysis of data should take care whether the selected cell line or methodology cannot bias the results interpretations. Why so many data are done with transfected cell lines and not with the edited cell lines?*

Response to Point #8: We thank the reviewer for the comments. As we mentioned in the **Authors' response to General comments Point #4**, we have provided a more clear justification of cell line selection and detailed description of methodology as suggested in the revised manuscript. Regarding the results done with stable transfected cell lines overexpressing hRNase 1 (Figs. S7a–S7d of the original manuscript; Figs. S8a–S8c and S9a of the revised version), we have provided a more clear explanation in the **Authors' Response to RESULTS Point #10**. Briefly, we intended to study the impact of EphA4 downstream signaling pathways on the hRNase 1-mediated sphere-forming ability, using stable transfected cell lines overexpressing hRNase 1 would be a simpler and more appropriate model, compared with the KO system, in response to treatments with various pharmacologic inhibitors. In the **Authors' Response to RESULTS Point #10**, we also summarized our key results done with the KO edited cell lines in KPL4 by examining the ALDH1 activity, sphere-forming capacity, and *in vivo* TIC frequency, in order to study the functional roles of endogenous hRNase 1.

Specific comments:

TITLE: *it might be better to substitute “Eph receptor A4” with “ephrin A4 receptor”*

Response to TITLE: We thank the reviewer for the comment. Following the reviewer’s suggestion, we have substituted “Eph receptor A4” with “ephrin A4 receptor” in the revised manuscript.

ABSTRACT: *revise sentence **Point #1:** “Here, we demonstrate that hRNase 1, independently OF its ribonucleolytic activity...”; **Point #2:** “The discovery of hRNase 1 as a secretory ligand of EphA4 to enhance breast cancer stemness”. Substitute TO, with “that enhances” or another phrasal expression. The physiological role of rnase1 should not be to promote cancer cells.*

Response to ABSTRACT: We thank the reviewer for the comments. We have revised the sentences in the Abstract section of the revised manuscript as follows:

“Here, we demonstrate that hRNase 1, independently **of** its ribonucleolytic activity, enriches the stem-like cell population and enhances the tumor-initiating ability of breast cancer cells.”

“The discovery of hRNase 1 as a secretory ligand of EphA4 **that enhances** breast cancer stemness suggests a potential treatment strategy by inactivating the hRNase 1-EphA4 axis.”

INTRODUCTION: Point #1: *Page 2, line 3 from bottom: Apart from ref 1, add recent reference on RNase presence in biological fluids (PMID: 29867984); **Point #2:** Page 2, last line. Add together with refs 4 and 5: PMID: 32544330; **Point #3:** Page 3, line 4: add ref PMID: 29867984 after “innate immunity”; **Point #4,** Page 3, line 5: together with refs 7 and 8: add references PMID: 24201302 and PMID: 25354936.*

Response to INTRODUCTION Points #1–#4: We have included these references as suggested in the revised manuscript.

RESULTS:

Point #1: *Page 5, line 4. Subtitle and section should be reorganized as not all the data reported here is related to the contribution of the catalytic activity. Maybe the section can be subdivided in two parts with their respective subtitles and specific conclusions.*

Response to RESULTS Point #1: We thank the reviewer for the comment and suggestion. To more clearly conclude our results, we have subdivided this section on **pages 6 and 8** into two parts with their respective subtitles as shown below.

“The ribonucleolytic activity-independent function of hRNase 1 enriches the population of breast CSC-like cells and enhances the tumor-initiating capability”

“Silencing hRNase 1 reduces the population of breast CSC-like cells and decreases the tumor-initiating capability”

Point #2: Page 5: it is important to indicate first the rationale for the choice of the mutation to remove the protein catalytic activity. Be careful. Which is residue H40? Are the authors counting the signal peptide? Does it refer to His12? It does not make sense to number the residues keeping the immature protein. It should be confirmed that the expression protocol to obtain recombinant protein achieves a proper signal peptide cleavage? Also indicate that mutation is not altering the protein 3D structure, by including either the own data (for example a circular dichroism) or reference to previous work on H12A.

Response to RESULTS Point #2: We thank the reviewer for pointing this out. Indeed, the residue H40 that we numbered in the original manuscript was based on the immature protein including the signal peptide, and it does refer to His12 of the mature hRNase 1 protein as the reviewer advised. We have corrected the residue number throughout the manuscript and figures accordingly. In addition, we have referred to previous studies (PMID: 25336120; PMID: 29606349) and confirmed that the expression protocol to obtain recombinant protein can achieve a proper signal peptide cleavage. We also indicated the rationale for the choice of the mutation to remove the protein catalytic activity and the statement that the H12A variant does not disturb the overall three-dimensional structure of hRNase 1 protein on **page 6** in the revised manuscript as shown bold texts below.

“Considering Histidine 12 (H12) is critical for ribonucleolytic activity to cleave RNA by hRNase 1 (PMID: 11848924), we generated a catalytically inactive H12A variant of hRNase 1 with the amino acid histidine to alanine substitution (R1-H12A; Fig. S5b), which does not disturb the overall three-dimensional structure of hRNase 1 protein (PMID: 11305910). Indeed, the H12 variant exhibited virtually no enzymatic activity compared with its wild-type counterpart (R1; Fig. S5c).”

Point #3: Page 5, line 10: before commenting on results of glycosylated samples, they should be introduced (number of glycosylated forms, MW, etc...), together with reference to previous available information. The vector used here for expression of wt and mutant should be indicated and justified.

Response to RESULTS Point #3: We thank the reviewer for the suggestions. We have introduced more background of N-linked glycosylation of hRNase 1 on **page 7** as shown below. We also indicated molecular weight with markers for the hRNase 1 and β -actin blots throughout the figures (Figs. 2a, 2e, 2h, and S4a of the original manuscript; Figs. S5d, S5m, 2l, and S5b of the revised version). In addition, we used the lentiviral vector pCDH-CMV-MCS-EF1-puro/NEO as a vector for the expression of hRNase1, which we have described under “*Plasmids, siRNAs, shRNA clones, and knocking out constructs*” in the Method section in the revised manuscript.

“In addition, hRNase 1 has three N-linked glycosylation sites reported at Asn-34, Asn-76, and Asn-88, and the heterogeneous pattern of hRNase 1 on gel electrophoresis reflected

different levels of glycosylation depending on cell and tissue types by a range of bands from 15 to 36 kDa, in which the relatively lowest molecular weight band at ~15 to 20 kDa corresponds to a non-glycosylated hRNase 1 (PMID: 32544330; PMID: 12626415)."

"*Plasmids, siRNAs, shRNA clones, and knocking out constructs.* The pCDH-R1 (Flag-tagged hRNase 1), pCDH-hRNase 5 (Flag-tagged hRNase 5), and pCDH-A4 (Myc-tagged EphA4) expression vectors were generated by inserting the full-length cDNA (hRNase 1: NM_002933.4; hRNase 5: NM_001145; EphA4: BC105002) into the lentiviral vector pCDH-CMV-MCS-EF1-puro/NEO. pCDH-R1-H12A (Flag-tagged hRNase 1 with H12A mutation) was generated and derived from pCDH-R1 plasmid by site-directed mutagenesis."

Original Fig. 2a (Now Fig. S5d)

Original Fig. S4a (Now Fig. S5b)

Original Fig. 2e (Now Fig. S5m)

Original Fig. 2h (Now Fig. 2I)

Point #4: Page 6, line 1. Indicate in text and figure legends the antibodies naming, referred to residue of tyr phosphorylation.

Response to RESULTS Point #4: We thank the reviewer for the comment. We performed human phospho-RTK antibody array (R&D Systems, #ARY001B) according to the manufacturer's instruction to detect the activation signals of potential RTKs by hRNase 1 treatment. We apologize for not clearly describing the principle of this assay kit in the original manuscript. In brief, we used a pan anti-phospho-tyrosine antibody conjugated to HRP for the detection of

phosphorylated tyrosines bound to the captured RTKs, thus there is no individual residue of tyrosine phosphorylation provided in this assay. We have made a statement carefully under “*Human phospho-RTK antibody array*” in the Methods section in the revised version.

“*Human phospho-RTK antibody array*. Proteome Profiler Human Phospho-RTK Array Kit (R&D Systems, #ARY001B) was used to detect the potential activation of RTK signals by hRNase 1 treatment. All procedures were performed according to the manufacturer’s instruction. Briefly, capture antibodies for specific RTKs were spotted in duplicate onto nitrocellulose membranes, prepared with the kit. Cell lysates (600 µg) were incubated with the array membrane at 4°C overnight. After washing away unbound material, proteins in cell lysates containing phosphorylated tyrosine residues bound to the capture antibodies of RTKs were **detected by a pan anti-phospho-tyrosine antibody conjugated to horseradish peroxidase**. Last, the binding signal was measured by using chemiluminescent detection reagents and ImageQuant LAS 4010 (GE Healthcare).”

Point #5: Page 6, line 7: indicate protocols and references for expression of recombinant proteins. Include more details here and in methodology about glycosidase treatment and quantities of recombinant protein added. Justify selection of cell lines for each assay, every time a new cell type is used.

Response to RESULTS Point #5: We thank the reviewer for the comments. We produced recombinant hRNase 1 protein from *E. coli* by using a similar protein expression protocol referred to previous studies (PMID: 29606349; PMID: 33031845). Briefly, for producing purified GST-hRNase 1 recombinant protein, hRNase 1 cDNA without signal peptide sequence was inserted into pGEX6P1 to express GST-hRNase 1 protein in BL21(DE3) competent *E. coli*, followed by GST-tagged protein purification assay. For generating purified hRNase 1 protein with a Myc and a 6XHis tag at the C-terminus, hRNase 1-Myc-His fusion cDNA fragment was amplified from the hRNase 1 expression plasmid previously established in pcDNA6/Myc-His A vector, and inserted into pSJ3 vector to express hRNase 1 protein in BL21(DE3) competent *E. coli*, followed by His-tagged protein purification assay. We have provided a more detailed protocol under “*Generation of recombinant hRNase 1 proteins*” in the Methods section in the revised manuscript as shown below. On the other hands, recombinant hRNase 1 protein purified from HEK293 cells was purchased from Sino Biological Inc. (#13468-H08H), constructed by expressing a DNA sequence encoding the human RNASE1 (Accession# P07998) (Met1-Thr156) containing a poly-histidine tag at the C-terminus with > 95 % of purity as determined by SDS-PAGE. For the glycosidase pretreatment experiments, we pretreated 3 µg of recombinant hRNase 1 protein purified from HEK293 cells with or without 5% PNGase F glycosidase treatment, followed by *in vitro* binding assay through incubating the pretreated mixture with 1 ug of N-EphA4-Fc or human IgG. The more detailed description was provided in the Results section on **page 10** and under “*Glycosidase pretreatment of recombinant hRNase 1 protein*” in the Methods section as shown below. In addition, we have justified the selection of a new cell type used in the revised manuscript. Here, we used HeLa cell line as a general model for pilot experiments in order to explore which RTK was activated by hRNase 1 before pursuing this topic in breast cancer research.

“Generation of recombinant hRNase 1 proteins. We produced recombinant hRNase 1 protein from *E. coli* by using a similar protein expression protocol referred to previous studies (PMID: 29606349; PMID: 33031845). For producing purified GST-hRNase 1 recombinant protein, hRNase 1 cDNA without signal peptide sequence was inserted into pGEX6P1 to express GST-hRNase 1 protein in BL21(DE3) competent *E. coli*, followed by GST-tagged protein purification assay. For generating purified hRNase 1 protein with a Myc and a 6XHis tag at the C-terminus, hRNase 1-Myc-His fusion cDNA fragment was amplified from the hRNase 1 expression plasmid previously established in pcDNA6/Myc-His A vector, and inserted into pSJ3 vector to express hRNase 1 protein in BL21(DE3) competent *E. coli*, followed by His-tagged protein purification assay.”

“Considering hRNase 1 as a *N*-linked glycosylated protein (PMID: 32544330; PMID: 12626415; PMID: 17229815), we performed *in vitro* binding assay by incubating N-EphA4-Fc with recombinant hRNase 1 protein produced from HEK293 cells, which we pretreated with *N*-glycosidase (peptide-*N*-glycosidase F; PNGase F) to remove the glycan moieties from hRNase 1 (Fig. 3i), and found that non-glycosylated hRNase 1 still harbored the binding ability to N-EphA4-Fc (Fig. 3k).”

“Glycosidase pretreatment of recombinant hRNase 1 protein. Following the manufacturer’s instruction of glycoprotein treatment with PNGase F glycosidase (NEB Inc., #P0704), 3 µg of recombinant hRNase 1 protein (Sino Biological Inc., #13468-H08H) was combined with 1 µl of 10× Glycoprotein Denaturing Buffer and water to make up a 10-µl total reaction volume. The mixture was denatured by heating at 100°C for 10 min and chilled on ice for 2 min, and 2 µl of 10× GlycoBuffer 2, 2 µl of 10% Nonidet P-40, and 6 µl of water were then added to make up a 20-µl total reaction volume. The mixture was then incubated at 37°C overnight with or without 1 µl of PNGase F to keep the final glycerol concentration equal to 5% and subjected to *in vitro* binding assay.”

Point #6: Page 7, line 1: DuoLink results (Figure 3f) should be explained a bit more. As RNase5 control is included, text must also include the specific drawn conclusions for both proteins.

Response to RESULTS Point #6: We thank and agree with the reviewer’s comment. We have included the conclusions for both hRNase 1 and hRNase 5 proteins on **page 10** in the Results section as shown below. In addition, to make the PLA signal clearer, we enlarged the Duolink images (Top; **Modified Fig. 3f**; BT-549, enlargements of the boxed areas at 9× magnification; KPL4, enlargements of the boxed areas at 12.25× magnification) and added the quantified results counted from three independent fields of each pool (Bottom; **Modified Fig. 3f**). We have moved the Original Fig. 3f to the supplement Fig. S6g in the revised version.

“Notably, numerous PLA signals were solely detected by anti-EphA4 and anti-Flag antibodies in the recipient BT-549 and KPL4 cells treated with the CM containing Flag-tagged hRNase 1 (293T-pCDH-R1), but not the CM containing Flag-tagged hRNase 5 (293T-pCDH-hRNase 5) (Fig. 3f). These results suggested that EphA4 is in close proximity to hRNase 1 but not hRNase 5 as a comparison, in line with the previous results from phospho-RTK

antibody array indicating that hRNase 5 as an EGFR ligand induces phosphorylation of EGFR but not other RTKs (PMID: 29606349).”

Modified Fig. 3f

Original Fig. 3f (Now Fig. S6g)

Point #7: Page 7, 3 lines from bottom: justify residues selection for mutant construct.

Response to RESULTS Point #7: We thank the reviewer for the comment. We have justified residues selection for the GST-hRNase 1 deletion mutant constructs on **page 11** in the revised manuscript as shown bold texts below.

“To map the region of hRNase 1 required for EphA4 binding, **we generated GST-hRNase 1 deletion mutants of N-terminus (Δ N; amino acids 1-21 deletion) and C-terminus (Δ C; amino acids 113-128 deletion), which lost the domains containing the catalytically active residues, H12 and H119, respectively** (Figs. 4e,f). The results from Duolink *in situ* PLA demonstrated that the hRNase 1-EphA4 interaction significantly decreased in the absence of hRNase 1 C-terminus (Δ C) but not WT or the N-terminal deletion mutant (Δ N) as indicated by the reduced PLA signals, suggesting that the C-terminal domain of hRNase 1 is required for its binding to EphA4 (Fig. 4g).”

Point #8: *Page 8, line 7: provide more information, maybe a reference specifically talking on receptor formation of clusters.*

Response to RESULTS Point #8: We have followed the reviewer’s suggestion to add references talking on Eph receptor formation of clusters on **page 11** as follows:

“We demonstrated that hRNase 1 induced EphA4 dimerization/oligomerization (Fig. 5a), similar to the classical ephrin ligand-dependent Eph receptor activation, followed by the higher-order clustering for Ephs and ephrins interactions assembled into dimerization, tetramerization, and oligomerization (PMID: 23959867; PMID: 11780069; PMID: 17928214). It is worthwhile to mention that the recruitment of A and B type Eph receptors into signaling clusters can occur independent of ephrin contacts via direct Eph-Eph receptor interactions (PMID: 14993233; PMID: 22144690).”

Point #9: *Page 8: the term ectopic expression might not be the most accurate to make reference of the presence of secretory protein in the media. As RNase1 is a soluble protein not anchored to the membrane, to what extent does it make sense to evaluate the potential juxtacrine signaling mechanism? Is the term only referencing to an action mediated by ephrins, which are working by a juxtacrine mechanism? This section might need some more justification.*

Response to RESULTS Point #9: We thank the reviewer for the comments. We have replaced the term “ectopic expression” with “overexpression” as commended. We apologize for not clearly describing the rationale of this section. We would like to validate whether our proposed model namely hRNase 1-stimulated EphA4 activation is directly mediated by hRNase 1, by excluding the possibility that it could be an indirect outcome caused by hRNase 1 to elevate the levels of classical ephrin ligands, which turns to transmit activating signals through a classical juxtacrine signaling mechanism. We have rephrased the description of this section in the revised manuscript as shown below.

“We further asked whether hRNase 1-stimulated EphA4 activation might not be directly mediated by hRNase 1, but an indirect result of elevated levels of classical ephrin ligands.

To this end, we screened the entire ephrin ligand family, including ephrin A1–A5 and B1–B3. Among them, six ephrin ligands, ephrin-A1, -A2, -A4, -A5, -B1, and -B2, did not increase significantly in BT-549 clones overexpressing hRNase 1 compared with vector control cells (Figs. 5f,g), and the levels of ephrin-A3 and -B3 were undetectable in BT-549 cells (Fig. 5h). These results suggested that hRNase1-mediated EphA4 activation is specific and direct, instead of coming from an indirect effect by augmenting expression levels of classical ephrin ligands.”

Original Fig. 5f

Original Fig. 5g

Original Fig. 5h

Point #10: Page 9, line 7: why most of the assays were performed with recombinant proteins or clones overexpressing RNase1 instead of analyzing directly the KO cell line?

Response to RESULTS Point #10: We thank the reviewer for pointing this out. In the original manuscript, we have established KPL4 stable clones knocking out hRNase 1, KPL4-KO-R1 and KPL4-A4-KO-R1, respectively, in order to study the functional roles of endogenous hRNase 1 (Original Figs. 2h and S8a; Figs. 2l and S10h of the revised manuscript). We observed that hRNase 1 increases cancer stem-like properties through EphA4 by examining the ALDH1 activity (Original Figs. 2i and 6e; Now Figs. 2m and 6d), sphere-forming capacity (Original Fig. 6a; Now Fig. 6e), and *in vivo* TIC frequency (Original Fig. 6g; Now Fig. 6f).

For different purposes in the Original Supplementary Figs. S7a–S7d on **pages 12–14**, we intended to study whether hRNase 1 activates well-recognized EphA4 downstream signaling molecules, such as NF- κ B, Erk, Src, and Akt. To this end, we adapted approaches through addition of recombinant hRNase 1 proteins or stable clones overexpressing hRNase 1 to demonstrate the signaling pathways activated by hRNase 1 (Figs. S7a–S7c of the original manuscript; Figs. S8a–S8c of the revised version). To further study the impact of the above-mentioned signaling pathways on the hRNase 1-mediated sphere-forming ability, stable clones overexpressing hRNase 1 would be a simpler and more appropriate model, compared with the KO system, in response to

treatments with various pharmacologic inhibitors (Original Fig. S7d; Fig. S9a of the revised manuscript).

It is worthwhile to mention that, in addition to basal-like BT-549 breast cancer cell model, we also stimulated luminal subtype MCF7 breast cancer cells with hRNase 1 to check EphA4 downstream signaling molecules, including Erk, Src, Akt and NF- κ B (**New Figs. S8d and S8e**). We found that nuclear localization of p65 NF- κ B was accumulated time-dependently in response to hRNase 1 treatment in MCF7 cells (**New Fig. S8d**). Moreover, phosphorylation of Erk, but not Src or Akt, was detected in MCF7 treated with hRNase 1 in a time-dependent manner (**New Fig. S8e**). In line with the original observations in BT-549 basal-like cells, hRNase 1 also activated EphA4 downstream signaling pathways including NF- κ B and Erk, but not Akt and Src, in MCF7 luminal cells. Furthermore, we performed new sphere-forming experiments in MCF7 luminal cells in response to inhibitor treatment and observed similar results. In brief, hRNase 1-overexpressing MCH7 stable cells treated with various inhibitors, including those against IKK/NF- κ B (Bay 11-7821) and MEK/Erk (PD-0325901), exhibited attenuated sphere-forming ability compared with control cells (**New Figs. S9c and S9d**). However, there is no inhibitory effect on sphere-forming ability in cells treated with inhibitors against Akt (MK-2206; 20 nM) and Src (Dasatinib; 2 nM) (**New Figs. S9c and S9d**), consistent with the results from MCF7 cells showing that hRNase 1 activated NF- κ B and Erk, but not Akt and Src (**New Figs. S8d and S8e**). These results together with the original findings in BT-549 cells demonstrated that hRNase 1 activates IKK/NF- κ B and MEK/Erk pathways, which positively regulates stem-like cell properties, but not Src and Akt pathways.

Original Fig. 2h (Now Fig. 2l)

Original Fig. 2i (Now Fig. 2m)

Original Fig. S8a (Now Fig. S10h)

Original Fig. 6e (Now Fig. 6d)

Original Fig. 6a (Now Fig. 6e)

Original Fig. 6g (Now Fig. 6f)

Cell number	KPL4-NEO	KPL4-A4	KPL4-A4-KO-Ctrl	KPL4-A4-KO-R1
1 x 10 ⁵ cells	6/6	6/6	6/6	6/6
1 x 10 ⁴ cells	4/6	6/6	6/6	5/6
5 x 10 ³ cells	1/6	3/6	4/6	1/6
2 x 10 ³ cells	0/8	2/8	1/8	0/8
TIC frequency	1/16,026	1/5,082	1/4,878	1/12,472

**Original Fig. S7a
(Now Fig. S8a)**

**Original Fig. S7b
(Now Fig. S8b)**

**Original Fig. S7c
(Now Fig. S8c)**

**Original Fig. S7d
(Now Fig. S9a)**

New Fig. S8d

New Fig. S8e

New Fig. S9c

New Fig. S9d

Point #11: Page 10, line 2. Explain first why the soft agar methodology is selected.

Response to RESULTS Point #11: We have added the relevant explanation on pages 7 and 15 in the revised manuscript, namely that we evaluated the anchorage-independent cell growth performed by soft agar colony formation assay in the original Fig. 2b and S8e (Fig. S5f and S11e of the revised version) is to study whether EphA4 positively regulates hRNase 1-mediated *in vitro* oncogenic transformation potential, in addition to the stem-like properties.

Point #12: Page 10, line 14. Figures 6 j,k Significance is not very good. It would have been important to have more data and more conclusive results.

Response to RESULTS Point #12: We thank the reviewer for the comment. We have reanalyzed the original results with more data of mice included (Figs. 6j and 6k of the original manuscript; Fig. S12c of the revised version). The new results displayed the improved significance of p-values which showed a similar trend as observed in the original findings. In brief, we found that nude mice subcutaneously injected with 4T1-mRNase 1 (n = 6) developed tumors that weighed more than those from mice injected with vector control cells (n = 8; p = 0.009). The mRNAse 1-mediated tumorigenesis (n = 6) was suppressed when mice were treated with compound 1 (cpd1) (n = 8), a small molecule that binds to the EphA4-LBD (p = 0.026). Collectively, these findings suggested that the hRNase 1-EphA4 axis contributes to breast tumor progression. We have described the new data in the Results section on **page 15** of the revised manuscript. To mainly focus on the significance of stem-like cell properties regulated by the hRNase 1-EphA4 axis, we moved these *in vivo* tumorigenesis results (Original Figs. 6j and 6k) to Supplementary Fig. S12c with the enlargement of representative tumor images in the revised manuscript.

Original Figs. 6j and 6k (Now Fig. S12c)

FIGURES:

Point #1: Before figure 2a it is necessary to include information on glycosylated forms, as shown in figure 3, with MW, and digestion analysis to identify glycosylated and non-glycosylated samples, etc...

Response to FIGURES Point #1: We thank the reviewer for the comment. We have indicated molecular weight with markers for the hRNase 1 and β -actin blots in the Original Fig. 2a (Fig. S5d of the revised version). To identify glycosylated and non-glycosylated forms of hRNase 1, we followed the reviewer's suggestion to perform the digestion assay with recombinant glycosidase (peptide-N-glycosidase F; PNGase F) to remove the N-linked glycan moieties (**New Fig. S5e**). The

results showed that treatment with PNGase F in hRNase 1-overexpressing BT-549 cells (Flag-tagged BT-549-R1) resulted in a homogenous pattern of hRNase 1 at the relatively lowest molecular weight band below 20 kDa corresponding to a non-glycosylated hRNase 1 (black asterisk, non-glycosylated hRNase 1; red asterisk, glycosylated hRNase 1). In addition to basal-like subtype BT-549 cells, non-glycosylated form of hRNase 1 was also demonstrated as the same pattern in response to PNGase F digestion in HER2+ subtype KPL-4 (New Fig. S5n) and luminal subtype T-47D-expressing R1 or R1-H12A (New Fig. S5h) (black asterisk, non-glycosylated hRNase 1; red asterisk, glycosylated hRNase 1).

Original Fig. 2a (Now Fig. S5d)

New Fig. S5e

New Fig. S5n

New Fig. S5h

Point #2: Figure 2. it is necessary to include the quantities of protein added, specify better the antibodies used. Stars for glycosylated and non-glycosylated are only indicated later in figure 3.

Response to FIGURES Point #2: We thank the reviewer for the comments. As we described in the **Response to RESULTS Point #5**, we added 3 μg of recombinant hRNase 1 protein purified from HEK293 cells (Sino Biological Inc., #13468-H08H) for glycosidase pretreatment assay with or without 5% PNGase F (NEB Inc., #P0704). The pretreated mixtures were then subjected to *in vitro* binding assay through incubating with 1 ug of N-EphA4-Fc (Sino Biological Inc., #11314-H03H) or

human IgG, followed by pull-down assay with protein G beads. In addition, we have specified the antibodies used and included the description of Stars for glycosylated and non-glycosylated of hRNase 1 in Fig. 2l (Original Fig. 2h) and the end of Supplementary Fig. S5 (Original Figs. 2a and 2e; Figs. S5d and S5m of the revised manuscript) as follows:

“WB of KPL4 stable transfectants knocking out hRNase 1 and empty control with hRNase 1 (Sigma-Aldrich, #HPA001140) and β -actin (Sigma-Aldrich, #A2228) antibodies. Red asterisk, glycosylated hRNase 1; black asterisk, non-glycosylated hRNase 1.”

“Antibodies used in WB, hRNase 1 (Sigma-Aldrich, #HPA001140); Flag (Sigma-Aldrich, #F3165); β -actin (Sigma-Aldrich, #A2228); tubulin (Sigma-Aldrich, #T5168). Red asterisk, glycosylated hRNase 1; black asterisk, non-glycosylated hRNase 1.”

Point #3: *Figure 3a: screening of phosphorylated kinases seem to reveal some other positive hits apart from ephrin receptor, egfr and Tie-2. Detail information for ABCDEF in figure legend. Likewise, explain better in figure legend and text results the screening shown in extended figure 5a.*

Response to FIGURES Point #3: Following the reviewer’s suggestions, we have provided a more detailed information of the Original Figs. 3a and S5a (Figs. 3a and S6a of the revised version) in the Figure Legend and Results sections in the revised manuscript as bold texts shown below.

“*Fig. 3a of the original and revised versions.* Top, human phospho-RTK antibody array analysis of HeLa cells treated with or without recombinant hRNase 1 protein purified from HEK293 cells (1 μ g/ml) for 5 min after serum starvation for 3 hr. **Three pairs of positive signals in duplicate coordinates (- hRNase 1 comparing to + hRNase 1) are shown in EphA4 (D23/D24), HGFR (C3/C4), and Tie-2 (D1/D2).** Bottom, quantification of detected signals by ImageJ.”

“*Original Fig. S5a (Fig. S6a of the revised version).* Human phospho-RTK antibody array analysis of BT-549 cells treated with or without recombinant hRNase 1 protein purified from HEK293 cells (1 μ g/ml) for 30 min after serum starvation for 3 hr. **Three pairs of positive signals in duplicate coordinates (- hRNase 1 comparing to + hRNase 1) are shown in EphA4 (D23/D24), EphA10 (E21/E22), and ROR2 (C21/C22).**”

“*Results.* Considering the above results that an abundance of hRNase 1 was detected in the serum of breast cancer patients and that hRNase 1 played a positive role in breast tumor initiation, we explored the mechanistic aspects of secretory hRNase 1 and its biological impacts on tumor cells. **To this end, we performed an unbiased antibody array of human phospho-RTKs in HeLa epithelial cancer cells as a general model for pilot experiments before pursuing this topic in breast cancer research (Fig. 3a) and BT-549 breast cancer cells (Fig. S6a).** The results showed that treatment with recombinant hRNase 1 (purified from HEK293 cells) increased tyrosine phosphorylation of EphA4, hepatocyte growth factor receptor (HGFR), and TEK receptor tyrosine kinase (Tie-2) in HeLa cells, and that of

EphA4, EphA10, and receptor tyrosine kinase like orphan receptor 2 (ROR2) in BT-549 cells. Among them, EphA4 was the only receptor whose phosphorylation increased in both cell lines and previously reported to maintain breast CSC-like cells (PMID: 25266422). Hence, we focused on whether hRNase 1 affects the EphA4 pathway.”

Original Fig. 3a

Original Fig. S5a (Now Fig. S6a)

Point #4: Figure 3b panel, indicate RNase1 protein concentration added and from where it comes from (HEK; E. coli...).

Response to FIGURES Point #4: We thank the reviewer for the comment. We have made it more clear in the Figure Legend of Fig. 3b and under “Cell lines and treatment” in the Methods section in the revised manuscript as shown below.

“Fig. 3b of the original and revised versions. WB of BT-549 cells treated with recombinant hRNase 1 protein purified from HEK293 cells (1 μg/ml) at various time points, blotted with phospho-EphA4-Y779 (pY779-EphA4), EphA4, and β-actin antibodies.”

“Cell lines and treatment. Treatment with recombinant hRNase 1 protein purified from HEK293 cells (Sino Biological Inc. #13468-H08H-100) was carried out at a concentration of 1 μg/ml for 30 min or the indicated time after serum starvation for 3 hr.”

Point #5: Figure 3, panels I,j,k should be shown before figure 2.

Response to FIGURES Point #5: We thank the reviewer for the suggestion regarding data rearrangement to identify glycosylated and non-glycosylated forms of hRNase 1 before the Original Fig. 2a (Fig. S5d of the revised version). Results from Figures 3i–k showed a direct binding of hRNase 1 to EphA4; however, EphA4 has not been studied in Figure 2 yet. Thus we did not

move the Original Figs. 3i–k before Figure 2 in the revised manuscript. Alternatively, as we mentioned in the **Response to FIGURES Point #1**, we have followed the reviewer’s suggestion to perform the digestion assay with PNGase F glycosidase and found that treatment with PNGase F in hRNase 1-overexpressing BT-549 cells (BT-549-R1) resulted in a homogenous pattern of hRNase 1 below 20 kDa corresponding to a non-glycosylated hRNase 1 (**New Fig. S5g**; black asterisk, non-glycosylated hRNase 1; red asterisk, glycosylated hRNase 1).

Point #6: *Figure 2k, ref. 27. Better add the methodology in the present manuscript instead of merely referencing to ref.27. Also, the reference here is misleading, as it is unclear whether it includes related results or only the methodology description.*

Response to FIGURES Point #6: We thank the reviewer for the comments. We have described the methodology of *in vivo* limiting dilution assay more clearly under “*Animal studies*” in the Methods section as shown below. In addition, we cited the reference 40 (Original ref. 27, PMID: 19567251) only for the methodology description, and we have rephrased the sentence in the Results section on **page 7**.

“*Animal studies.* For the *in vivo* limiting dilution assay, six-week-old female BALB/c nude mice were purchased from Jackson Laboratories (Bar Harbor, ME, USA). The total number of mice for each experiment is indicated in the figure or table. The indicated number of cells in suspension in 50 μ l of DMEM/F12 (Corning, #10-090-CV) was mixed with 50 μ l of the Matrigel (Thermo Fisher Scientific, #CB40230C). The cell mixtures were subcutaneously injected into the flanks of mice. Tumor incidence was monitored 12 weeks or 15 weeks after inoculation of tumor cells. TIC frequencies of each experiment were estimated using the ELDA web-tool⁴⁰.”

“Notably, results from an *in vivo* xenotransplantation assay⁴⁰ showed that R1 or R1-H12A-expressing BT-549 cells increased tumor-initiating cell (TIC) frequency (Fig. 2g).”

Point #7: *Figure 4: better insert panel b before panel a, to describe the chimera used. Specify the two domains of the receptor and indicate that K653A refers to the receptor phosphorylation site.*

Response to FIGURES Point #7: We thank and have followed the reviewer’s suggestions to rearrange the order of Figure 4. We also included EphA4-ECD and EphA4-ICD chimera in the revised schematic diagram (Original Fig. 4b; Fig. 4a of the revised manuscript), and specified EphA4-ECD, EphA4-ICD, and EphA4-K653A in the Figure Legend as suggested (bold texts shown below).

“**a**, Schematic diagram of various constructs of Myc-tagged EphA4. The numbers represent amino acid residues. **EphA4-ECD, amino acids 1-547 of EphA4; EphA4-ICD, amino acids 570-986 of EphA4; *, EphA4-K653A, mutation of EphA4 phosphorylation site.** **b**, *In vitro* GST pulldown assay of GST-tagged hRNase 1/glutathione magnetic beads incubated with lysate from 293T transfected with the indicated expression plasmids, including WT,

ECD, and ICD of EphA4, and pCDH empty vector, followed by three times of PBS washing. Left, input lysates.”

Original Fig. 4b (Now Fig. 4a)

Original Fig. 4a (Now Fig. 4b)

Point #8: Figure 5 d,e. Indicate what is exactly measured by optical density.

Response to FIGURES Point #8: The values of optical density were the determination for hRNase 1. We have indicated hRNase 1 in the Y-axis of the Figures 5d and 5e. In addition, we have described it more clearly in the Figure Legend in the revised manuscript as shown below.

“**d** and **e**, Binding assay measuring hRNase 1 binding affinity toward EphA4 in BT-549 cells with increasing concentrations of ephrin-A5 (**d**) and KYL or KYL-P7A peptide (**e**) as indicated. The optical density was determined at 450 nm, corrected by subtraction of reading at 570 nm. All error bars represent mean \pm SD.”

Original Fig. 5d

Original Fig. 5e

Point #9: Figure 6: panel b: nomenclature R1-2 can be misleading and is not explained elsewhere. Panels i,j,k. Better order them by alphabetical order within the figure. Figure 6k is incredibly too small. Also, indicate more clearly data from knock down cell line, silencing protocol,.. In panel 6j, indicate each pair analyzed for significance. Why a significance cut-off of $p \leq 0.05$ was not applied? Panels g and h: explain more clearly significance test, mark significant data. Why so many decimals are included?

Response to FIGURES Point #9: We thank the reviewer for the comments. **First**, we have replaced R1-1 with R1#1 and R1-2 with R1#2 throughout the manuscript and figures when we silenced hRNase 1 by shRNA knockdown (e.g. Original Fig. 6b; Fig. 2k of the revised version). **Second**, as we mentioned in the **Response to RESULTS Point #12**, we enlarged the representative tumor images and moved these *in vivo* tumorigenesis results (Original Figs. 6i–6k) to Supplementary Fig. S12a–S12c in the revised manuscript. Furthermore, we followed the reviewer’s suggestion to improve the significance of Original Fig. 6j (Fig. S12c of the revised manuscript) through reanalyzing the original results with more data of mice included. The new results displayed the improved significance of p-values which showed a similar trend as observed in the original findings. In brief, we found that nude mice subcutaneously injected with 4T1-mRNase 1 (n = 6) developed tumors that weighed more than those from mice injected with vector control cells (n = 8; $p = 0.009$). The mRNAse 1-mediated tumorigenesis (n = 6) was suppressed when mice were treated with compound 1 (cpd1) (n = 8), a small molecule that binds to the EphA4-LBD ($p = 0.026$). Collectively, these findings suggested that the hRNase 1-EphA4 axis contributes to breast tumor progression. We have described the new data in the Results section on **page 15** of the revised manuscript. **Third**, regarding the experiments performed by the reconstitution of hRNase 1 in hRNase 1-knockdown KPL4 cells (Original Figs. 6b–6d; Figs. 2k and S5o of the revised version), we have provided a more clear description in the Results section on **page 8** of the revised manuscript. We also included the silencing protocol under “*Generation of hRNase 1 knockdown and reconstitution stable cells*” in the Methods section as shown below. **Last**, in the Original Figs. 6g and 6h (Figs. 6f and 6k of the revised version, respectively), we have marked significant data of TIC frequency with a more detailed explanation in the Results section on **pages 14 and 15** of the revised manuscript as shown below. In addition, we used commas every three decimal places

in numbers of four or more digits in the results of TIC frequency (e.g. 1/16,026 in KPL4-NEO cells and 1/351,989 in BT-549-NEO cells).

“The results were further validated by the reconstitution of hRNase 1 in KPL4 stable clones knocking down hRNase 1 (KPL4-sh-R1#2), in which hRNase 1 was successfully reconstituted into KPL4-sh-R1#2 cells (+ R1 vs. + vector; Fig. S5o), showing restored sphere-forming ability (KPL4-sh-R1#2 + R1 vs. KPL4-sh-R1#2 + vector; Fig. 2k).”

“Generation of hRNase 1 knockdown and reconstitution stable cells. To establish stable cell lines with hRNase 1 knockdown or with the control counterpart, we conducted lentiviral packaging via transient transfection of 1 μ g pGIPZ-sh-R1#1, pGIPZ-sh-R1#2 or pGIPZ-sh-Ctrl together with 1 μ g pCMV-VSVG and 0.5 μ g pCMV-dR8.91 expression plasmids in 5×10^5 293T cells. After 72 hr, 3 ml of conditioned medium from the transfectants were collected, centrifuged at $6,000 \times g$ for 15 min and flew through 0.45 μ m filter, followed by incubating with targeted cells at 5 μ g/ml polybrene for lentiviral transduction. After transduction for 16 hr, cells were replenished with 3 ml of complete medium for one day, and subjected to puromycin selection at 1 mg/ml for another 3 days to establish stable cells. For the reconstitution of hRNase 1 resistant to sh-R1#2-mediated knockdown in the stable cells, we first generated a modified hRNase 1 construct by introducing silent mutations of hRNase1 (from TCCACCTACTGTAACCAA to TCAACATATTGCAATCAA corresponding to the peptide sequence STYCN at amino acids 23-27) on the pCDH-R1 plasmid through site-directed mutagenesis. Then the modified pCDH-R1 with silent mutations and vector control plasmid were utilized in pGIPZ-sh-R1#2-mediated knockdown stable cells through lentiviral transduction as mentioned above, followed by G418 antibiotic selection at 750 μ g/ml to generate stable cells for hRNase1 reconstitution or the control counterpart, respectively.”

“The estimated TIC frequency of EphA4-expressing KPL4 was 3-fold higher than control cells (1/5,082 vs. 1/16,026; KPL4-A4 vs. KPL4-NEO; Fig. 6f), but the increase was attenuated when we knocked out hRNase 1 (1/12,472 vs. 1/4,878; KPL4-A4-KO-R1 vs. KPL4-A4-KO-Ctrl; Fig. 6f).”

“hRNase 1-expressing BT-549 cells initiated tumors more frequently by 6.5-fold compared with control counterparts (1/52,780 vs. 1/351,989; BT-549-R1 vs. BT-549-NEO; Fig. 6k), whereas knocking out EphA4 in hRNase 1-expressing BT-549 cells decreased hRNase 1-mediated TIC frequency (1/412,616 vs. 1/69,591; BT-549-R1-KO-A4 vs. BT-549-R1-KO-Ctrl; Fig. 6k).”

Original Fig. 6b (Now Fig. S5o)

Original Fig. 6b (Now Fig. 2k)

Original Fig. 6i
(Now Fig. S12a)

New Fig. S12b

Original Figs. 6j and 6k
(Now Fig. S12c)

Original Fig. 6g (Now Fig. 6f)

Cell number	KPL4-NEO	KPL4-A4	KPL4-A4-KO-Ctrl	KPL4-A4-KO-R1
1 x 10 ⁵ cells	6/6	6/6	6/6	6/6
1 x 10 ⁴ cells	4/6	6/6	6/6	5/6
5 x 10 ³ cells	1/6	3/6	4/6	1/6
2 x 10 ³ cells	0/8	2/8	1/8	0/8
TIC frequency	1/16,026	1/5,082	1/4,878	1/12,472

Original Fig. 6h (Now Fig. 6k)

Cell number	BT-549-NEO	BT-549-R1	BT-549-R1-KO-Ctrl	BT-549-R1-KO-A4
1 x 10 ⁶ cells	5/6	6/6	6/6	5/6
1 x 10 ⁵ cells	3/8	6/8	5/8	3/8
1 x 10 ⁴ cells	1/8	3/8	3/8	0/8
TIC frequency	1/351,989	1/52,780	1/69,591	1/412,616

Point #10: *Extended figure 1: it could be worth to include in the same panel the results for hRNase1, to facilitate a side-by-side direct comparison. What happened with RNases 8, 9 and 10? Indicate in the text why information is not provided for these RNases.*

Response to FIGURES Point #10: We thank the reviewer for the suggestions. We agree that including the results of hRNase 1 (Fig. 1a) in the same panel of other RNases (Supplementary Figs. S1a and S1b) will facilitates a side-by-side direct comparison. However, hRNase 1 is the leading protein studied in our proposed model, and following the paper publication policy, we may not show the same data simultaneously in the main and supplementary figures. Therefore we would like to keep the original order to demonstrate it in the Main Figure. Our prognostic correlation of survival analyses were based on the database of Kaplan-Meier Plotter, in which the microarray contains 22,277 genes detected for breast cancer prognosis, but RNases 8, 9, and 10 are not included in the analysis (PMID: 20020197; *Breast Cancer Res Treatment*. 2010;123:725-31). We have described this reason in the revised manuscript on **page 4** as bold texts shown below.

“Unexpectedly, only expression of hRNase 1 (Fig. 1a) but not that of the other seven hRNases, including RNases 2, 3, 4, 5, 6, 7, and 11 (Figs. S1a and S1b), exhibited a significant negative correlation with breast cancer patient survival. **Of note, the survival analysis was based on the database of Kaplan-Meier Plotter, in which the microarray contains 22,277 genes detected for breast cancer prognosis, but RNases 8, 9, and 10 are not included in the analysis (PMID: 20020197; *Breast Cancer Res Treatment*. 2010;123:725-31).**”

Point #11: Extended figure 2. Some results seem also significant but the authors are not commenting them. Why differences observed in panel 2d are not mentioned. Is breast cancer the single significant case? How far selection is justified.

Response to FIGURES Point #11: We apologize for not describing the results clearly in the original manuscript. For the purpose of the Original Fig. S2d, we intended to search for other cancer types in addition to breast cancer that may also display poor prognosis associated with high hRNase 1 expression. Therefore, we utilized a public database named Prognoscan to analyze Kaplan-Meier survival curves in multiple cancer types (brain, lung, prostate, skin, bladder, colorectal, ovarian, blood, and head and neck cancers), and then performed the prognosis prediction with low and high hRNase 1 expression. As the results shown in the Original Fig. S2d, we found that hRNase 1 expression correlated with poor prognosis in four cancer types, including astrocytoma, glioma, prostate cancer, and melanoma.

However, considering Prognoscan database does not use the median values for patient stratification, which may not be the most appropriate way for survival curve analysis, we have deleted the description and figures of Prognoscan analysis (Original Figs. S2b and S2d) in the revised manuscript. We replaced them with **New Figs. S3a and S3c**, respectively, assessed by Kaplan-Meier plotter analysis using the median values for patient stratification. In **New Fig. S3a**, we analyzed breast cancer survival curves with hRNase 1 expression by pan-cancer module of Kaplan-Meier plotter in which the gene expression was detected by RNA sequencing. This new data shown high hRNase 1 expression associated with poor overall survival of breast cancer patients was in line with the original data analyzed by Prognoscan (Original Fig. S2b). We further analyzed the prognostic correlation of patient survival with hRNase 1 expression in different cancer types by pan-cancer module of Kaplan-Meier plotter, and similar results were observed in patients with liver hepatocellular carcinoma and esophageal squamous cell carcinoma (**New Fig. S3c**). All of the results in the revised version were analyzed by using the median values for patient stratification.

New Fig. S3a

New Fig. S3c

Point #12: Extended figure 4b. Indicate the number of assays for kinetic measurements. Volumes used for the assay. Graphic values are the average of how many replicates? It is also surprising that no activity is detected in the cell media from HEK cells, as other RNases, such as RNase2, would certainly be expressed in the kidney cell line.

Response to FIGURES Point #12: We thank the reviewer for the suggestions and comments. The data are representative of two independent experiments in triplicate. We have provided more detailed experimental condition of the Original Fig. S4b (Fig. S5c of the revised version) in the Figure Legend section and under “*RNase enzymatic activity assay*” in the Methods section as shown below. Our purpose by using the RNase enzymatic activity assay (Fig. S4b of the original manuscript; Figs. S5c and S5i of the revised version) was to detect the ribonucleolytic activity of hRNase 1 in cells overexpressing hRNase 1-wild type in comparison to those overexpressing hRNase 1-H21A. The CM containing abundant hRNase 1 expression, either wild type or H12A, supposedly contributes to the major population for the detection of ribonucleolytic activity. To acquire an appropriate window for the detection of hRNase 1 activity, we titrated the tested conditioned media (CM) with 1/1,000 dilution by RNase-free water (45 μ l as the total volume of the tested CM) to reduce background RNase activity. We also observed that the substrate of the RNaseAlert Lab Test kit was relatively limited and consumed quickly in the presence of undiluted or less diluted CM. Considering the influence of sample dilution on RNase activity detection, we agree with the reviewer that there would exist a certain level of other endogenous RNase activities in the CM from HEK cells. We have discussed this possibility in the Results section on **page 7** as shown below.

“Analysis of RNase enzymatic activity by an RNaseAlert® Lab Test kit in the conditioned medium (CM) collected from HEK293T transfected with the indicated plasmids. Data are representative of two independent experiments in triplicate. Error bars represent mean \pm SD.”

“*RNase enzymatic activity assay.* Ambion RNaseAlert Lab Test kit (Thermo Fisher Scientific, #AM1964) was used to detect RNase enzymatic activity according to the manufacturer’s instruction. Briefly, 5 μ l of 10-fold RNaseAlert buffer was added to a tube containing the fluorescent substrate, and then mixed with a total of 45 μ l of the tested CM with 1/1,000 dilution by RNase-free water to reduce background RNase activity. The mixture was sequentially placed on a well of a 96-well plate. The real-time fluorescence data were collected at 1-min intervals for 21 min using a BioTek Synergy™ Neo multi-mode reader (BioTek Instruments).”

“It is worthwhile to mention that the CM containing abundant hRNase 1 expression, either wild type or H12A, supposedly contributes to the major population for the detection of ribonucleolytic activity, although there would also exist a certain level of other endogenous RNase activities in the CM from HEK cells.”

Point #13: *Extended figure 7. Indicate significance in panel 7d.*

Response to FIGURES Point #13: We have indicated the significance tested by Student’s t test of the Original Fig. S7d (Fig. S9a of the revised manuscript) in the Figure Legend section as show bold texts below.

“Quantification of spheroid formation assay in BT-549-Ctrl and BT-549-R1 stable clones incubated with the inhibitors against IKK/NF- κ B (QNZ) and MEK/Erk (GSK1120212 and PD0325901). Data are representative of two independent experiments in triplicate. Error bars represent mean \pm SD. * $p < 0.05$, ** $p < 0.01$, Student's t test.”

Original Fig. S7d (Now Fig. S9a)

Point #14: *Extended figure 8 panel a: what is KO control exactly?*

Response to FIGURES Point #14: We thank the reviewer for the comment. We inserted a nontargeting control gRNA sequence (TAAACAAAAAGGAAATAGTT) from the GeCKOv2 libraries, which does not target any human genes based on prediction (PMID: 25075903; *Nat Methods*. 2014;11:783-4), into pLentiCRISPRv2 vector as a control (KO-Ctrl) for knockout experiments. We have included the above information under “*Plasmids, siRNAs, shRNA clones, and knocking out constructs*” in the Methods section in the revised manuscript. Briefly, in the Original Fig. S8a (Fig. S10h of the revised version), EphA4-expressing KPL4 cells with neomycin resistance were subjected to lentiviral transduction with pLentiCRISPRv2-KO-control (KO-Ctrl) and pLentiCRISPRv2-KO-hRNase 1 (KO-R1) to generate EphA4-expressing KO-control cells (A4-KO-Ctrl) and EphA4-expressing hRNase 1-knockout cells (A4-KO-R1) with puromycin resistance, respectively.

DISCUSSION:

Point #1: *first sentence: “showing an unconventional function of hRNase 1 as a secretory ligand of EphA4 to promote breast cancer progression”. It is very important to rephrase this sentence. The word “function” here is not correct.*

Response to DISCUSSION Point #1: We thank the reviewer for the comment. As we mentioned above in the **Response to General comments Point #2**, we have rephrased the first few sentences of the Discussion section in the revised manuscript as bold texts shown below.

“On the basis of our findings, we present a model (Fig. 7f) showing **an unconventional role of hRNase 1 as a secretory ligand of EphA4 that induces breast tumor initiation**. Elevated serum hRNase 1 binds to EphA4 and triggers EphA4 signaling in an autocrine/paracrine manner, which in turn promotes breast cancer initiation via the IKK/NF- κ B and MEK/Erk activating pathways. **In addition to the physiological roles of hRNase 1 involved in hemostasis, inflammation, and innate immunity², the overabundance of hRNase 1 can contribute to cancer progression as a dysregulation disease process.**”

Point #2: Discussion, line 4: Care must be taken when using terms autocrine/paracrine. Here it might be better not to specify as the only thing we know for certain is that action is not juxtacrine.

Response to DISCUSSION Point #2: We thank the reviewer for the comment. To the best of our knowledge, it has been well documented that a soluble ligand, such as EGF and FGF, can activate its cognate receptors on the cells of its origin (autocrine) or in nearby cells (paracrine), whereas a ligand remains membrane-bound, such as ephrins, activates receptors mainly through a juxtacrine manner (PMID: 22622641; PMID: 25772309; PMID: 24003208). In our studies, hRNase 1 as a natively secretory protein freely circulating in various body fluids, e.g., serum and plasma, is demonstrated as a ligand of EphA4 receptor that enhances breast cancer stem-like properties, namely that hRNase 1-mediated EphA4 activation is mainly through an autocrine/paracrine mechanism. However, we do not exclude a possibility that any other unknown pathways are involved as well. We have included this paragraph in the Discussion section on **page 17** in the revised manuscript.

Point #3: Page 12, first line: even if we do not have a cocrystal, we can infer some conclusions from an overall structural comparative analysis. It would certainly be interesting to know if there is any structural similarity between RNases and Ephrins.

Response to DISCUSSION Point #3: We thank the reviewer for the comment and agree that “it would certainly be interesting to know if there is any structural similarity between RNases and Ephrins”. To this end, we utilized T-coffee multiple sequence alignment server (PMID: 10964570; <http://tcoffee.crg.cat/apps/tcoffee/index.html>) to align the primary sequence of hRNase 1 with those for eight classical ligands of EphA4, including ephrin-A1-A5 and B1-B3. Interestingly, we found that hRNase 1 was scored 25, which belonged to a green portion in the colored index, representing unlikely to be correctly aligned with others (**New Fig. A**; see below; not shown in the manuscript). Results from the primary sequence alignment indicated that hRNase 1 only harbors low percentage of consensus regions with those classical EphA4 ligands, suggesting that the hRNase 1 was distinct and unique in comparison to the classical ligands of EphA4. In addition, crystal structural analyses show that EphA4 has a high degree of conformational plasticity in its ligand binding domain (LBD), able to facilitate the ephrin binding and signaling cross class A and B (PMID: 19836338). This significant conformational plasticity of EphA4 as a structural chameleon may provide another molecular basis for hRNase 1 reactivity. We have included the relevant description in the Discussion section on **page 18** as shown below.

“Of note, although EphA4 and EphA5 share high sequence homology and structural similarity, their LBD exhibit distinct ligand-binding specificities through conformational changes⁶¹. In addition, crystal structural analyses show that EphA4 has a high degree of conformational plasticity in its LBD, able to facilitate the ephrin binding and signaling cross class A and B⁵⁹ (PMID: 19836338). This significant conformational plasticity of EphA4 as a structural chameleon may provide another molecular basis for hRNase 1 reactivity. Further investigations and a co-crystal structure analysis would be required to reveal more detailed mechanistic insights toward molecular interactions between EphA4 and hRNase 1 in the future.”

New Fig. A

BAD AVG GOOD

hRNase 1 : 25
 ephrin-A1 : 64
 ephrin-A1_1 : 64
 ephrin-A2 : 64
 ephrin-A3 : 60
 ephrin-A4 : 64
 ephrin-A4_1 : 64
 ephrin-A4_2 : 64
 ephrin-A5 : 63
 ephrin-B1 : 59
 ephrin-B2 : 61
 ephrin-B3 : 59
 cons : 6

```

hRNase 1      MALEKSI-----VRLLL-----LVLIL-LVLGWVQF-SL GKESRAKKFQRQHMDSDSSPSSSSTYCNQ
ephrin-A1_a   MEFL-----WAPL-----LGLCC-----SL-A-AA-----DRHTVFWNS-----SNPK
ephrin-A1_b   MEFL-----WAPL-----LGLCC-----SL-A-AA-----DRHTVFWNS-----SNPK
ephrin-A2     MAPAQR-PLLP LLLLLLPLPPPFARAE---DAAR-A-NS-----DRYAVYWNR-----SNPR
ephrin-A3     MAAAPL-LLLLLLVPVPL-----LP LLA-QQPGG-A-LG-----NRHAVYWNS-----SNQH
ephrin-A4_a   MRLLP-----LLRTVL-----WA-AF-LGSPLR-GGSS-----LRHVYWNNS-----SNPR
ephrin-A4_b   MRLLP-----LLRTVL-----WA-AF-LGSPLR-GGSS-----LRHVYWNNS-----SNPR
ephrin-A4_c   MRLLP-----LLRTVL-----WA-AF-LGSPLR-GGSS-----LRHVYWNNS-----SNPR
ephrin-A5     MLHVEM-LT---LVFLVL-----WM-CVFSQDPGS-KAVA-----DRYAVYWNS-----SNPR
ephrin-B1     MARPGQRWL GKWL VAMVV-----WALCR-LA--TP-L-AK-----NLEPVSWS-----LNPK
ephrin-B2     MAVRRDSVW-KYCWGVI-----MVL CR-TA--IS-K-SI-----VLEPIYWN-----SNPK
ephrin-B3     MGP PHS GPGGVRV GALLL-----LGVLG-LV---S-GL-----SLEPVIWN-----ANKR

cons          *-----:
  
```

```

hRNase 1      MMRR-----RNMTQGRCKPVNTFVHEPLVDVQ-----N--VCFQEKVTCCKNGQGN-CYKS-----
ephrin-A1_a   FRNE-----DYTIHVQLNDYVDIICPHYEDHSV-----A--DAAMEQYILY-----LVEHEEYQLC
ephrin-A1_b   FRNE-----DYTIHVQLNDYVDIICPHYEDHSV-----A--DAAMEQYILY-----LVEHEEYQLC
ephrin-A2     FHAGAGDDGGGYTVEVSINDYLDIYCPHYGAPLP-----P--AERMEHYVLY-----MVN GEGHASC
ephrin-A3     LRRE-----GYTVQVNVNDYLDIYCPHYNSSGVGPAGPGP-----GGGAEQVLY-----MVS RNYRTC
ephrin-A4_a   LLRG-----DAVVELGLNDYLDIVCPHYEGPG-----P--PEGPETFALY-----MVDWPGYESC
ephrin-A4_b   LLRG-----DAVVELGLNDYLDIVCPHYEGPG-----P--PEGPETFALY-----MVDWPGYESC
ephrin-A4_c   LLRG-----DAVVELGLNDYLDIVCPHYEGPG-----P--PEGPETFALY-----MVDWPGYESC
ephrin-A5     FQRG-----DYHIDVCINDYLDVFCPHYEDSV-----P--EDKTERVLY-----MVNFDGYSAC
ephrin-B1     FLSG-----KGLVIYPKIGDKLDIICPRAEAGR-----P--PYEYKLY-----LVRPEQAAAC
ephrin-B2     FLPG-----QGLVLYPQIGDKLDIICPKVDSKT-----P--VGQY EYKLY-----MVDKQDQDRC
ephrin-B3     FQAE-----GGYVLYPQIGDRDLLCPRARPPG-----P--HSSPNYEFYKLY-----LVGGAQGRRC

cons          :-----*-----:
  
```

```

hRNase 1      -----NSSMHITDCRLTNGSRYPNCA YRTSP-----
ephrin-A1_a   QPQ-SKDQVRWQCNRPSAKHGPEKLFSEKQRFRTPF T LGKEFKEGHSY Y I S K P I H Q H E D R-----C
ephrin-A1_b   QPQ-SKDQVRWQCNRPSAKHGPEKLFSEKQRFRTPF T LGKEFKEGHSY Y I S H S P Q A H D N F-----QEKRLAA
ephrin-A2     DHR-QRGFKRWECNRPAAPGGPLKFSEKQFLFTPFSLGFEFRPGHEYYYI S A T P P N A -VDR-P-----C
ephrin-A3     NA--SQGFKRWECNRPHAPSP I K F S E K F Q R Y S A F S L G Y E F H A G H E Y Y I S T--PTH-NLHWK-----C
ephrin-A4_a   QAEGPRAYKRWVCSLPF---GHVQFSEKIQRFTPFSLGFEFLPGETYYYYI S V P T P E S S G Q-----C
ephrin-A4_b   QAEGPRAYKRWVCSLPF---GHVQFSEKIQRFTPFSLGFEFLPGETYYYYI S V P T P E S S G Q-----C
ephrin-A4_c   QAEGPRAYKRWVCSLPF---GHVQFSEKIQRFTPFSLGFEFLPGETYYYYI S V P T P E S S G Q-----C
ephrin-A5     DHT-SKGFKRWECNRPHSPNGPLKFSEKQFLFTPFSLGFEFRPGREYFYI S S A I P D N -GRR-S-----C
ephrin-B1     STV-LDPNVLVTCNRPE---QEIRFTIKFQEFSPNYMGLFKKHHDYI T S T S N G S L -E-----G
ephrin-B2     TIK-KENTPLL NCAKPD---QDIKFTIKFQEFSPNLWGLEFQKNKDYI I S T S N G S L -E-----G
ephrin-B3     EAP-PAPNLLLTCDRPD---LDRFTIKFQ E Y S P N L W G H E F R S H H D Y I I A T S D G T R -E-----G

cons          :-----:
  
```

New Fig. A (cont.)

hRNase 1
 ephrin-A1_a LRLKVT-----VSG-KITHSPQAH-----D
 ephrin-A1_b DDPEVR-----VLH-SIGHSAA-----
 ephrin-A2 LRLKVY-----V-----RP--TN-----ET
 ephrin-A3 LRMKVF-----V-CCASTSHSG-----EKPVP-----LPQF--TMGPNVKINVLEDFEGE
 ephrin-A4_a LRLQVSVCKKERKSESAHF-VGS-P-----
 ephrin-A4_b LRLQVSVCKERRARVL-PRS--PGGG-----G-----
 ephrin-A4_c LRLQVSVCKERNL---P-SH-P-----
 ephrin-A5 LKLVKVF-----V-----RP--TNSCMKTIGVHDRVFDVND-KVENSLEPADDT
 ephrin-B1 LENREGGVCRTTRMKIIM-KVGQDPNA--VTPEQLTTS-----RPSK--EADNTVKMATQAPGSRG
 ephrin-B2 LDNQEGGVCQTRAMKILM-KVGQDASSAGSTRNKDPTR-----RPEL--EAGTNGRSSTTSFFVKP
 ephrin-B3 LESLQGGVCLTRGMKVLL-RVGQSPRGG-AVPRKPVSE-----MP-M--ERDR-GAAHSLEPGKEN

cons

hRNase 1
 ephrin-A1_a -----NPQE-KRLAADDPEVRVL-HS-----IGHSAAPRLF-PLAW
 ephrin-A1_b -----PRLF-PLAW
 ephrin-A2 -----LYEAPPEPIF-T-SNNS-----CS-----SPGGCRL-FLSTIPVL
 ephrin-A3 -----N-----PQVPK-LEKS--ISGTSPKR-----EHLF--LAVGIAF-FLMTFLAS
 ephrin-A4_a -----GESG-----TSGWRGG-----D-----TPS-----PLCLL-LLLLLLIL
 ephrin-A4_b -----IPAACT--GGANSRQDG-----ALMGEIRG
 ephrin-A4_c -----KEPE--SSQDPLEEEG-----SLLP-ALGVPIQT
 ephrin-A5 -----VHESA-----E-PSRGENAQAOT-PR-----IPS--R-LLAILLFL
 ephrin-B1 SLGSDSGKH-----ETVNQ--EEKS--GPCASGGSSGDPDGGFNSKVALFAAVGAGCVI-FLLIIIFL
 ephrin-B2 NPGSSTD-----GNSAGHSGN--NILGSEVALFAGIASGCI-FIVIIITL
 ephrin-B3 LPGDPTSN-----ATSRGAEG--PLPPSPMPAVAGAAGGLAL-LLLGVAGA

cons

hRNase 1
 ephrin-A1_a TV--LLLPLL-----L-
 ephrin-A1_b TV--LLLPLL-----L-
 ephrin-A2 WT--LLGS-----
 ephrin-A3 -----
 ephrin-A4_a RL--LRIL-----
 ephrin-A4_b SEVTLGAC-----P-
 ephrin-A4_c DK--MEH-----
 ephrin-A5 LA--MLLTL-----
 ephrin-B1 TV--LLLKLRKRHRKHTQ-QRAAALSLSLSTLASPK---GGSGTAGTEPSDIIPLRT---TENNYCPH
 ephrin-B2 VV--LLLYRRRHRKHSP-QHTTTLSTSLATPK---RSGNNNGSEPSDIIPLRT---ADSVFCPH
 ephrin-B3 GG--AMC-WRRRRRAKPSERSRHPGPGSFGRRGSLGLGGGGMGPREAEPGELGIALRGGGAADPPFCPH

cons

hRNase 1
 ephrin-A1_a -----KERHIIVACEGSPYVPVHFDASVEDST
 ephrin-A1_b -----LQTP-----
 ephrin-A1_b -----LQTP-----
 ephrin-A2 -----
 ephrin-A3 -----
 ephrin-A4_a -----
 ephrin-A4_b -----LITG-----
 ephrin-A4_c -----
 ephrin-A5 -----
 ephrin-B1 YEKVSGDYGHPVYIVQEMPPQSPANIYYKV---
 ephrin-B2 YEKVSGDYGHPVYIVQEMPPQSPANIYYKV---
 ephrin-B3 YEKVSGDYGHPVYIVQDGGPPQSPANIYYKV---

cons

Point #4: Page 12, line 18: Discussion should include reference to results from glycosylated form and recent work in the field (PMID: 30633504; PMID: 32544330).

Response to DISCUSSION Point #4: We thank the reviewer for the comment. These references have been included on **page 19** in the Discussion section.

Point #5: Page 12, line 4 from bottom: it could be worth to comment on the protein hyperglycosylation observed in some cancer cell lines, such as reported for RNase2 (PMID: 7616105).

Response to DISCUSSION Point #5: We thank the reviewer for the suggestion. We have included the discussion of RNase 2 hyperglycosylation on **pages 19 and 20** in the revised manuscript as follows:

“Although the biological functions of hRNase 1 are not completely defined, its biochemical properties and post-translational modifications, such as glycosylation, have been extensively investigated^{7,67} (PMID: 30633504; PMID: 32544330). There are three N-linked glycosylation sites of hRNase 1 reported (Asn-34, Asn-76, and Asn-88), and the pattern of hRNase 1 on gel electrophoresis is usually heterogeneous due to heavy glycosylation as illustrated by a range of migrated bands (~15 to 36 kDa)^{6,7} (PMID: 32544330; PMID: 12626415). In addition, hRNase 1 exhibits differential levels of glycosylation in the urine, seminal plasma, kidney, and brain^{5,68}, raising a possibility that glycosylation may affect certain tissue-specific roles of hRNase 1. Moreover, the N-linked glycan on Asn 88 of serum hRNase 1 acts as a diagnostic marker for pancreatic cancer⁴⁰. Thus, it would be of interest to further explore whether different glycosylated forms of hRNase 1 associate with any specific roles during cancer progression. **It is worthwhile to note that, in addition to hRNase 1, hyperglycosylation of RNase proteins in cancer cell lines was also observed in human RNase 2 in promyelocytic leukemia cells; treatment of PNGase F glycosidase resulted in a change of RNase 2 from a heterogeneous to a homogenous pattern, in terms of molecular weight from ~22–45 kDa reduced to ~15 kDa, respectively⁶⁹ (PMID: 7616105). Together, these studies have suggested that RNase protein hyperglycosylation would be a common feature with heterogeneity shared in cancer cells.”**

Point #6: Page 12, line 2 from bottom: change “function” with “role”.

Response to DISCUSSION Point #6: We have replaced “functions” with “roles” as follows (**page 19** in the Discussion section of the revised manuscript):

“Thus, it would be of interest to further determine whether different glycosylated forms of hRNase 1 associate with specific **roles** during cancer progression.”

Point #7: The authors conclude that glycosylations are not altering the protein interaction with the receptor. They discard an enhancement of activity, but the possibility should be also considered that the glycosylation might cover the receptor binding site and reduce the interaction.

Response to DISCUSSION Point #7: We thank and agree with the reviewer’s insightful comment. Using an immune checkpoint PD-L1 as an example, heavy *N*-linked glycosylation of PD-L1 indeed causes structural hindrance for the interaction with and subsequent detection by PD-L1 antibodies; removing the glycan moieties from the PD-L1 antigen via PNGase F glycosidase digestion significantly improves PD-L1 signal intensity and binding affinity (PMID: 31327656). Thus hRNase 1 glycosylation with a heterogeneous pattern in general is likely to affect its binding affinity to EphA4 at different levels under various cellular, physiological, or pathological conditions. To avoid misleading statements, we modified the description on **page 10** in the revised manuscript as shown bold texts below.

“Considering hRNase 1 as a *N*-linked glycosylated protein⁵⁻⁷, we performed *in vitro* binding assay by incubating N-EphA4-Fc with recombinant hRNase 1 protein produced from HEK293 cells, which we pretreated with glycosidase PNGase F (Fig. 3i), and found that non-glycosylated hRNase 1 still harbored the binding ability to N-EphA4-Fc (Fig. 3k). These results **suggested that the *N*-linked glycosylation is not required for binding to N-EphA4-Fc; however, we do not rule out a possibility that heterogeneous glycosylation of hRNase 1 may contribute to its binding to EphA4 at different levels under various cellular, physiological, or pathological conditions.**”

Point #8: Discussion is a bit poor and too short. It could be worth to comment recent data on RNase 5 and egfr receptor, discuss the results from the screening using the phosphoreceptors kit, the regions identified to be involved in interaction by site directed mutagenesis of protein and receptor, etc.

Response to DISCUSSION Point #8: We thank the reviewer for the suggestion. We have included recent studies of the hRNase 5-EGFR axis on **page 20** in the Discussion section.

“It is worthwhile to mention that another RNase protein, called hRNase 5, has been known as a ligand of EGFR RTK in pancreatic cancer as evidenced by the results of a phospho-RTK antibody array (PMID: 29606349), which is a similar approach used to identify the hRNase 1-EphA4 pair. Analysis through primary sequence alignment between hRNase 5 and EGF, an EGFR classical ligand, shows that a conserved residue on glutamine 93 (Q93) of the C-terminus of hRNase 5 is critical for EGFR binding, and EGFR-ECD is required for binding to hRNase 5 (PMID: 29606349). In line with these results, we demonstrated that hRNase 1 binds to EphA4 via its C terminus and the interaction requires EphA4-ECD, suggesting an intriguing mechanism that the C-terminal region of hRNases may generally participate in the essential binding of their cognate RTKs-ECD. Of note, the ribonucleolytic activity of hRNase 5 is not required for EGFR ligand-like function (PMID: 29606349); likewise we found that hRNase 1 increased breast cancer stem-like properties independently of its ribonucleolytic activity. Recently, human RNase 7 has been found to serve as a high-affinity ligand for ROS1 RTK in hepatocellular carcinoma (PMID: 33031845). Together, our findings with another two studies including hRNase 5-EGFR and hRNase 7-ROS1 pairs uncover important roles of secretory RNases in human malignancies, which may help to create a

paradigm shift in the understanding of the ligand-receptor relationship between secretory hRNase and cell membrane RTK families.”

METHODS:

Point #1: Page 35, elisa. “briefly the plates were precoated with Elisa plates are prepared manually or they are provided with the kit? Specify otherwise the antibodies used.

Response to METHODS Point #1: We thank the reviewer for point this out. Pre-coated and ready to use 96-well ELISA plates are provided with the kit (Cloud-Clone Corp., #SEA297Hu).

“*Detection of hRNase 1 in human serum and CM by ELISA.* Serum collection from breast cancer patients and healthy individuals was approved by Institutional Review Board of MD Anderson Cancer Center and informed consent was obtained from all subjects. The concentration of hRNase 1 in serum or CM was determined by ELISA Kit for hRNase 1 (Cloud-Clone Corp., #SEA297Hu) according to the manufacturer’s instructions. Briefly, standards or samples were added to the appropriate wells of a **pre-coated and ready to use 96-well plate** for 1 h at 37 °C, followed by incubating with a biotin-conjugated antibody against hRNase 1 (Detection Reagent A) for 1 h at 37 °C. After washing, HRP-conjugated avidin (Detection Reagent B) was added to each well, and the mixture was incubated for 1 h at 37 °C. After washing, 90 µl of TMB substrate solution was added to each well for 10 to 20 min at 37 °C. Finally, the enzyme-substrate reaction was stopped by 50 µl of sulfuric acid solution. Only the well added with hRNase 1, biotin-conjugated antibody, and HRP-conjugated Avidin displayed a change of color, which was subsequently measured at a wavelength of 450 nm by a BioTek Synergy™ Neo multi-mode reader (BioTek Instruments). The concentration of hRNase 1 in the samples was calculated by comparing the OD value of the samples to the standard curve.”

Point #2: Page 38: indicate the antibodies against phosphorylated receptors that nomenclature is referred to tyr residue,...

Response to METHODS Point #2: We thank the reviewer for the comment. As we mentioned above in the **Response to RESULTS Point #4**, we performed human phospho-RTK antibody array (R&D Systems, #ARY001B) by using a pan anti-phospho-tyrosine antibody conjugated to HRP for the detection of phosphorylated tyrosines bound to the captured RTKs. There is no individual residue of tyrosine phosphorylation provided in this assay. In the revised manuscript, we have made a more clear statement under “*Human phospho-RTK antibody array*” in the Methods section as shown below.

“*Human phospho-RTK antibody array.* Proteome Profiler Human Phospho-RTK Array Kit (R&D Systems, #ARY001B) was used to detect the potential activation of RTK signals by hRNase 1 treatment. All procedures were performed according to the manufacturer’s instruction. Briefly, capture antibodies for specific RTKs were spotted in duplicate onto nitrocellulose membranes, prepared with the kit. Cell lysates (600 µg) were incubated with the array membrane at 4°C overnight. After washing away unbound material, proteins in

cell lysates containing phosphorylated tyrosine residues bound to the capture antibodies of RTKs were **detected by a pan anti-phospho-tyrosine antibody conjugated to horseradish peroxidase**. Last, the binding signal was measured by using chemiluminescent detection reagents and ImageQuant LAS 4010 (GE Healthcare).”

REVIEWER COMMENTS

Reviewer #1 (Remarks to the Author):

The reviewer understands that it is difficult to use patient-derived cancer cells, though it is very important to strengthen their conclusion. However, following points should be addressed carefully. Fig. 1a, b, c and New Fig. S3

The authors described that they analyzed patient's prognosis using the median values for patient stratification. If so, the number of patients with high hRNase1 and low hRNAase1 should be similar. Please carefully check the results.

Fig. 2j, k

They described that they performed extreme limiting dilution analysis (ELDA). The reviewer noticed that they inoculated 1×10^5 cells for BT549 cells and 1×10^4 cells for KPL4 cells only. For limiting dilution assay, they should inoculate several numbers of serially diluted cells for each cell line, 105 cells, 104 cells, 103 cells for example.

Reviewer #2 (Remarks to the Author):

The authors have been responsive to comments and concerns. The manuscript is clear, comprehensive, the work is novel, and these findings are likely to be highly impactful and of great interest.

Reviewer #3 (Remarks to the Author):

The authors have performed the required additional experiments and addressed properly all the comments. Only few sentences might be revised for better clarity.

For example:

"RNases and RTKs, which are considered as two unrelated families, have been recently discovered the significance of ligand-receptor relationship in pancreatic and liver cancers"

the sentence should be rephrased. I suggest:

"RNases and RTKs, which were considered as two unrelated families, have been recently revealed the significance of a novel ligand-receptor relationship in pancreatic and liver cancers"

Sentence below read better by adding THAT (as highlighted):

To the best of our knowledge, it has been well documented that a soluble ligand, such as EGF and FGF, can activate its cognate receptors on the cells of its origin (autocrine) or in nearby cells (paracrine), whereas a ligand

THAT remains membrane-bound, such as ephrins, activates receptors mainly through a juxtacrine manner.

Point-by-Point Response to the Reviewers' Comments
NCOMMS-20-19685A

Reviewer #1 (Remarks to the Author):

The reviewer understands that it is difficult to use patient-derived cancer cells, though it is very important to strengthen their conclusion. However, following points should be addressed carefully. Fig. 1a, b, c and New Fig. S3

The authors described that they analyzed patient's prognosis using the median values for patient stratification. If so, the number of patients with high hRNase1 and low hRNAase1 should be similar. Please carefully check the results.

Fig. 2j, k

They described that they performed extreme limiting dilution analysis (ELDA). The reviewer noticed that they inoculated 1×10^5 cells for BT549 cells and 1×10^4 cells for KPL4 cells only. For limiting dilution assay, they should inoculate several numbers of serially diluted cells for each cell line, 105 cells, 104 cells, 103 cells for example.

Authors' Response: We appreciate the reviewer's constructive criticism to further strengthen our findings and improve the scientific merits of this revised manuscript. We have carefully addressed each of the remaining issues as shown below in a point-by-point manner.

Point #1: *Fig. 1a, b, c and New Fig. S3*

The authors described that they analyzed patient's prognosis using the median values for patient stratification. If so, the number of patients with high hRNase1 and low hRNAase1 should be similar. Please carefully check the results.

Response to Point #1: We thank the reviewer for the comment and apologize for making this mistake due to an oversight. We have carefully checked all of the relevant results (Original Figs. 1a, 1b, 1c, S1, S2, and S3) and replaced them with the updated database analyses using the median values for patient stratification in the revised manuscript.

Consistent with the original results which patients were split by the Auto select best cutoff setting from the Kaplan-Meier plotter database (<http://kmplot.com/analysis/>), the new results using the median values for patient stratification also demonstrated that only expression of hRNase 1 (**New Fig. 1a**; OS, overall survival; RFS, the relapse-free survival; DMFS, distant metastasis free survival) exhibited a significant negative correlation with breast cancer patient survival, but not that of the other seven hRNases, including hRNases 2, 3, 4, 5, 6, 7, and 11 (OS in **New Fig. S1a**; RFS in **New Fig. S1b**). We would like to draw the reviewer's attention to the RFS analysis of hRNase 11 as we have carefully checked and confirmed the result using the median value from the Kaplan-Meier plotter database, although the number of patients with low hRNase 11 ($n = 995$) and high hRNase 11 ($n = 769$) provided are not quite similar (**New Fig. S1b**). We speculate that the number difference between low and high groups of hRNase 11 may come from a highly asymmetric distribution of probe gene sets. In addition, we reanalyzed the Original Figs. 1b and 1c using the median cutoff setting and found that higher hRNase 1 expression correlated with poorer survival in tumors with grade 2 in a near-significant trend ($p = 0.056$, **New Fig. 1b**),

and hRNase 1 appeared to be a more significant poor prognostic factor in breast cancer patients with lymph node (LN)-negative non-metastasis, compared with those with LN-positive metastasis ($p = 0.015$ versus $p = 0.056$, **New Fig. 1c**).

Next, we reanalyzed the Original Fig. S3 regarding the prognostic correlation of hRNase 1 levels with patient survival in patients with different cancer types. In the Original Figs. S3a and S3c using the Auto select best cutoff setting (<http://kmplot.com/analysis/>), we showed that high hRNase 1 expression correlates with poor patient survival in patients with breast cancer (Original Fig. S3a) as well as liver hepatocellular carcinoma and esophageal squamous cell carcinoma (Original Fig. S3c). However, using the median values for patient stratification only displayed statistical significance in patients with liver hepatocellular carcinoma, but not breast cancer and esophageal squamous cell carcinoma (**New Figs. A and B**; see below; not shown in the manuscript). To prevent cutoff selection bias from misleading the readers, if the reviewer agrees, we would like to focus on the prognostic correlation of hRNase 1 levels with patient survival in patients with breast cancer and withdraw the Original Figs. S3a and S3c from the current manuscript, which requires further investigations into analyses of breast cancer in comparison with other cancer types.

In the revised manuscript, we further strengthened and demonstrated our findings that higher hRNase 1 expression associated with poorer patient survival in patients with breast cancer in another independent database called GENT2 (<http://gent2.appex.kr/gent2/>) (**New Fig. S3a**). Moreover, we reanalyzed the Original Fig. S3b which breast cancer patients were originally split by the mean value from the UCSC Cancer Genome Browser (<http://xena.ucsc.edu/welcome-to-ucsc-xena/>) using the interpreted expression profile of TCGA breast invasive carcinoma by RNA sequencing (Original Fig. 3b). We replaced the original data with the new analysis using the median value and similar results were obtained (**New Fig. S3b**). We have rephrased the subtitle and updated the information with references of **New Fig. S3** in the Figure Legends and Methods sections in the revised manuscript as bold texts shown below.

*“Figure Legends. **Supplementary Fig. 3. High hRNase 1 expression correlates with poor patient survival in two other independent databases of patients with breast cancer. a, Prognostic correlation of the OS of breast cancer patients with high and low hRNase 1 expression divided by the median value, analyzed the Kaplan-Meier Plotter from the GENT2 database (<http://gent2.appex.kr/gent2/>). b, Prognostic correlation of the OS of breast cancer patients with high and low hRNase 1 expression divided by the median value, analyzed the Kaplan-Meier survival curve from the UCSC Cancer Genome Browser (<http://xena.ucsc.edu/welcome-to-ucsc-xena/>) using the interpreted expression profile of TCGA breast invasive carcinoma by RNA sequencing (dataset ID: TCGA_BRCA_exp_HiSeqV2).”***

*“Methods. **Prognostic analysis of cancer patients from databases.** The Kaplan-Meier plotter database (<http://kmplot.com/analysis/>)³⁵ was used to analyze the correlation between the expression levels of hRNase family and survival of cancer patients. **A Kaplan-Meier overall survival analysis of breast cancer patients divided by the median***

expression level of hRNase 1 was performed in a platform for exploring Gene Expression patterns across Normal and Tumor tissues named GENT2 (<http://gent2.appex.kr/gent2/>)³⁷. The UCSC Cancer Genome Browser (<http://xena.ucsc.edu/welcome-to-ucsc-xena/>)³⁸ was utilized to validate prognostic effect of hRNase 1 with Kaplan-Meier survival analysis. In brief, survival analysis was performed using the interpreted expression profile of TCGA breast invasive carcinoma by RNA sequencing (dataset ID: TCGA_BRCA_exp_HiSeqV2) downloaded from the UCSC Cancer Genome Browser. The median expression of hRNase 1 was used for patient stratification. A corrected p-value < 0.05 was considered as significant.”

“References. ³⁵Gyorffy, B. *et al.* An online survival analysis tool to rapidly assess the effect of 22,277 genes on breast cancer prognosis using microarray data of 1,809 patients. *Breast Cancer Res Treat* **123**, 725-731 (2010). ³⁷Park, S.J., Yoon, B.H., Kim, S.K. & Kim, S.Y. **GENT2: an updated gene expression database for normal and tumor tissues. *BMC Med Genomics* 12, 101 (2019).** ³⁸Goldman, M.J. *et al.* Visualizing and interpreting cancer genomics data via the Xena platform. *Nat Biotechnol* **38**, 675-678 (2020).”

In addition to the figures commented by the reviewer (*Fig. 1a, b, c and New Fig. S3*), we further replaced the Original Fig. S2a with the new data using the median values (<http://kmplot.com/analysis/>) (**New Fig. S2a**). Similar results that hRNase 1 correlated negatively with patient survival were obtained in breast cancer patients with the basal-like, luminal A, and luminal B subtypes, in a trend towards statistical significance ($p = 0.055$, $p = 0.018$, $p = 0.0008$, respectively). These new results together with the Original Figs. S2b and S2c acquired by the median values, we certified that all of the prognostic correlation of survival analyses in this revised manuscript were analyzed by using the median values for patient stratification (**Figs. 1a–1c, S1a, S1b, S2a–S2c, S3a, and S3b**).

New Fig. 1a

New Fig. 1b

New Fig. 1c

New Fig. S1a

New Fig. S1b

Original Fig. S3a
(Auto select best cutoff)

Original Fig. S3c
(Auto select best cutoff)

New Fig. A
(Median cutoff)

New Fig. B
(Median cutoff)

New Fig. S3a

New Fig. S3b

New Fig. S2a

Original Fig. S2b

Original Fig. S2c

Point #2: Fig. 2j, k

They described that they performed extreme limiting dilution analysis (ELDA). The reviewer noticed that they inoculated 1×10^5 cells for BT549 cells and 1×10^4 cells for KPL4 cells only. For limiting dilution assay, they should inoculate several numbers of serially diluted cells for each cell line, 105 cells, 104 cells, 103 cells for example.

Response to Point #2: We thank the reviewer for the comment. We apologize that we overinterpreted the original results for using one dose of cell number to estimate the tumor-initiating cell (TIC) frequency through the web-based ELDA statistical software at <http://bioinf.wehi.edu.au/software/elda/index.html> (⁴²Hu, Y. and Smyth, G.K. *J Immunol Methods* 347, 70-78, 2009). We also apologize for only showing the representative tumor images of 1×10^5 for BT-549 stable clones and 1×10^4 for KPL-4 stable clones in the original Figs. 2g and 2j (Fig. S5g and Fig. S6c of the revised version, respectively). In the revised manuscript, we have included our original results completely with three dosages of serially diluted cell number as the reviewer advised. We re-estimated the TIC frequency calculated using ELDA through inoculation of three serially diluted cell dosages for BT-549 stable clones (1×10^6 , 1×10^5 , 1×10^4 cells; **New Fig. 2g**) and KPL4 stable clones (1×10^5 , 1×10^4 , 1×10^3 cells; **New Fig. 2j**). In the **New Fig. 2g**, the TIC frequencies were 1/498,951 for BT-549-Ctrl, 1/129,357 for BT-549-R1, and 1/97,089 for BT-549-R1-H12A. The p values of pairwise comparisons for differences in TIC frequencies using the ELDA web-tool⁴² between BT-549-Ctrl *versus* BT-549-R1 and BT-549-Ctrl *versus* BT-549-R1-H12A were 0.031 and 0.008, respectively ($p < 0.05$, statistically significant, Chi-squared test), whereas there was no significant difference in TIC frequencies between BT-549-R1 and BT-549-R1-H12A ($p = 0.641$). The results demonstrated that hRNase 1 overexpression significantly increased TIC frequency independently of its ribonucleolytic activity, which is in line with our original conclusion. In the **New Fig. 2j**, the TIC frequencies were 1/7,057 for KPL4-sh-Ctrl and 1/45,903 for KPL4-sh-R1#2 with a significant p value displayed between these two group ($p = 0.006$), indicating that knocking down hRNase 1 significantly reduced the TIC frequency. Together, the results from an *in vivo* limiting dilution assay in a subcutaneous mouse model supported the role of hRNase 1 in contributing to stem-like properties of breast cancer.

New Fig. 2g

Cell number	BT-549-Ctrl	BT-549-R1	BT-549-R1-H12A
1 x 10 ⁶ cells	5/6	6/6	6/6
1 x 10 ⁵ cells	2/8	4/8	5/8
1 x 10 ⁴ cells	0/8	1/8	1/8
TIC frequency	1/498,951	1/129,357	1/97,089
p (BT-549-Ctrl)	–	0.031	0.008
p (BT-549-R1)	0.031	–	0.641

New Fig. 2j

Cell number	KPL4-sh-Ctrl	KPL4-sh-R1#2
1 x 10 ⁵ cells	6/6	5/6
1 x 10 ⁴ cells	5/6	2/6
1 x 10 ³ cells	0/6	0/6
TIC frequency	1/7,057	1/45,903
p (KPL4-sh-Ctrl)	–	0.006

Original Fig. 2g (Now Fig. S5g)

Original Fig. 2j (Now Fig. S6c)

Reviewer #2 (Remarks to the Author):

The authors have been responsive to comments and concerns. The manuscript is clear, comprehensive, the work is novel, and these findings are likely to be highly impactful and of great interest.

Authors' Response: We would like to thank the reviewer for recognizing the extensive effort we put into revising the manuscript and the importance of our study. The reviewer's constructive suggestions and comments further strengthen our findings and improve the scientific merits of the manuscript.

Reviewer #3 (Remarks to the Author):

The authors have performed the required additional experiments and addressed properly all the comments. Only few sentences might be revised for better clarity.

For example:

"RNases and RTKs, which are considered as two unrelated families, have been recently discovered the significance of ligand-receptor relationship in pancreatic and liver cancers" the sentence should be rephrased. I suggest:

"RNases and RTKs, which were considered as two unrelated families, have been recently revealed the significance of a novel ligand-receptor relationship in pancreatic and liver cancers"

Sentence below read better by adding THAT (as highlighted):

To the best of our knowledge, it has been well documented that a soluble ligand, such as EGF and FGF, can activate its cognate receptors on the cells of its origin (autocrine) or in nearby cells (paracrine), whereas a ligand THAT remains membrane-bound, such as ephrins, activates receptors mainly through a juxtacrine manner.

Authors' Response: We would like to thank the reviewer for recognizing the extensive effort we put into revising the manuscript. We have followed the reviewer's suggestions to revise the sentences on **pages 4 and 17** of the revised manuscript as shown below, respectively.

*"RNases and RTKs, **which were considered as two unrelated families, have been recently revealed the significance of a novel ligand-receptor relationship** in pancreatic and liver cancers^{14,34}."*

*"To the best of our knowledge, it has been well documented that a soluble ligand, such as EGF and FGF, can activate its cognate receptors on the cells of its origin (autocrine) or in nearby cells (paracrine)^{54,55}, whereas a ligand **that** remains membrane-bound, such as ephrins, activates receptors mainly through a juxtacrine manner²⁰."*

REVIEWERS' COMMENTS

Reviewer #1 (Remarks to the Author):

All my concerns are adequately addressed by the authors. The manuscript is now in a good shape for publication.

Point-by-Point Response to the Reviewers' Comments
NCOMMS-20-19685B

Reviewer #1 (Remarks to the Author):

All my concerns are adequately addressed by the authors. The manuscript is now in a good shape for publication.

Authors' Response: We would like to thank the reviewer for recognizing the extensive effort we put into revising the manuscript. The reviewer's constructive suggestions and comments further strengthen our findings and improve the scientific merits of the manuscript.